



# Horizontal distribution of tropospheric NO₂ and aerosols derived by dual-scan multi-wavelength MAX-DOAS measurements in Uccle, Belgium

Ermioni Dimitropoulou[1], François Hendrick[1], Martina M. Friedrich[1], Frederik Tack[1], Gaia Pinardi[1], Alexis Merlaud[1], Caroline Fayt[1], Christian Hermans[1], Frans Fierens[2] and Michel Van Roozendael[1]

[1] Royal Belgian Institute for Space Aeronomy (BIRA-IASB), Brussels, 1180, Belgium
[2] IRCEL-CELINE, Brussels, Belgium

*Correspondence to*: Ermioni Dimitropoulou (ermioni.dimitropoulou@aeronomie.be)

## Abstract

Dual-scan ground-based Multi-AXis Differential Optical Absorption Spectroscopy (MAX-DOAS) measurements of tropospheric nitrogen dioxide (NO₂) and aerosols were carried out in Uccle (50.8°N, 4.35°E; Brussels region, Belgium) for two years, from March 2018 to February 2020. The MAX-DOAS instrument was operating in both UV and Visible wavelength ranges in a dual-scan configuration consisting of two sub-modes: (1) an elevation scan in a fixed viewing azimuthal direction and (2) an azimuthal scan in a fixed low elevation angle (2°). By analyzing the O₄ and NO₂ dSCDs at six different wavelength intervals along every azimuthal direction and by applying a new Optimal-Estimation-based inversion approach, the horizontal distribution of the NO₂ near-surface concentrations and vertical column densities (VCDs) and the aerosols near-surface extinction coefficient are retrieved along ten azimuthal directions. The retrieved horizontal NO₂ concentration profiles allow the identification of the main NO₂ hotspots in the Brussels area. Correlative comparisons of the retrieved horizontal NO₂ distribution were conducted with airborne, mobile, and satellite datasets, and overall a good agreement is found. The comparison with TROPOMI observations reveals that the characterization of the horizontal distribution of tropospheric NO₂ VCDs by ground-based measurements, the appropriate sampling of TROPOMI pixels, and an adequate a priori NO₂ profile shape in TROPOMI retrievals lead to a better consistency between satellite and ground-based datasets.

## 1 Introduction

Aerosols and nitrogen dioxide (NO₂) play a crucial role in the tropospheric chemistry. NO₂ is an important tropospheric pollutant mainly emitted by combustion processes and nitrogen fertilizers used in agriculture (Seinfeld and Pandis, 1998). Traffic, domestic heating, industrial activities, and power plants are the largest NO₂ emitters (Tack et al., 2021). Beyond its harmful effects on human health (Chen et al., 2007), NO₂ participates in the formation of tropospheric ozone (O₃) by a non-linear photochemical mechanism which involves volatile organic compounds (VOCs).



Aerosols with small diameter are estimated to cause millions of premature deaths per year globally because of their ability to
penetrate deeply into the lungs (Khomenko et al., 2021). Aerosols influence the Earth's climate system by changing its
radiation budget by scattering and absorbing sunlight (Quaas et al., 2008). In the boundary layer of urban regions, the horizontal
distribution of $NO_2$ is highly heterogeneous given the fact that it is a short-lived species (Beirle et al., 2003). For those reasons,
the regional and global monitoring of $NO_2$ and aerosols at high spatial resolution is crucial.

Since 1995, with the ERS-2 GOME (Global Ozone Monitoring Experiment) instrument (Burrows et al., 1999), satellite nadir
air-quality measurements of atmospheric backscattered sunlight in the UV-visible range have provided daily global
tropospheric column measurements of numerous trace gases, such as $NO_2$. Many satellite missions dedicated to air-quality
monitoring followed over the next years with increasing spatial resolution. More recently, the TROPOspheric Monitoring
Instrument (TROPOMI) sensor launched onboard the Sentinel-5P Precursor (S5P) platform in October 2017 reached an initial
spatial resolution of 7x3.5 $km^2$, and augmented on 6 August 2019 to 5.5x3.5 $km^2$. Due to TROPOMI's fine spatial resolution,
monitoring the horizontal distribution of $NO_2$ in urban regions and identifying specific emission sources is made easier than
with previous satellite missions but still, TROPOMI cannot fully capture the fine-scale (sub-kilometer) structures in the
effective $NO_2$ field. Consequently, TROPOMI requires further attention concerning its measurements validation.

Tropospheric vertical columns of many trace gases like $NO_2$, formaldehyde (HCHO), sulphur dioxide ($SO_2$), nitrous acid
(HONO) and $O_3$ can be retrieved by the Multi-AXis Differential Optical Absorption Spectroscopy (MAX-DOAS) technique
(Hönninger et al., 2004; Wittrock et al., 2004; Pinardi et al., 2008, 2013; Clémer et al., 2010; Hendrick et al., 2014; Irie et al.,
2011, 2012; Sinreich et al., 2007; Wagner et al., 2011; Wang et al., 2018). In recent years, MAX-DOAS measurements have
been widely used as reference datasets for the validation of nadir airborne and space-borne air-quality measurements. MAX-
DOAS instruments measure the scattered sunlight in the UV and Visible spectral ranges at multiple elevation angles above the
horizon. For absorbers located close to the surface, such as tropospheric $NO_2$, the higher sensitivity is achieved for low MAX-
DOAS elevation angles. During the last years, MAX-DOAS measurements in more than one azimuthal direction are emerging
(Ortega et al., 2016; Wang et al., 2014; Chan et al., 2020; Schreier et al., 2021). Multi-azimuthal MAX-DOAS measurements
offer many possibilities on air-quality monitoring, such as a better characterization of the effective $NO_2$ field around the station.
These ground-based datasets can be valuable for validating satellite missions with fine spatial resolution in regions where the
$NO_2$ horizontal distribution is heterogeneous, such as urban and sub-urban areas.

In this study, a new aerosol and $NO_2$ horizontal distribution inversion approach based on two years (March 2018-February
2020) of dual-scan multi-wavelength MAX-DOAS measurements in Uccle (Brussels-Capital region, Belgium) is presented.
In every azimuthal viewing direction, parameterized $NO_2$ near-surface concentrations, $NO_2$ tropospheric columns and aerosol
extinctions measured at six different wavelengths are used as input in a new horizontal distribution inversion approach. On
this basis, the near-surface aerosol extinction and $NO_2$ horizontal distributions are retrieved at a spatial resolution of about
3km in a range of about 20 km around the measurement site. These horizontal profiles are used to validate collocated
TROPOMI tropospheric $NO_2$ columns. One complete year of data (March 2018-March 2019) and two wavelength intervals
(one in the UV and one in the Visible) have already been used in Dimitropoulou et al. (2020). It is proven that multi-azimuthal



(the so-called dual-scan) MAX-DOAS measurements significantly improve the agreement between ground-based and TROPOMI tropospheric $NO_2$ column observations over the Brussels' area. By adding the multi-wavelength aspect, the present

work represents an extension of the former study.

The manuscript is organized into six sections: in Sect. 2, the measurement site with the MAX-DOAS experimental set-up and the multi-wavelength DOAS analysis are presented. In Sect. 3, the TROPOMI tropospheric $NO_2$ measurements are described. Section 4 is composed of two main parts: Sect. 4.1 is a detailed description of the dual-scan multi-wavelength MAX-DOAS retrieval method and Sect. 4.2 is the horizontal aerosol and $NO_2$ distribution inversion approach. In Sect. 5, main results

followed by correlative comparisons of the retrieved ground-based and satellite horizontal $NO_2$ distribution are presented. Finally, in Sect. 6, conclusions and future perspectives are given.

## 2 Dual-scan multi-wavelength MAX-DOAS measurements

### 2.1 Measurement site and experiment set-up

Brussels-Capital Region is the most densely populated area in Belgium, where pollutant concentrations, such as $NO_2$, are often

high because of anthropogenic activities (Tack et al., 2021).

A MAX-DOAS dual-scan instrument was operated by BIRA-IASB (Koninklijk Belgisch Instituut voor Ruimte-Aeronomie – Institut Royal d'Aeronomie Spatiale de Belgique) in Uccle from January 2017 to February 2020. Uccle is located to the South of the city-center of Brussels and to the West of a large forested area (Bois de la Cambre). Therefore, it is an ideal site to perform MAX-DOAS observations under moderate to high pollution level conditions. Additionally, the characterization of the

horizontal distribution of $NO_2$ and aerosols at high spatial resolution is of great interest here because of the heterogeneity of the pollution sources (car traffic, national airport, power plant) in the capital region of Brussels.

The MAX-DOAS dual-scan instrument is composed of the following parts: the optical head mounted on a sun tracker, two spectrometers (UV and Visible) inside a thermo-regulated box and the data-acquisition unit. The optical head and the two spectrometers are connected with optical fibers. A more detailed description of the BIRA-IASB MAX-DOAS dual-scan

instrument can be found in Dimitropoulou et al. (2020).

From March 2018 to February 2020, the MAX-DOAS instrument operated in a dual-scan viewing mode. Two different sub-modes compose one complete measurement scan (see Dimitropoulou et al., 2020): (1) a vertical scan in nine different elevation angles (EAs) in one fixed telescope azimuthal direction (TAA; Northeast direction i.e., towards the city center and the national airport) and (2) a horizontal scan in nine different azimuthal directions at a fixed elevation angle (2° above the horizon).

Several azimuthal viewing directions were tested to obtain an optimal horizontal sampling without any obstacle in the different viewing directions (see Fig. 1). The selection of more azimuthal directions towards the North, Northeast, and Northwest directions was made considering the location of the main $NO_2$ emission sources and, consequently, the highly variable $NO_2$ horizontal distribution towards these directions.

The integration time for each measured radiance spectrum is 60s, resulting in a full scan duration of approximately 20 minutes.


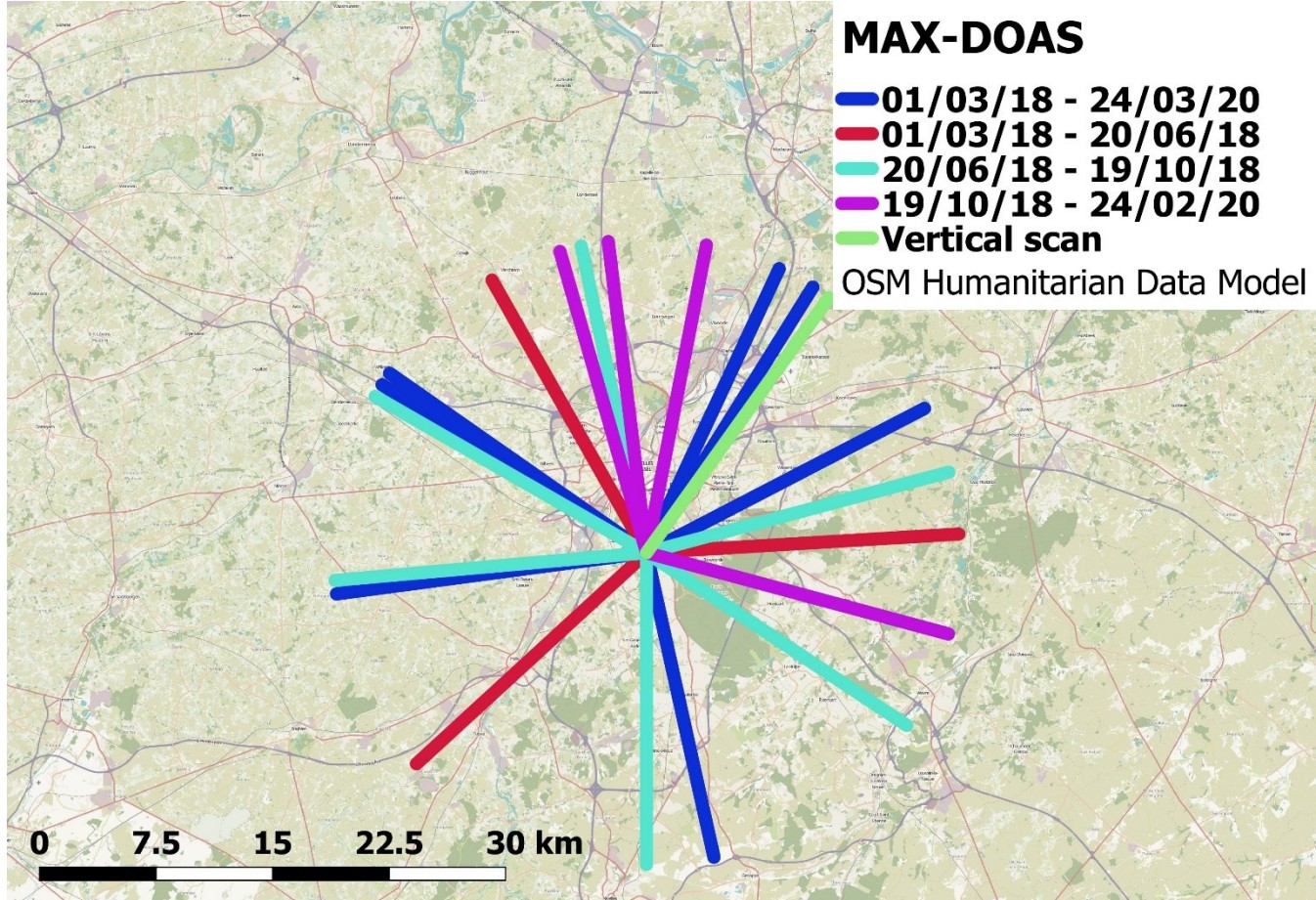

**Figure 1. The experimental set-up of the BIRA-IASB dual-scan MAX-DOAS instrument. Each line is color-coded**
**according to the different set-ups that were used from March 2018 to February 2020. The length of each line is equal**
**to 20 km, which corresponds to the typical horizontal sensitivity for the MAX-DOAS measurements in the present**
**study (see Fig. 18). © OpenStreetMap contributors 2021. Distributed under the Open Data Commons Open Database**
**License (ODbL) v1.0.**

## 2.2 Multi-wavelength DOAS analysis

The measured radiance spectra of a full measurement scan are analyzed using the QDOAS spectral fitting software developed
by BIRA-IASB (Fayt et al., 2011). The DOAS technique separates the narrow absorption features of trace gases in the UV-
Visible spectral range from a spectral background caused mainly by Mie and Rayleigh scattering and instrumental effects. The





trace gas concentration integrated along the light-path in a measured spectrum relative to the amount of the same absorber in a reference spectrum is the primary product of the DOAS analysis and is called differential slant column density (dSCD). Here,

average zenith spectra before and after each measurement scan are used as a reference.

The $O_4$ and $NO_2$ dSCDs are retrieved in six different wavelength intervals: Three intervals in the UV spectral range (330-361 nm, 350-370 nm, and 360-383.5 nm) and three in the Visible range (420-460 nm, 450-490 nm, and 510-540.1 nm). These fitting windows were selected to optimize the determination of the $O_4$ and $NO_2$ dSCDs at the maximum number of different $O_4$ absorption bands available in the wavelength domain of the instrument. Figure 2 shows an example of the $O_4$ and $NO_2$ fits

in all the intervals used in the present work. In each chosen fitting window, we select a reference wavelength, which corresponds to the maximum of an $O_4$ absorption peak (or close to it) in the respective wavelength intervals (see Fig. 2), and it is subsequently used for radiative transport calculations and further analysis. The different reference wavelengths are 343 nm, 360 nm, 380 nm, 447 nm, 477 nm and 530 nm (see Fig. 2). To optimize the derivation of the dSCDs at the six selected wavelengths, the fit of a slope parameter, which accounts for the variation of the dSCD within the fitting interval (Puķīte et

al., 2010), is necessary. This is especially important when the reference wavelength is not located in the center of the fitting window (i.e., 330-361 nm, 350-370 nm, 420-460 nm, and 450-490 nm). The DOAS settings used for each fitting interval are presented in Table S1. As shown in this table, two different $O_4$ cross-sections are used in this study: (1) Finkenzeller (private communication) in the UV fitting intervals and (2) Thalman and Volkamer, (2013) in the Vis fitting intervals. The main motivation for using the $O_4$ cross-section from Finkenzeller (measured at 25°C) in the UV fitting intervals is the significant

improvement of the fit quality and the reduction of the uncertainties for the UV retrievals. Sensitivity tests and comparisons with radiative transport simulations also show that the resulting $O_4$ and $NO_2$ dSCDs are consistent throughout the whole wavelength range covered by the six intervals. For $NO_2$ and $O_3$, which are the strongest absorbers in all the fitting windows, a correction for the solar $I_0$ effect (Aliwell et al., 2002) is applied. A high-resolution solar atlas (Kurucz et al., 1984) is used for the wavelength calibration of the measured spectra.






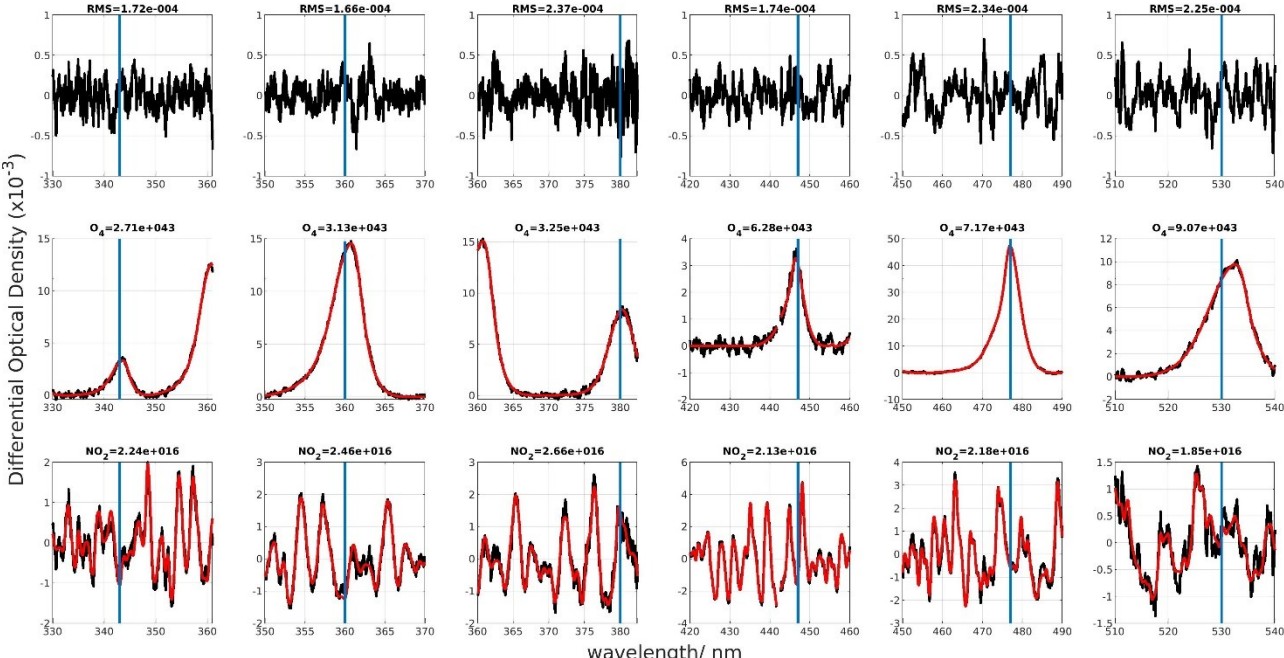

**Figure 2. Fit results of the O₄ and NO₂ fit at the six selected fitting windows from the dual-scan MAX-DOAS measurements in Uccle (2 June 2019 at 07:05 UTC). The measured spectra are represented with black lines, while the fit results are shown with red lines. The blue lines represent the six reference wavelengths.**

## 3 TROPOMI tropospheric NO₂ measurements

In the present study, MAX-DOAS tropospheric $NO_2$ VCDs are used to validate collocated TROPOMI satellite observations.
TROPOMI is a passive grating imaging spectrometer flying onboard the S5P satellite platform. It covers the UV-Visible (250-500 nm), near-infrared (710-770 nm), and short-wave infrared (2314-2382 nm) spectral ranges (Veefkind et al., 2011). TROPOMI measures in a push-broom configuration with a full swath width as wide as 2600 km, and it provides daily global coverage at a spatial resolution (true-nadir pixel size) of 7x3.5 km$^2$, further improved to 5.5x3.5 km$^2$ on 6 August 2019. The TROPOMI tropospheric $NO_2$ algorithm has been developed at KNMI and uses a retrieval-assimilation-modeling system that
is based on the 3-D global TM5 chemistry transport model (van Geffen et al., 2019; Williams et al., 2017).

We use the reprocessed (RPRO) and offline (OFFL) datasets of the TROPOMI L2 tropospheric $NO_2$ column product (see Table 1 for the corresponding versions). According to the guidelines provided by van Geffen et al. (2019), RPRO dataset are available only for the first period of the present study (see Table 1). For the remaining periods, OFFL datasets are used, which are the main data products being available within two weeks from the TROPOMI measurement. To ensure best measurements'
quality, only pixels with a quality assurance value larger than 0.75 are used. This quality flagging eliminates pixels with a cloud radiance fraction larger than 0.5, snow or ice, and erroneous retrievals.





Next to operational products, two additional TROPOMI data sets are also used (see Section 5). In the first one, the TROPOMI retrieval is performed with different a priori profiles (Douros et al., in preparation). The coarse TM5-MP a priori $NO_2$ profiles, using a spatial resolution of 1° x 1°, is replaced by $NO_2$ profile shapes from the CAMS (Copernicus Atmospheric Monitoring

Service) regional Chemistry Transport Model (CTM) ensemble at a spatial resolution of 0.1° x 0.1°. The replacement of a coarse a priori information by a finer one can lead to significant changes in the TROPOMI retrieved $NO_2$ tropospheric columns. The available dataset covers October 2018 to March 2020 (OFFL dataset, L2, and version 01.03.01, 01.03.02, and 02.00.00). In the second additional product, the TROPOMI retrieval is performed with an improved cloud product (Eskes et al., 2021; van Geffen et al., 2021). According to van Geffen et al. (2021), the improvement in the FRESCO-S cloud pressure retrieval

scheme to the FRESCO-wide product, has an impact on the $NO_2$ AMFs and consequently, on the $NO_2$ tropospheric columns over polluted areas. More precisely, the existing FRESCO-S product had a negative bias in the cloud top pressure values, which resulted in a low $NO_2$ tropospheric column (Compernolle et al., 2020). The TROPOMI tropospheric $NO_2$ columns are retrieved using an improved FRESCO-S cloud retrieval scheme, called FRESCO-wide, in v1.4 since 29 November 2020. In the present study, the diagnostic data sets (DDS) are used, which are an ensemble of reprocessed data for past periods analyzed

with new versions (van Geffen et al., in preparation). Over the MAX-DOAS measurement time-period, only DDS2 data corresponding to OFFL datasets (v1.2 and v1.3) are available. Excluding the spin-up period needed by TM5-MP, only four data periods are available for our comparisons (i.e., 30/06/2018 - 06/07/2018, 30/12/2018 - 5/01/2019, 30/03/2019-05/04/2019, and 17/09/2019-23/09/2019).

**Table 1. TROPOMI $NO_2$ processor versions used in the present study.**

| Dataset | Version | Starting date | End date |
|---------|---------|---------------|----------|
| **RPRO** | 01.02.02 | 30/04/2018 | 17/10/2018 |
| **OFFL** | 01.02.00 | 17/10/2018 | 28/11/2018 |
| **OFFL** | 01.02.02 | 28/11/2018 | 20/03/2019 |
| **OFFL** | 01.03.00 | 20/03/2019 | 30/04/2019 |
| **OFFL** | 01.03.01 | 23/04/2019 | 26/06/2019 |
| **OFFL** | 01.03.02 | 26/06/2019 | 24/02/2020 |

## 4 Description of the dual-scan multi-wavelength MAX-DOAS inversion approach

First, the measured radiance spectra in the UV and Visible wavelength ranges is analyzed in six different fitting windows with the main output being the $O_4$ and $NO_2$ dSCDs at six wavelengths, which are 343 nm, 360 nm, 380 nm, 447 nm, 477 nm and

530 nm (see Section 2.2). Then, the OEM-based MMF algorithm is applied to the $O_4$ and $NO_2$ dSCDs in the main azimuthal



direction (and at 477 nm) to retrieve vertical $NO_2$ profiles and obtain information about the vertical extent of $NO_2$ in the troposphere ($MLH_{NO2}$; see Section 4.1).

As an intermediate step, radiative transfer model (RTM) simulations are performed (see Table 2 and Section 4.2) to obtain information about the horizontal sensitivity ($L_{NO2}$) and AOD as a function of $O_4$ dSCDs, wavelength, and $MLH_{NO2}$.

Then, in the next step, a new dual-scan parameterization technique is applied to the $O_4$ and $NO_2$ dSCDs at the six different wavelengths and in all the azimuthal directions with $MLH_{NO2}$, measured $O_4$ dSCDs, and measurement geometry being the main input parameters to retrieve the horizontal sensitivity of $NO_2$ and, consequently, the $NO_2$ near-surface concentrations and VCDs, and near-surface aerosol extinction (see Section 4.2).

In the final step, a new OEM-based horizontal distribution inversion approach is developed using the six near-surface $NO_2$

concentrations and aerosol extinction values per azimuthal direction to retrieve horizontal $NO_2$ and aerosol extinction horizontal profiles in an output horizontal grid of 500m thickness (see Section 4.3).

A flow chart describing the dual-scan multi-wavelength MAX-DOAS inversion approach is shown in Fig. 3.

**Initial input**

measured spectra: UV and VIS channel
• elevation scan: 1 TAA, 9 EA
• horizontal scan: 1 EA, TAA

QDOAS analysis in 6 wavelength windows (see Sect. 2.2)

$O_4$ & $NO_2$ DSCD in 1 TAA and 9 EA at 1 wavelength

$O_4$ & $NO_2$ DSCD in many TAA and 1 EA at 6 wavelengths

vertical profile retrieval with MMF, Sect. 4.1

1 vertical aerosol and $NO_2$ profile

assuming box profile: VCD = MLH x c

measurement geometry

DSCDs

suit of RT simulations with VLIDORT (Table 2)

fit of $L_{NO2}$ as function of geometry, $O_4$ DSCD wavelength, MLH

inversion of fit, AOD as function of geometry, $O_4$ DSCD, wavelength

$L_{NO2}$ function

AOD function

dual scan retrieval similar to Dimitropoulou 2020, adjusted, see Sect. 4.2

1 MLH

1 MLH

adjusted and extended part

mean surface extinctions for 6 different distances $L_{NO2}$ per azimuth

mean surface concentrations for 6 different distances $L_{NO2}$ per azimuth

**Final output**

horizontal surface concentration profiles & extinction profiles in each azimuth direction

new horizontal inversion using 6 surface concentrations/ extinctions per azimuth, see Sect. 4.3

entirely new part



**Figure 3. Dual-scan multi-wavelength MAX-DOAS inversion approach flow chart.**

### 4.1 Aerosol and NO$_2$ OEM-based profile retrievals

The Optimal-estimation-based Mexican MAX-DOAS Fit (MMF) inversion algorithm (Friedrich et al., 2019) is applied to retrieve the aerosol extinction coefficient and NO$_2$ vertical profiles for each MAX-DOAS elevation scan in the main azimuthal

direction at 360 nm and 477 nm. First, the O$_4$ measurements are used to retrieve the aerosol extinction profile. Several studies indicated the importance of applying a scaling factor (<1) to the observed O$_4$ dSCDs to bring them in agreement with simulated O$_4$ dSCDs by radiative transfer modeling (see Wagner et al., 2019 / Table 1 for a comprehensive list of all those studies). However, there is no consensus on the fundamental reason for applying this scaling (see e.g. Ortega et al., 2016). As found by Tirpitz et al. (2021), the choice of the scaling factor has only a small effect on the performance of the trace gas retrieval, so we

decided not to apply it in the present study. The aerosol extinction profile retrieved from each scan is used as an input to the radiative transfer calculations used to retrieve the NO$_2$ retrieval profile. Further details about the MMF inversion algorithm, the input a priori parameters, the quality check of each scan, and the estimated uncertainties of the aerosol and NO$_2$ vertical profile can be found in Dimitropoulou et al. (2020).

A broken cloud-filtering approach based on Gielen et al. (2014) is applied to the MAX-DOAS measurements to exclude MAX-DOAS aerosol and NO$_2$ scans influenced by the presence of clouds, which are known to potentially degrade the quality of the retrievals (Gielen et al. 2014, Wagner et al. 2014). Three sky conditions can be distinguished with this flagging approach: (1) clear sky, (2) homogeneous cloud coverage and (3) broken clouds conditions. Retrievals under broken cloud conditions are rejected from the present study.

The profile retrieval was performed to estimate the Mixing Layer Height of NO$_2$ (MLH$_{NO2}$). The MLH$_{NO2}$ is estimated per measurement scan, and it is the ratio of VCD$_{NO2}$ to the NO$_2$ near-surface concentration as retrieved in the main azimuthal direction by the MMF inversion algorithm. Therefore, during one measurement scan, two assumptions were made: (1) the homogeneous distribution of NO$_2$ inside the MLH$_{NO2}$ and (2) the homogeneous MLH$_{NO2}$ around the measurement site and its use in all the azimuthal directions. The validity of the second assumption is tested in Sect. 4.2.2.


### 4.2 Dual-scan MAX-DOAS retrieval method

A complete MAX-DOAS measurement scan is composed of two different sub-scans, as described in Sect. 2.1. The aerosol and NO$_2$ vertical profiles are retrieved from the elevation scan in the main azimuthal direction. In the other azimuthal directions, measurements are performed only in a single low elevation angle (2°), and therefore, the retrieval of aerosol and

NO$_2$ vertical profiles is not possible. Using the fact that the lowest elevation angles have the highest sensitivity to trace gases located nearby the surface due to the long light path in this layer, a new dual-scan MAX-DOAS retrieval strategy was



developed here. This new retrieval strategy is an extension of the work presented in Dimitropoulou et al. (2020) and aims to retrieve the near-surface $NO_2$ box-averaged volume mixing ratios (VMRs) and the $NO_2$ VCDs at six different wavelengths. In Dimitropoulou et al. (2020), the applied dual-scan $NO_2$ MAX-DOAS retrieval was itself an adaptation of the parameterization

technique proposed by Sinreich et al. (2013). More precisely, in the presence of sufficient aerosols in the atmosphere (i.e., sufficient aerosols to constrain the light path in a near-surface layer and ensure that the near-surface $NO_2$ concentration can be approximated by a near-surface box profile), the measured $NO_2$ dSCDs at one low elevation angle (2°) can be related to the near-surface $NO_2$ box-averaged concentration as follows:

$$dSCD_{NO_2} = c_{NO_2} L_{NO_2} \tag{1}$$

where $dSCD_{NO2}$ is the differential slant column density of $NO_2$ and $c_{NO2}$ its mean concentration along the differential effective light path, $L_{NO2}$.

Consequently, the knowledge of the differential effective light-path's length (i.e., $L_{NO2}$) is crucial to derive the near-surface

$NO_2$ concentrations. The oxygen collisional complex ($O_4$) can be used as a tracer for the effective light-path in the atmosphere: as its concentration is well-known (it is the square of $O_2$ concentration). As a result, observed changes of the $O_4$ dSCDs can be directly attributed to changes in the light-path due to the presence of particles like aerosols and clouds. $L_{O4}$ is calculated as follows:

$$L_{O_4} = \frac{dSCD_{O_4}}{c_{O_4}} \tag{2}$$

where $c_{O4}$ is the typical $O_4$ concentration at the altitude of the instrument.

However, the direct use of the $O_4$ light-path length in the $NO_2$ retrieval is not possible under moderate to high pollution conditions, such as those in Brussels, because the profile shapes of $O_4$ and $NO_2$ are not the same. In Dimitropoulou et al. (2020),

we used radiative transfer model (RTM) simulations to estimate a unitless correction factor, which accounts for these profile shape differences. This unitless correction factor indicates that under moderate to high pollution conditions, $L_{NO2}$ is equal to or smaller than $L_{O4}$. For a correction factor equal or close to one, $L_{O4}$ is equal to $L_{NO2,}$ which means that there is a moderate to high aerosol load in the atmosphere during the measurement. On the other hand, correction factors smaller than unity are obtained for measurements performed under aerosol-free conditions or a thin MLH. Assuming a homogeneous $NO_2$

distribution inside the MLH, the MLH is derived from the $NO_2$ vertical profiles in the main azimuthal direction and is defined as the ratio of the $NO_2$ VCD to the near-surface concentration of $NO_2$. For more information, we refer the reader to Dimitropoulou et al. (2020). The RTM simulations are performed for eight different MLH values of aerosols and $NO_2$ in the range of 500-2000 m (i.e. eight different combinations) and for different measurement viewing geometries (Solar Zenith Angle



(SZA), Relative Azimuth Angle (RAA) and the corresponding elevation angle of 2°). For every MAX-DOAS measurement,
one value of the correction factor is given according to its viewing geometry and MLH value during the measurement.

In the present study, a new dual-scan $NO_2$ MAX-DOAS retrieval method, which is more suitable for interpreting multi-wavelength measurements than the previous approach (Dimitropoulou et al., 2020), is developed. It is presented in detail in the following subsection.

### 4.2.1 Developed dual-scan MAX-DOAS retrieval method

The main advantages of the new dual-scan $NO_2$ MAX-DOAS retrieval method (which are also the main differences with respect to Dimitropoulou et al., 2020) are the following: (1) the direct use of the measured $O_4$ dSCDs to estimate $L_{NO_2}$ for every measurement, (2) retrieval of near-surface aerosol extinction close to the ground, and (3) the exploitation of the wavelength dependency of the horizontal path representative of MAX-DOAS measurements for the retrieval of the horizontal distribution of aerosols (and therefore $NO_2$) around the measurement site. The latter is done using $O_4$ and $NO_2$ dSCDs measured
at six different wavelengths. This new method is described below.

Assuming that the $NO_2$ vertical distribution can be approximated by a box profile of height equal to mixing layer height ($MLH_{NO_2}$), the following equation can be used:

$$c_{NO_2} = \frac{VCD_{NO_2}}{MLH_{NO_2}} = \frac{dSCD_{NO_2}}{L_{NO_2}} \qquad (3)$$


This means that the $NO_2$ near-surface concentration can be expressed as a ratio of the $dSCD_{NO_2}$ to the $L_{NO_2}$ (see Eq. 1) or as a ratio of the $VCD_{NO_2}$ to the $MLH_{NO_2}$. Using this equation, $L_{NO_2}$ can be estimated as follows:

$$L_{NO_2} = dSCD_{NO_2\,simulated} \cdot \frac{MLH_{NO_2}}{VCD_{NO_2}} \qquad (4)$$


Here, $O_4$ dSCDs and $L_{NO_2}$ are simulated using the radiative transfer model VLIDORT version 2.7 (Spurr, 2006). Seasonal median MAX-DOAS $NO_2$ vertical profiles, as retrieved by applying the MMF inversion algorithm in the main azimuthal direction (see Sect. 4.1), show that the bulk (70 %) of the $NO_2$ concentration is located inside the $MLH_{NO_2}$, which is expected since $MLH_{NO_2}$ is estimated as the ratio of $VCD_{NO_2}$ to the near-surface $NO_2$ concentration. On the other hand, this is not the
case for aerosols (only 30 % of the aerosol content is seen to be located inside the $MLH_{NO_2}$). Considering this feature, for the VLIDORT simulations, the $NO_2$ a priori profiles are modeled as box profiles with a constant concentration equal to $1.5 \times 10^{11}$ molec/cm$^3$ from the surface to the $MLH_{NO_2}$. Two layers compose the aerosol a priori profiles: (1) the $MLH_{NO_2}$ and (2) the free troposphere. The equation, which is applied to estimate the aerosol extinction profile a(z), is the following (see Wang et al., 2014):





$a(z) = \text{AOD} \frac{p}{\text{MLH}_{NO_2}}, \text{for } z \leq \text{MLH}_{NO_2}$ (5a)

and,

$a(z) = b(\xi, \text{MLH}_{NO2}, p) \exp(-\frac{z}{\xi}), \text{for } z > \text{MLH}_{NO_2}$ (5b)

where AOD is the aerosol optical depth, p is the fraction of AOD inside the $\text{MLH}_{NO2}$, b is a normalizing constant for the exponential component (see Eq. 5 from Wang et al., 2014), z is the simulation altitude grid, and $\xi$ is the scaling height for the

aerosols located outside the $\text{MLH}_{NO2}$, which is set to 5 km (Wang et al., 2014). In the present study, the fraction of AOD located within the $\text{MLH}_{NO2}$ is set to p=0.3 (see above). The effect of the p value and the $NO_2$ profile shape on the retrieved $NO_2$ near-surface VMRs and VCDs were investigated and considered in the error budget (see Sect. 4.2.2).

The $\text{MLH}_{NO2}$ is estimated per measurement scan, as the ratio of $\text{VCD}_{NO2}$ to the $NO_2$ near-surface concentration as retrieved in the main azimuthal direction by the MMF inversion algorithm.

The RTM simulations have in total nine input parameters, which are the elevation angle, SZA, RAA, AOD, $\text{MLH}_{NO2}$, $c_{NO2}$, AOD (p and $\xi$), and wavelength. It should be noted that the elevation angle is kept constant (i.e., 2°). For the six different wavelengths (343 nm, 360 nm, 380 nm, 447 nm, 477 nm, and 530 nm), we separately perform RTM simulations and $L_{NO2}$ (see Eq. 4) are calculated for the assumed SZA, RAA, $\text{MLH}_{NO2}$, $c_{NO2}$, and AOD input scenarios presented in Table 2.

**Table 2. RTM inputs for the simulations of $L_{NO2}$ at the six selected wavelengths (343 nm, 360 nm, 380 nm, 447 nm, 477 nm, and 530 nm).**

| Parameter | Values |
|---|---|
| **Wavelength/ nm** | 343, 360, 380, 447, 477, 530 |
| **SZA/ °** | 20, 30, 40, 50, 60, 70, 80 |
| **RAA/ °** | 0, 10, 20, 30, 40, 50, 60, 90, 120, 150, 180 |
| **AOD** | 0, 0.1, 0.3, 0.4, 0.6, 0.8, 1 |
| **p of AOD** | 0.30 |
| **ξ of AOD/ km** | 5 |
| **asymmetry parameter** | 0.68 |
| **Single Scattering Albedo (SSA)** | 0.92 |
| **MLH/ m** | 500, 1000, 1500 |
| **Elevation angle/ °** | 2 |
| **$C_{NO2}$/ molec.cm$^{-3}$** | $1.5 \times 10^{11}$ |



The $O_4$ dSCDs are a function of the input parameter AOD. The relation between the simulated $O_4$ dSCDs and the input AOD values is shown in Fig. 4. A Piecewise cubic hermite interpolating polynomial fitting through the AOD as a function of the simulated $O_4$ dSCDs for each SZA, RAA, and $MLH_{NO2}$ combination can be used in order to perform an inverse method (i.e. to estimate the near-surface aerosol extinction from the measured $O_4$ dSCDs).

For every combination of all eight parameters (i.e., all the parameters of Table 2, except the AOD values), a polynomial fit of $L_{NO2}$ as a function of simulated $O_4$ dSCDs is applied. Fig. 5 shows simulated $L_{NO2}$ as a function of simulated $O_4$ dSCDs, and a second-order polynomial is fitted through the data points. Since $NO_2$ is an optically thin absorber, $L_{NO2}$ is not a function of $c_{NO2}$ and consequently, a $L_{NO2}$ value can be estimated for each measurement. Based on the corresponding SZA, RAA, measured $O_4$ dSCD, and $MLH_{NO2}$, a $L_{NO2}$ is attributed to each low elevation MAX-DOAS measurement through this polynomial fit. To express $L_{NO2}$ as a function of four different parameters (i.e., $O_4$ dSCD, SZA, RAA, and $MLH_{NO2}$), $L_{NO2}$ is interpolated linearly at the $O_4$ dSCD, SZA, RAA, and $MLH_{NO2}$ of each measurement. For example, a MAX-DOAS measurement with SZA=30$^o$, RAA=60$^o$, $MLH_{NO2}$=1 km, and measured $O_4$ dSCD=$6.10^{43}$ molec$^2$.cm$^{-5}$ will have a $L_{NO2}$ equal to 15 km at 477 nm (see Fig. 6). Based on this approach, the near-surface $NO_2$ concentration can be calculated at the six different wavelengths by using Eq. (1) and the derived $L_{NO2}$ values. The corresponding near-surface $NO_2$ VMR are obtained by dividing the $NO_2$ concentrations by the air number density. To derive the air number density, we use monthly averaged pressure and temperature profiles over a 20-year period. These profiles are extracted from the European Centre for Medium-Range Weather Forecasts (ECMWF) ERA-Interim reanalysis. In the last step, the tropospheric $NO_2$ VCD is calculated from the product of the near-surface $NO_2$ concentration with the $MLH_{NO2}$.

Regarding the aerosols, the AOD is estimated for every off-axis measurement (see Fig. 4). The near-surface aerosol extinction is then calculated as the ratio between the aerosols inside the $MLH_{NO2}$ (i.e., AOD times p) and $MLH_{NO2}$. The near-surface aerosol extinction refers to the layer that extends from the surface to the $MLH_{NO2}$. As discussed above, around 30% of the total aerosols is expected to be found inside this layer.

The effect of SZA, RAA, and $MLH_{NO2}$ on the simulated $L_{NO2}$ is investigated in the supplement. First, the simulated $L_{NO2}$ are presented in Fig. S1 as a function of RAA for different $MLH_{NO2}$ and wavelengths and a single AOD and SZA value. $L_{NO2}$ strongly depends on $MLH_{NO2}$. The lower the $MLH_{NO2}$, the shorter the $L_{NO2}$ is. The same $NO_2$ concentration and aerosol load are used for the three different $MLH_{NO2}$ scenarios. So, when aerosols are concentrated in a thin layer (i.e., $MLH_{NO2}$=0.5 km), $L_{NO2}$ becomes shorter. Secondly, we observe that $L_{NO2}$ depends on RAA. The larger the RAA, the longer the $L_{NO2}$. In Fig. S2, simulated $L_{NO2}$ are plotted for each wavelength and each considered $MLH_{NO2}$ as a function of SZA (at a constant AOD and RAA). $L_{NO2}$ depends strongly on SZA. The highest dependency is observed for large SZA values, where $L_{NO2}$ becomes maximum. Finally, in both Fig. S1 and S2, we observe that $L_{NO2}$ becomes longer with wavelength, which is expected because of the less pronounced Rayleigh scattering at longer wavelengths.





An example of dual-scan MAX-DOAS retrieval is shown in Fig. 6. Based on the RTM simulations described above, $L_{NO2}$ is derived for the wavelengths of interest, and ultimately, near-surface $NO_2$ concentrations and tropospheric $NO_2$ VCDs are estimated. In the last step, the near-surface aerosol extinction values are assigned to the six different wavelengths.


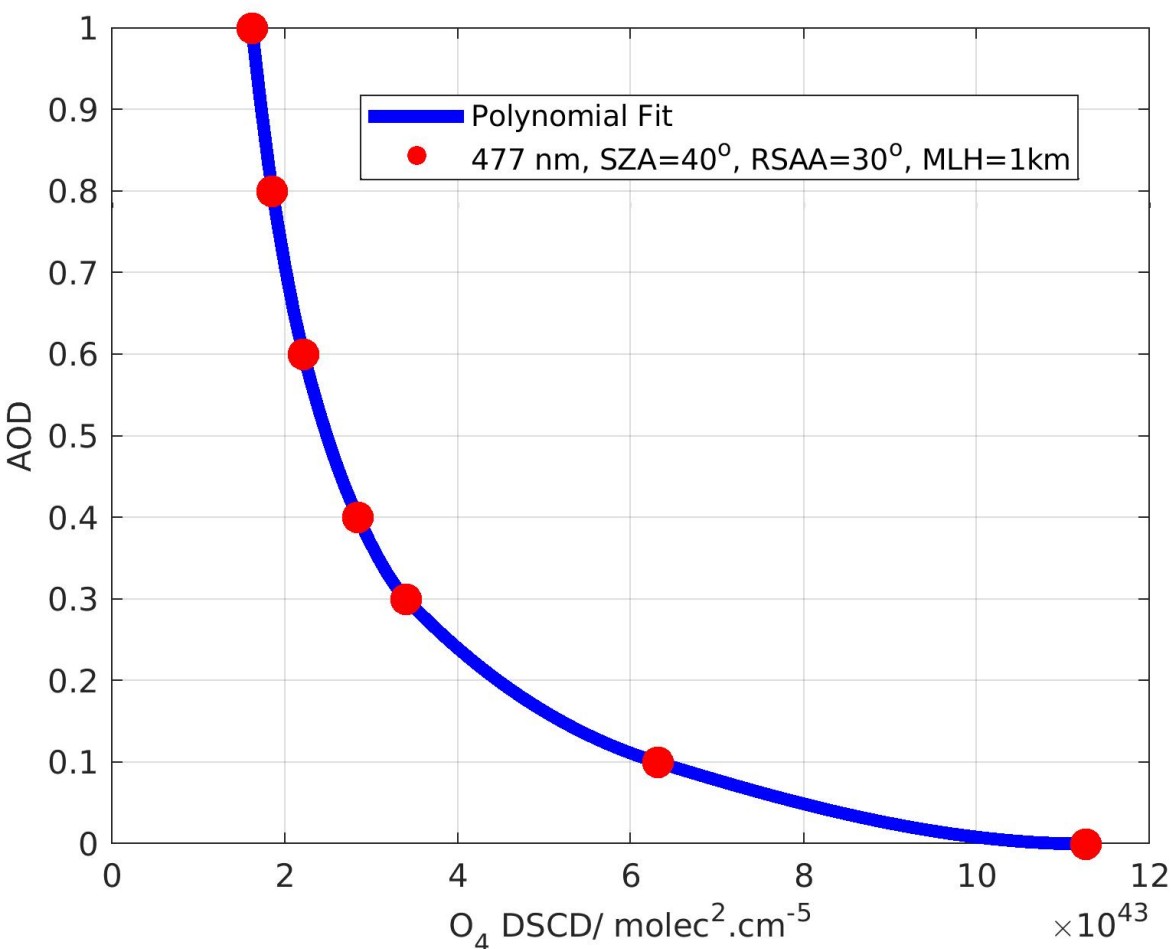

**Figure 4. Dots: Simulated AOD for NO₂ box profile of 1 km at 477nm for a SZA of 40° and RAA of 30° as a function**
**of the simulated O₄ DSCDs for the different AOD values (1, 0.8, 0.6, 0.4, 0.3, 0.1 and 0; see Table 2). Blue line: simulated AOD by applying an exponential fit through the data points.**



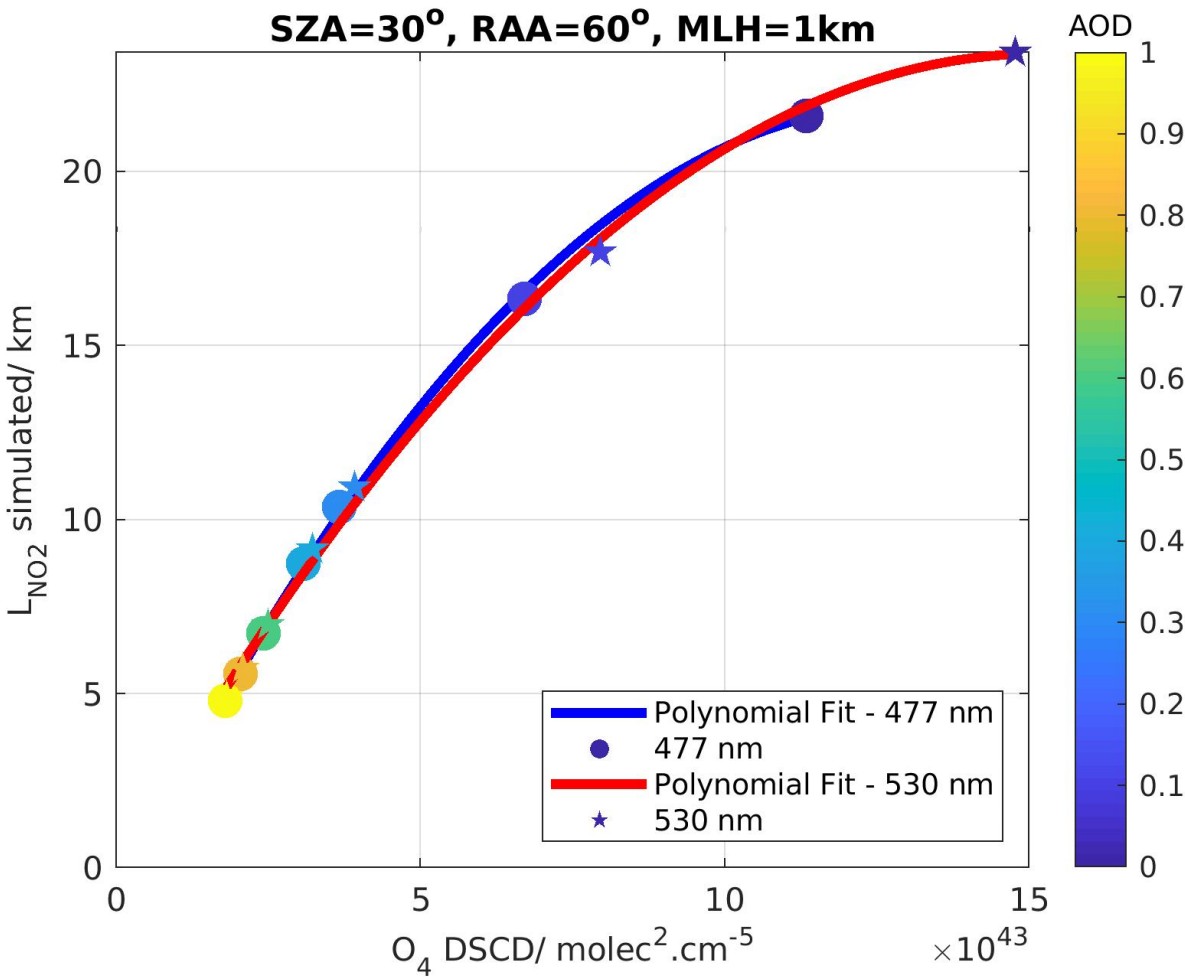

**Figure 5. Dots (stars): Simulated $L_{NO_2}$ for $NO_2$ box profile of 1 km at 477nm (530 nm) for a SZA of 30° and RAA of 60° as a function of the simulated $O_4$ DSCDs for the different AOD values (1, 0.8, 0.6, 0.4, 0.3, 0.1 and 0; see Table 2). Blue (red) line: 2nd-order polynomial fit through the data points.**



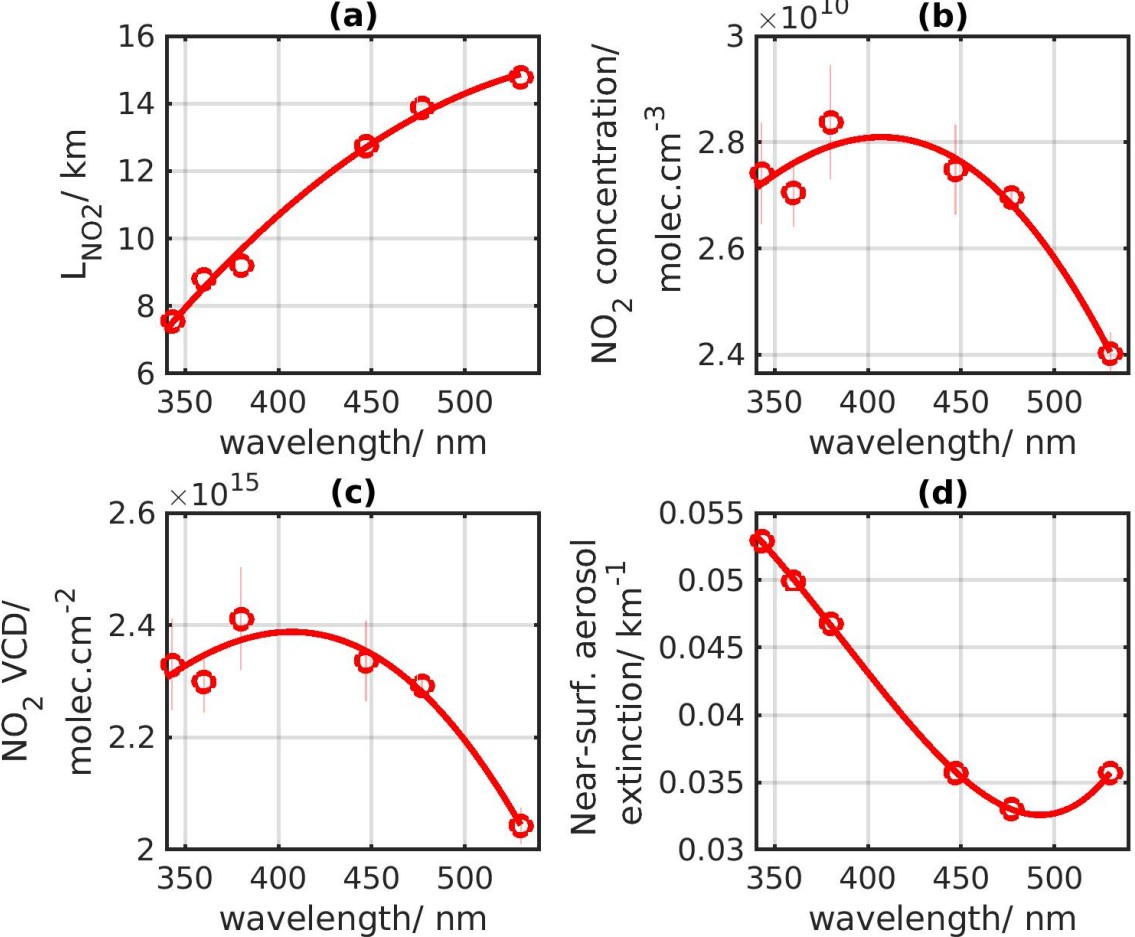

**Figure 6. (a) Corresponding L$_{NO2}$, (b) near-surface NO$_2$ concentrations, (c) NO$_2$ VCDs, and (d) aerosol optical densities as a function of the six wavelengths used in the retrieval (11 September 2018, 11:51 UTC, 123.5° azimuthal direction).**


### 4.2.2 Uncertainty budget

To estimate uncertainties on the dual-scan parameterized NO$_2$ near-surface concentration and VCD, the standard error propagation method is used as:

$$\sigma^2_{c_{NO2}} = \left(\sigma_{dSCD_{NO_2}} \frac{\partial c_{NO2}}{\partial dSCD_{NO_2}}\right)^2 + \left(\sigma_{L_{NO2}} \frac{\partial c_{NO2}}{\partial L_{NO2}}\right)^2 \tag{6}$$

which is solved as:



$$\sigma^2_{c_{NO2}} = \left(\sigma_{dSCD_{N\,O_2}} \frac{c_{NO2}}{dSCD_{NO_2}}\right)^2 + \left(\sigma_{L_{NO2}} \frac{c_{NO2}}{L_{NO_2}}\right)^2 \tag{7}$$

According to Kreher et al. (2019) and Bösch et al. (2018), in urban or suburban polluted conditions, the use of the DOAS fit uncertainty of $NO_2$ for the $dSCD_{NO2}$ uncertainty is not appropriate, because the $dSCD_{NO2}$ uncertainty is mostly driven by atmospheric variability as well as spatial and temporal fluctuations in the $O_4$ and $NO_2$ fields. In this study, a conservative value of $3.5 \times 10^{15}$ molec.cm$^{-2}$ is attributed to $\sigma_{dSCD_{N\,O_2}}$ (Kreher et al., 2019). This represents an error of up to 5% on the $NO_2$ dSCDs in the visible range (477 nm).

The second error source is related to the estimation of $L_{NO2}$ from the RTM simulations. To estimate this error, sensitivity tests on the input aerosol and $NO_2$ vertical profiles were performed. The fraction of aerosols located inside the $MLH_{NO2}$ (40% and 60% instead of 30%) and the $NO_2$ profile shape (linearly decreasing instead of box) were modified. The error related to the RTM simulations is about 11% in the Visible range (477 nm).

Combining all the error sources, the total uncertainties on the $NO_2$ near-surface concentration is about 16%, 15%, 14%, 11%, 10%, and 10% in 343 nm, 360 nm, 380 nm, 447 nm, 477 nm, and 530 nm, respectively.

The $NO_2$ VCD is the product of the $NO_2$ near-surface concentration and $MLH_{NO2}$. According to Dimitropoulou et al. (2020), the uncertainty related to $MLH_{NO2}$ is about 4%. It is found that the relative difference between $MLH_{NO2}$ values derived in the main azimuthal directions and in other three additional directions depends strongly from the direction (see Section 4.2.3). In a refined version, this direction dependent error source on the $NO_2$ near-surface concentration will be included in the uncertainty budget.

By using this finding, the total uncertainties on the $NO_2$ VCD is about 17%, 16%, 15%, 11%, 11%, and 10% in 343 nm, 360 nm, 380 nm, 447 nm, 477 nm, and 530 nm, respectively.

### 4.2.3 Validation of the dual-scan MAX-DOAS retrieval method

The sanity check and validation of the dual-scan MAX-DOAS retrieval method in Uccle is based on two different correlative comparisons.

The sanity check compares the $NO_2$ near-surface VMRs and tropospheric VCDs retrieved by the dual-scan parameterization in the main azimuthal direction to the same quantities retrieved with the MMF inversion algorithm at the two main wavelengths (360 nm and 477 nm). As can be seen in Figures 7 and 8, both data sets are in good agreement, with correlation coefficient values in the range of 0.86 to 0.95 and slope values close to unity for all the four comparisons.

The validation step is based on the same type of comparison as the first one but for three additional azimuthal directions, where elevation scans, and hence profile retrievals, are available for some periods. Onward July 3, 2019, elevation scans were performed in these three additional azimuthal directions to complement the already existing measurement set-up. These elevation scans were performed once per day, around noon, in the 11º, 105º, and 262.5º azimuthal directions. Figure 9 shows





the comparison between near-surface $NO_2$ VMRs and tropospheric VCDs retrieved by the dual-scan parameterization method

and the corresponding results obtained with the MMF inversion algorithm. Overall good agreement is obtained (R=0.79 and 0.84 for near-surface VMR and VCD, respectively). We observe that the comparison concerning the near-surface $NO_2$ VMR seems to be noisier than in the main azimuth direction. This is mainly due to the use of the $MLH_{NO2}$ calculated in the main azimuthal direction for all the different azimuth angles in the dual-scan method. Additionally, the parameterization technique slightly underestimated the near-surface $NO_2$ VMR (s=0.84) while a slope value of 1.00 is obtained for tropospheric VCDs.



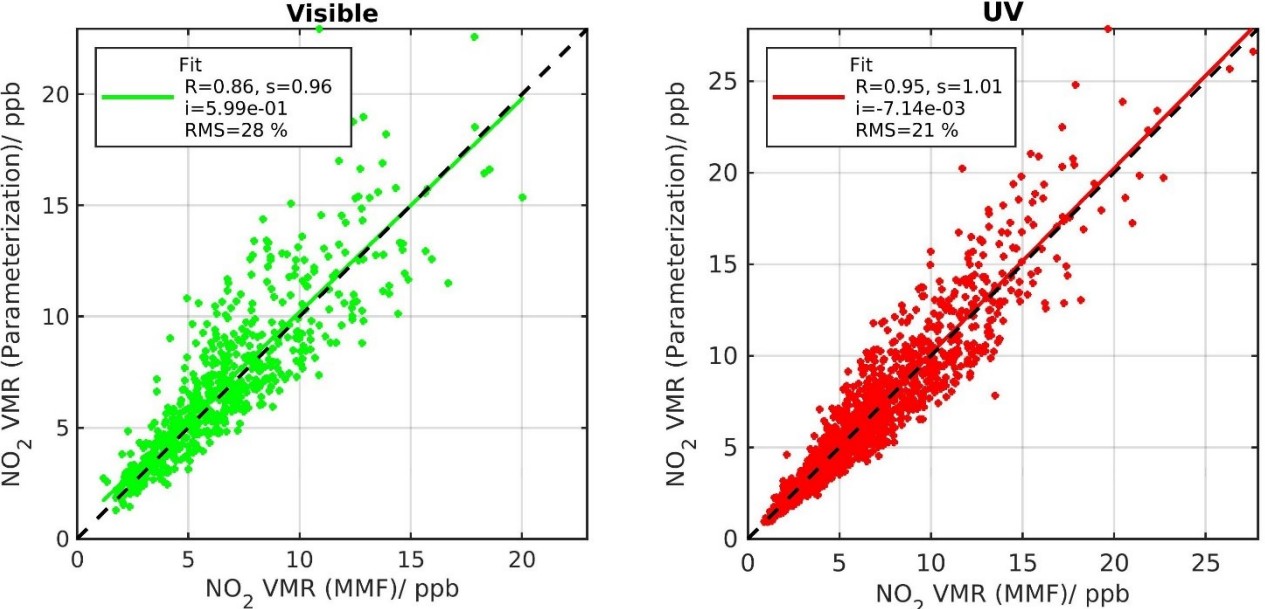

**Figure 7. Comparison between MMF and parameterized $NO_2$ near-surface VMR at 477 nm (Visible, left panel), and 360 nm (UV, right panel), as derived from the main azimuthal direction (i.e., 35.5° azimuthal direction).**






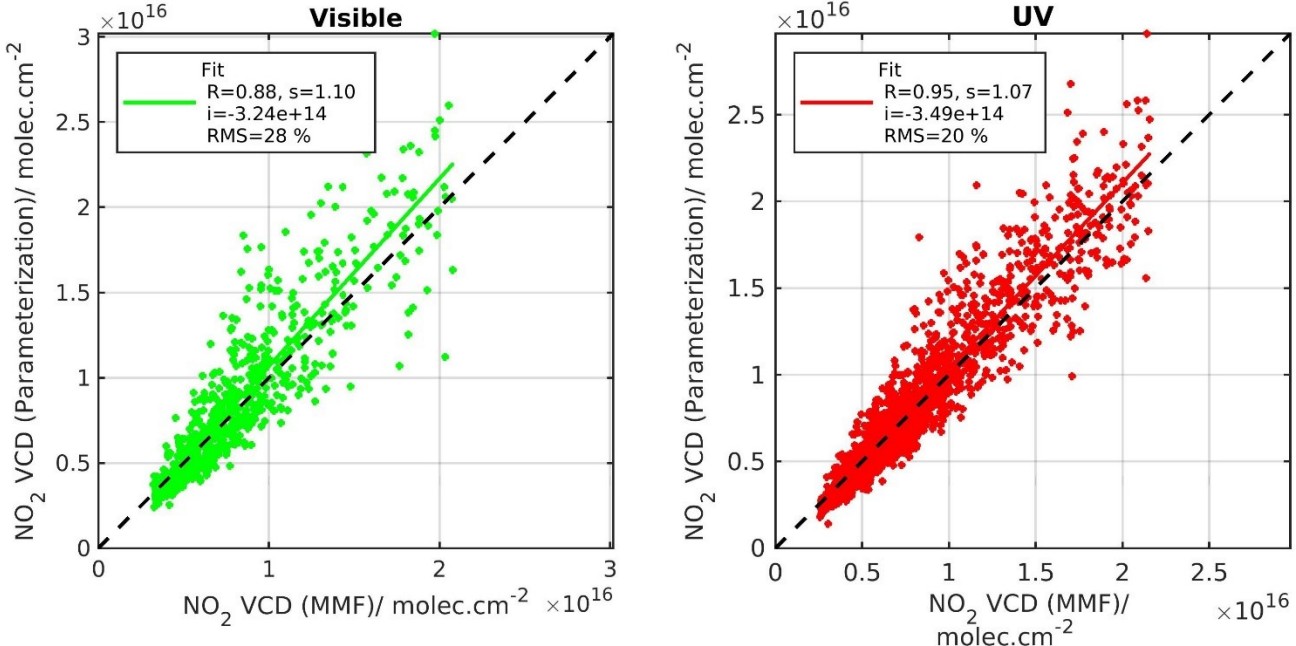

**Figure 8. Comparison between MMF and parameterized NO₂ VCD at 477 nm (Visible, left panel), and 360 nm (UV,**

**right panel), as derived from the main azimuthal direction (i.e., 35.5º azimuthal direction).**






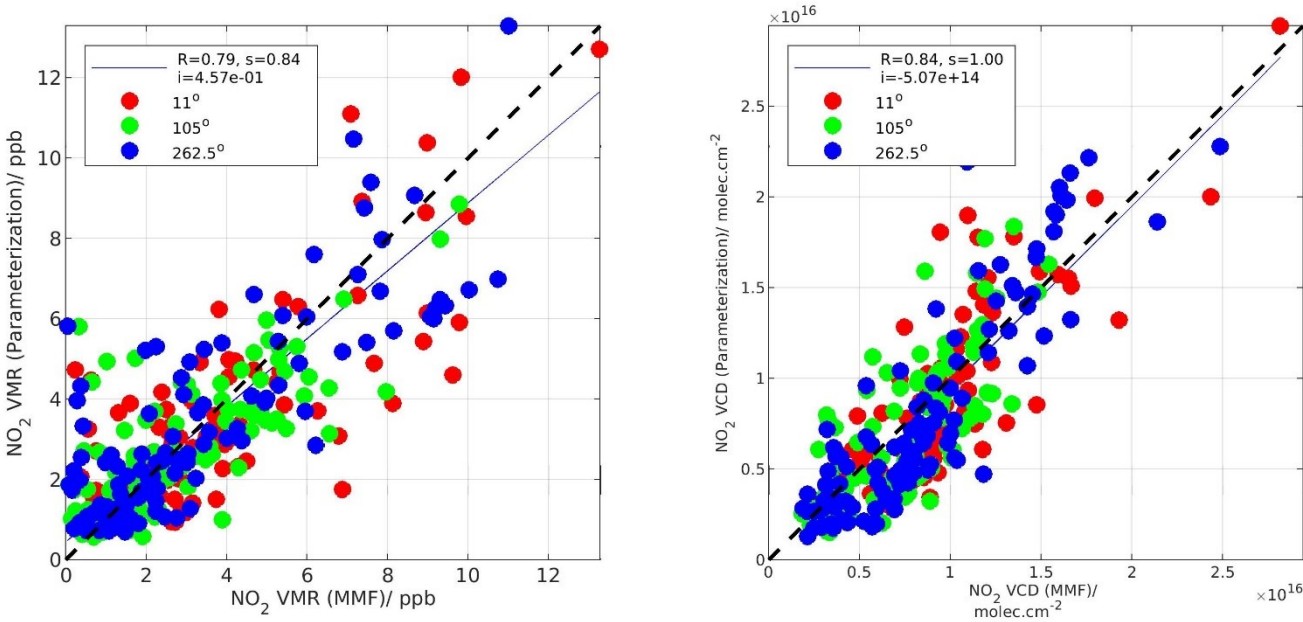

**Figure 9. Visible range: comparison between MMF and (left panel) parameterized NO₂ near-surface VMR and (right panel) parameterized NO₂ VCD at three different azimuthal directions, as indicated in the color bar (11°, 105°, and 262.5⁰ azimuthal directions). The elevation scans in these azimuthal directions were performed once per day from 3 July 2019.**

### 4.3 Horizontal distribution inversion approach

The parameterized NO₂ near-surface concentrations at the six different wavelengths are used as input in a new horizontal distribution inversion approach. As parameterized NO₂ near-surface concentrations, we refer to the conversion of the measured NO₂ dSCDs (i.e. at the elevation angle of 2°) to near-surface NO₂ concentrations by applying the dual-scan MAX-DOAS retrieval method as described in Sect. 4.2. Figure 10 shows a sketch of the assumed horizontal box model configuration, in which successive boxes of concentration $c_N$ between the horizontal distances $x_{N-1}$ and $x_N$ from the MAX-DOAS instrument are considered along the light path. The index N is equal to the total number of successive boxes.

The different horizontal lines illustrate the horizontal extent (or differential effective light path as described in Sect. 4.2) in which the NO₂ near-surface concentrations are extended for the six different wavelengths. Generally, the MAX-DOAS horizontal sensitivities are longer for larger wavelengths because of the less pronounced Rayleigh scattering (see also Fig. 6; Ortega et al., 2016; Dimitropoulou et al., 2020). In Fig. 10, the shortest line represents the smallest wavelength's horizontal

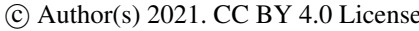



sensitivity (343 nm), and the longest line the largest wavelength's horizontal sensitivity (530 nm). As can be seen in the sketch,
the effective horizontal light path at the six different wavelengths passes through different number of horizontal bins.

The parameterized $NO_2$ near-surface concentrations at the different wavelengths are the mean concentrations along the horizontal effective light paths (see Section 4.2), which are also called differential effective light paths because they are linked to the $dSCD_{NO2}$. When having information coming from one wavelength only, it is not possible to know how the $NO_2$ is distributed along this light path. In the present work, the knowledge of mean $NO_2$ concentrations at six different wavelengths
is used to retrieve a horizontal $NO_2$ profile, assuming the horizontal box model described in Fig. 10. This new retrieval method is described below.

The measurement vector **y** consists of the six retrieved surface concentrations (called as $\bar{c}_{NO2}$; see method presented in Sect. 4.2) at the six different wavelengths. These near-surface concentrations can be expressed as functions of the different effective
light paths ($\mathbf{L_{NO2}}$) and correspond to the average surface concentrations along those $\mathbf{L_{NO2}}$:

$$y = \mathbf{F_{meas}}(c_{NO2\_true}) = \bar{c}_{NO2} = \frac{dSCD_{NO2}}{L_{NO2}} \tag{8}$$

$\mathbf{F_{calcul}}$, which represents the forward model, can be expressed as follows:

$$\mathbf{F_{calcul}}(c_{NO2\_true}) = \frac{1}{L_{NO2}} \int_0^{L_{NO2}} c_{NO2}(x)dx \tag{9}$$

where x is the horizontal distance and $c_{NO2}$ the $NO_2$ near-surface concentration as a function of x, the distance from the MAX-DOAS instrument.

Our retrieval of the horizontal distribution of $\mathbf{c_{NO2}}$ is based on the inversion theory (Rodgers, 2000), in which a horizontal profile $\mathbf{c_{NO2}}$ (state vector) is retrieved given an a-priori horizontal profile $\mathbf{x_a}$, the measurement vector **y**, the matrix of the weighting function **K**, the uncertainty covariance matrix of the a priori $\mathbf{S_a}$ and the uncertainty covariance matrix of the measurement $\mathbf{S_e}$:

$$c_{NO2} = \mathbf{x_a} + (\mathbf{K^T S_e^{-1} K + S_a^{-1}})^{-1} \mathbf{K^T S_e^{-1}}(\mathbf{y} - \mathbf{K x_a}) \tag{10}$$

The weighting function indicates the sensitivity of the measurement vector to a change in the horizontal profile. It is given in the present case by the following analytical functions:





$$K\left(x, L_{NO_2}\right) = \frac{dF_{calcul}}{dc_{NO_2}} = \begin{cases} \frac{dx}{L_{NO_2}} & \text{for } 0 < x < L_{NO_2} \\ \frac{A\,dx}{L_{NO2}} & \text{for } x \text{ (last)} > L_{NO_2} \\ 0 & \text{for } x > L_{NO_2} \end{cases} \tag{11}$$


where A is the coverage percentage of the differential effective light path length at the last horizontal grid.

An example of weighting functions is presented in Fig. 11. As can be seen, each measurement is sensitive from the MAX-DOAS instrument location to the horizontal distance equal to the differential effective light path length of each measurement. As each last horizontal grid is not fully covered by each measurement, the coverage percentage is considered for these grid

cells. It should be noted that since $NO_2$ is an optically thin absorber, the measurements depend linearly on each horizontal box's concentration. For this reason, OEM for the linear case is considered here, and only one inversion step is needed (see Eq. 10).

The selected output horizontal grid for the retrieval extends from the MAX-DOAS instrument to the maximum differential effective light path ($L_{NO2}$ at 530 nm) per azimuthal direction and consists of successive boxes of 0.5 km thickness on the

horizontal axis.

Since this inversion problem is ill-conditioned, more than one horizontal $NO_2$ profile can be consistent with the measurement vector. To reject unrealistic solutions, the a priori profile $\mathbf{x_a}$ and its uncertainty covariance matrix must be included in the retrieval. In the OEM, the a priori information usually comes from an independent source, like a model or other correlative measurements. In the present study, RIO model data were chosen as a priori. RIO is a land-use regression model based on the

interpolation of the hourly $NO_2$ near-surface concentrations measured by the in-situ telemetric air quality network in Belgium (Hooyberghs et al., 2006; Janssen et al., 2008). RIO provides hourly $NO_2$ concentration maps on a 4x4 km² spatial resolution. Seasonal average maps of RIO $NO_2$ near-surface concentration are constructed (see Fig. S3) and after, seasonal averages of RIO $NO_2$ near-surface concentration horizontal profiles were calculated in each azimuthal direction and interpolated on the retrieval's horizontal grid by regridding the initial 4x4 km² spatial resolution to a finer one (see Fig. 12). The shape of the RIO

a priori $NO_2$ profiles per azimuthal direction stays the same during different seasons of the year, indicating that the wind effect on $NO_2$ transportation disappears by the seasonal averaging and that the same sources contribute to the $NO_2$ horizontal field. A mean scaling factor equal to the mean ratio between the measured and RIO $NO_2$ near-surface concentrations is applied because of the systematic underestimation of $NO_2$ near-surface concentrations by MAX-DOAS when compared to in-situ measurements (Dimitropoulou et al., 2020).

For the aerosols horizontal distribution retrieval, there are not sufficient independent measurements that provide information about the horizontal distribution of AOD and can serve as an a priori AOD profile. Therefore, a horizontally constant a priori AOD profile is used in the AOD retrieval based on CIMEL observations. An AOD equal to 0.18, which is the yearly-averaged AOD value from CIMEL at 477 nm, is used. To construct the near-surface aerosol extinction a priori profiles, it is considered





that 30% of the total amount of AOD is located inside the MLH (i.e., known for each MAX-DOAS vertical scan from the
MMF inversion algorithm; see Section 4.1).

The diagonal elements of the $\mathbf{S_a}$ matrix are set equal to the square of a scaling factor times the $NO_2$ concentration a priori profile. The non-diagonal elements, which account for correlation between the different horizontal grid cells, are set as follows (Barret et al., 2002):

$$\mathbf{S_{a_{ij}}} = \sqrt{\mathbf{S_{a_{ii}}}\mathbf{S_{a_{ij}}}} \exp\left(-\ln(2)\left(\frac{x_i - x_j}{\gamma}\right)^2\right) \qquad (12)$$

where $x_i$ and $x_j$ are the horizontal distances at the $i^{th}$, and $j^{th}$ horizontal boxes and $\gamma$ is half of the correlation length. $\gamma$ is set equal to 3.5 km. To eliminate inversion instabilities, $\mathbf{S_a}$ elements which are smaller than 0.1% of the maximum $\mathbf{S_a}$ element are set equal to zero.

To estimate the correlation length, a covariance matrix was constructed by exploiting the airborne observations above Brussels (28 June 2019). The airborne observations have a spatial resolution of approximately 100 x 100 $m^2$. $NO_2$ horizontal profiles were constructed in different azimuthal directions in a spatial resolution of 500 x 500 $m^2$, expanding from the MAX-DOAS's position to a maximum distance of 20 km, and were used to calculate a covariance matrix. A correlation length equal to 7 km, and consequently, a gamma value equal to 3.5 km, is found to be representative for the $NO_2$ horizontal profiles in Brussels.

Using this correlation length, a variance of 45% is used. This choice was conducted based on the seasonal variance of the RIO a priori profiles compared to their seasonal mean value. It is found that the seasonal variance of RIO observations has a mean value of 45%. Additionally, it is found to be a good compromise for obtaining reasonable retrieval results e.g. in terms of information content, and while avoiding unrealistic oscillations in the retrieved aerosol and $NO_2$ profiles.

The measurement covariance matrix $\mathbf{S_e}$ is chosen to be diagonal, with elements corresponding to the uncertainties of the dual-
scan parameterized $NO_2$ near-surface concentration (see Section 4.2.2).

An example of the retrieved $NO_2$ horizontal profile is presented in Fig. 13, together with corresponding measured and simulated $\bar{c}_{NO2}$ at the six different wavelengths for July 2, 2018 (25° azimuthal direction). RMS is calculated between measured and simulated $NO_2$ near-surface concentrations of the horizontal retrieval normalized by the mean of the measured $NO_2$ near-surface concentrations (upper panel in Fig. 13). For distances smaller than the minimum $L_{NO2}$ (around 8 km), the measurements
do not give information about the horizontal distribution of $NO_2$. Consequently, the retrieved $NO_2$ horizontal profile at these ranges is coming from the a priori profile. Similarly, the measured and retrieved near-surface aerosol extinction coefficient and the retrieved aerosol horizontal profile are shown in Fig. 14, for one sample case on 11 September 2018 (167.5° azimuthal direction).

An essential condition of the dual-scan MAX-DOAS retrieval and the new horizontal inversion approach at six different
wavelengths is the increasing trend of the horizontal sensitivity as a function of wavelength. Consequently, every wavelength is sensitive to a different horizontal region and the six different wavelengths can be used to retrieve the horizontal distribution





of aerosols and trace gases. Sensitivity tests were conducted in which simulated $L_{O4}$ are expressed as a function of the six different wavelengths for different aerosol conditions. As can be seen in Fig. S4, the linear relationship between $L_{O4}$ (and $L_{NO2}$) and wavelength exists for AOD values ranging from 0 to 1. An AOD equal to unity is chosen as the maximum AOD of the

simulations because in Uccle, AOD values rarely exceed one (see in https://aeronet.gsfc.nasa.gov/ for the Brussels measurement site). Therefore, the relation stays linear as the aerosol load changes for the conditions observed in Uccle. The only condition leading to non-linearity is when clouds are present. However, as explained in Sect. 4.1, a cloud filtering approach is applied, rejecting the broken cloud scenes, which are the more problematic ones.


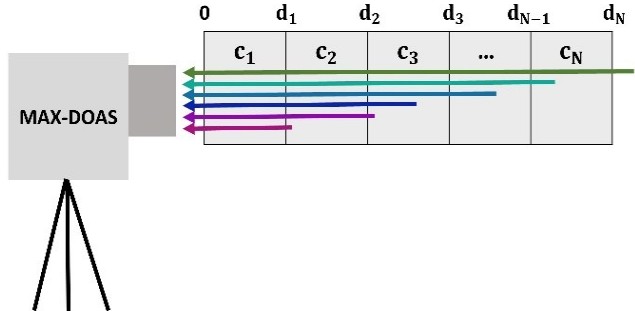

**Figure 10. Schematic representation of the six different $L_{NO2}$ (i.e., one horizontal line for each wavelength) used in the new horizontal distribution inversion approach. The length of each line shows the sensitivity of each wavelength as a function of the horizontal distance. The shortest line represents the smallest wavelength.**






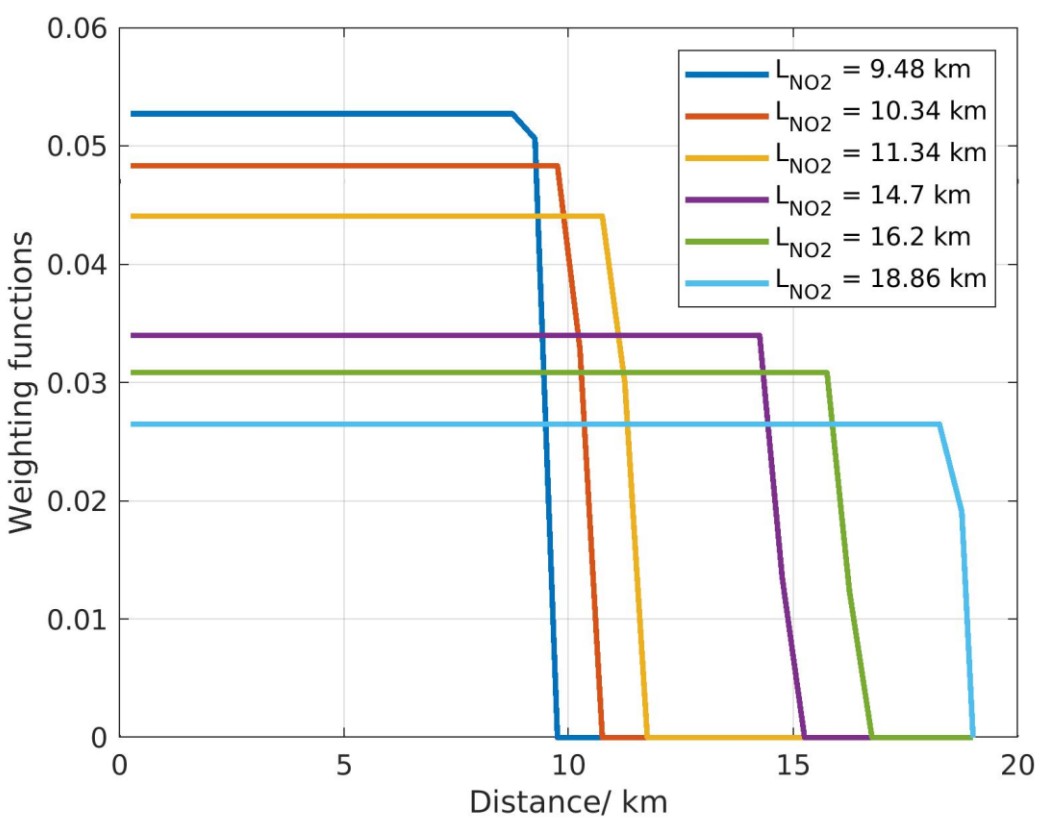

**Figure 11. Examples of weighting functions used in the new horizontal distribution inversion approach (11 September 2018).**







**Figure 12. Example of seasonal *a priori* NO₂ horizontal profiles for the new horizontal distribution inversion approach as a function of the horizontal distance from the MAX-DOAS instrument in six different azimuthal viewing directions, before the application of the scaling factor.**

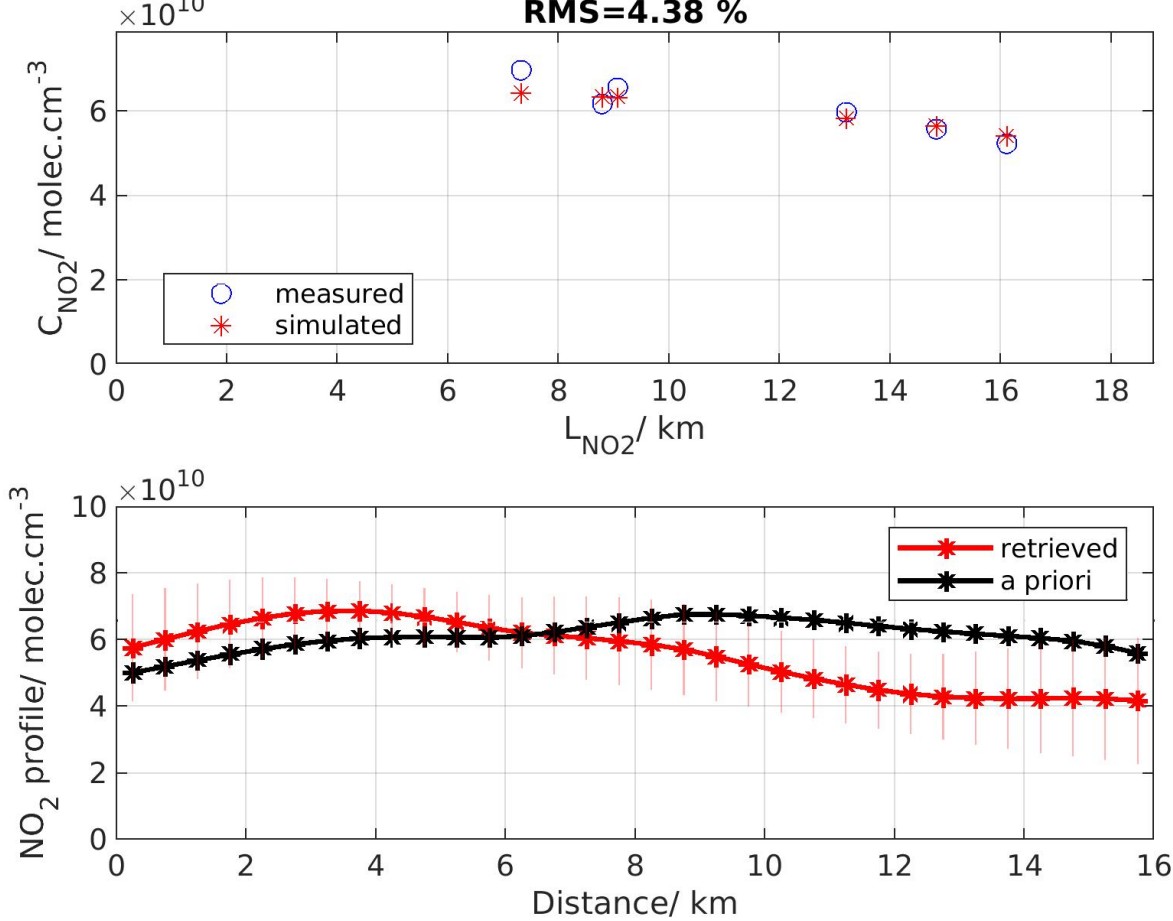

**Figure 13. (Upper panel) Measured and retrieved NO₂ near-surface concentrations at the six different wavelengths (i.e., horizontal distances) as a function of the estimated horizontal distances and (lower panel) the retrieved NO₂ near-surface horizontal profile and a priori profile (02 July 2018, 10.42 UTC, 25° azimuthal direction).**

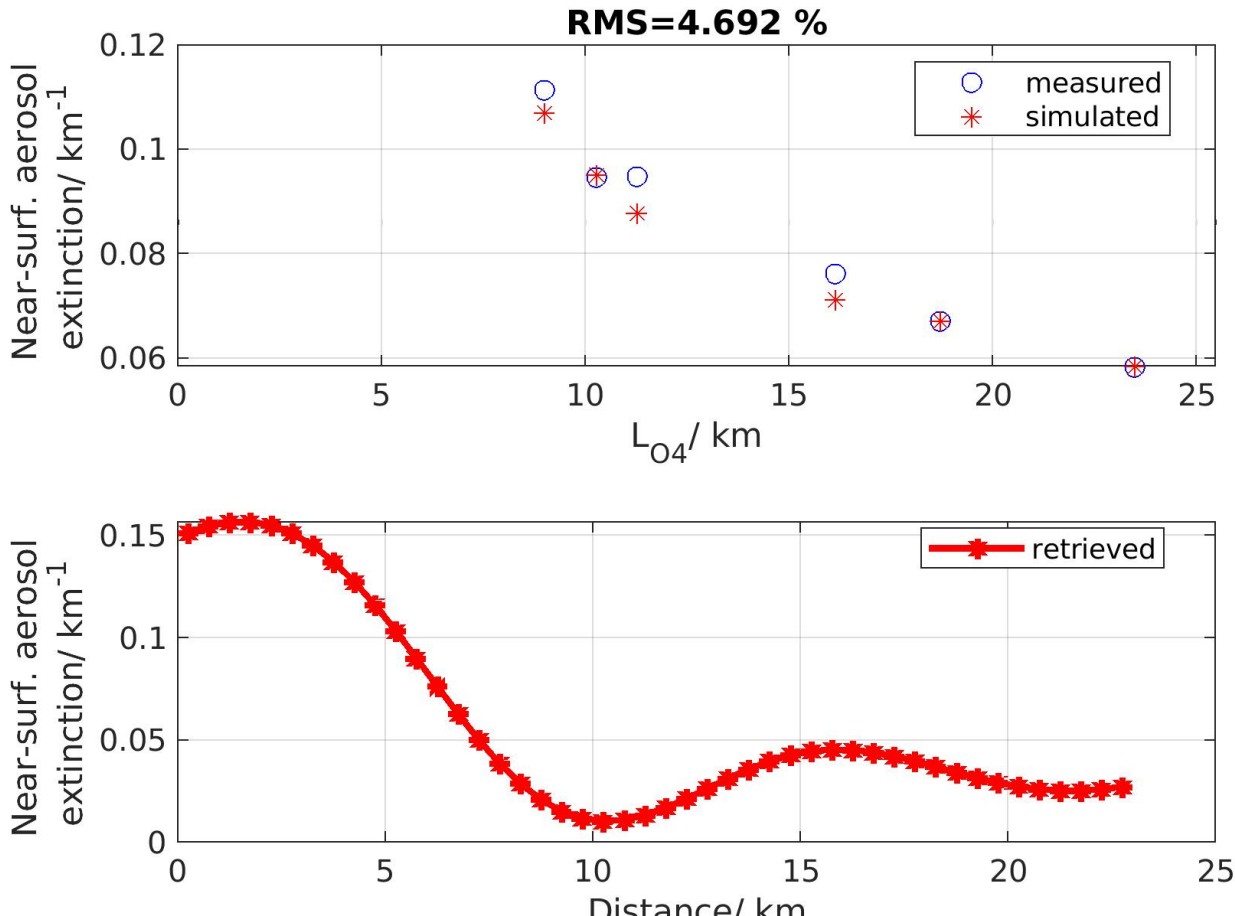

**Figure 14. (Upper panel) Measured and retrieved near-surface aerosol extinction at the six different wavelengths (i.e., horizontal distances) as a function of the estimated horizontal distances and (lower panel) the retrieved near-surface aerosol extinction horizontal profile (11 September 2018, 11.48 UTC, 167.5° azimuthal direction).**

**4.4 Characterization of the retrieval**

To characterize the retrieval, the averaging kernels, **AK**, play a crucial role. The **AK** matrix is calculated as follows (Rodgers, 2000):





$$AK = \frac{dc_{NO2}}{dc_{NO2\_true}} = (K^T S_e^{-1} K + S_a^{-1})^{-1} K^T S_e^{-1} K \tag{13}$$


The **AKs** are the rows of the **AK** matrix. They present the sensitivity of the retrieved ($c_{NO2}$) on the true ($c_{NO2\_true}$) atmospheric profile. Ideally, the **AK** matrix should be an identity matrix. In Fig. 15, an example of selected **AKs** is shown. As can be seen, for distances smaller than the first measurement (e.g., near-surface $NO_2$ concentration retrieved at 343 nm), the **AKs** are constantly zero (or have small values) from the MAX-DOAS instrument until these distances. This indicates a low sensitivity

on these short distances, and therefore information about the horizontal distribution of $NO_2$ is coming essentially from the a priori profile. The **AKs** create a maximum flat plateau close their nominal horizontal distance for larger distances (d=7.25 km, d=8.75 km, and d=15.75 km). For this particular example, the **AKs** do not exceed the values of 0.25.

Another important information about the retrieval is the trace of the **AK** matrix, which refers to the number of degrees of freedom for signal (DOFS). The DOFS are an indication of the number of independent pieces of information that one can

retrieve from the measurements. Ideally, the DOFS would be equal to the number of horizontal boxes for the horizontal distribution. In reality, the DOFS are lower, because of the limited horizontal resolution of the measurements. In Fig. 15, the DOFS are close to three, which means that three independent pieces of information are contained in the measurements for this particular example.

In the present work, the total retrieval error is equal to the error related to the measurement noise. According to Rodgers (2000), the retrieval noise error is estimated as:

$$S_{meas} = G S_e G^T \tag{14}$$

with, G being the gain matrix:

$$G = (K^T S_e^{-1} K + S_a^{-1})^{-1} K^T S_e^{-1} \tag{15}$$

The horizontal profiles of the measurement error in percentage are shown in Fig. 16. As can be seen, the measurement error becomes maximum for the longest distance.

To eliminate the unsuccessful retrievals, the percentage of accepted retrievals with respect to the total number of retrievals during the four seasons is investigated when a specific filtering on RMS and DOFS is applied (see Table 3). From these tests, it is found that most of the retrievals have DOFS larger than 1.5. RMS is defined as the root-mean-square deviation between measured and simulated $c_{NO2}$ normalized by the mean of the measured $c_{NO2}$ (e.g., same RMS as in Fig.13). Table 3 indicates that most of the retrievals have an RMS smaller than 6% with a median RMS value of around 4.5% during all seasons. Based

on these investigations, DOFS>1.5 and RMS<6% are used as retrieval quality control criteria.





**Table 3. Seasonally averaged root-mean-square (RMS) and DOFS values. RMS is calculated between measured and retrieved NO$_2$ near-surface concentrations of the horizontal retrieval (Fig. 13a). DOFS represent the degrees of freedom** 630 **of the horizontal retrieval (Fig. 15). The percentage of the accepted retrievals is presented for the different selection criteria.**

| Season | Spring | Summer | Autumn | Winter |
|---|---|---|---|---|
| **Medium RMS (%)** | 3.8 | 4.7 | 4.7 | 4.8 |
| **Median DOFS** | 1.7 | 1.7 | 1.6 | 1.6 |
| **Accepted retrievals (%) (DOFS>1.5)** | 90 | 91 | 81 | 78 |
| **Accepted retrievals (%) (RMS<6%)** | 87 | 72 | 70 | 73 |
| **Accepted retrievals (%) (RMS<5%)** | 75 | 57 | 55 | 55 |
| **Accepted retrievals (%) (RMS<4%)** | 56 | 33 | 36 | 32 |
| **Accepted retrievals (%) (RMS<3%)** | 27 | 11 | 15 | 11 |
| **Total accepted retrievals (%) (DOFS>2 & RMS<5%)** | 80 | 67 | 57 | 57 |






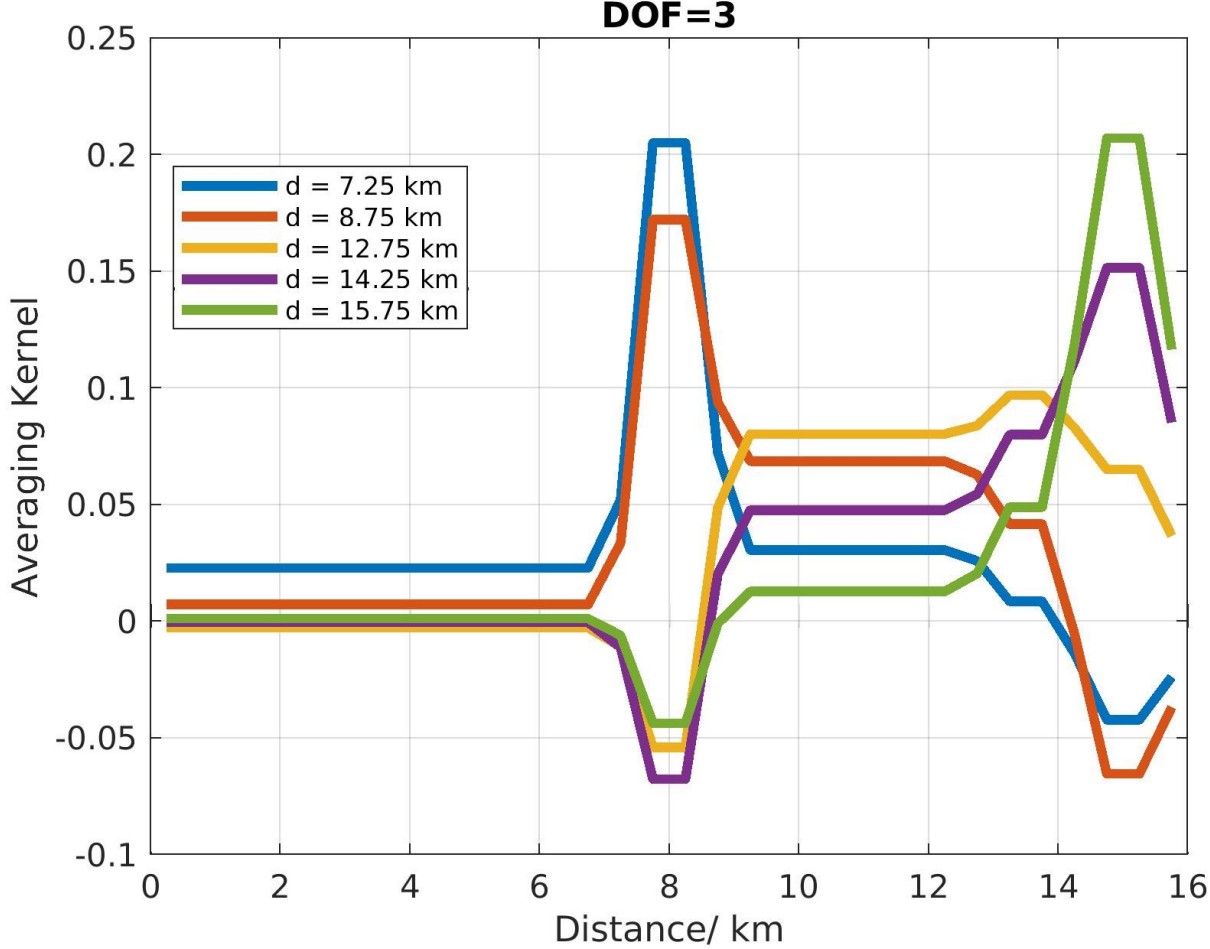

**Figure 15. Example of NO₂ averaging kernels. They are calculated for observations on 11 September 2018 at 11:51**
**UTC and 300º azimuthal direction.**





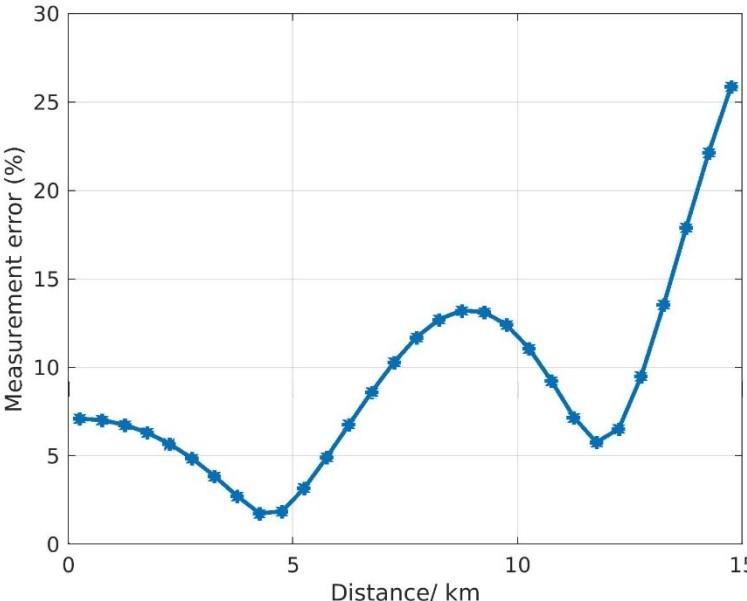

**Figure 16. Example of the NO₂ measurement error in percentage for the 2 July 2018 at 05:59 UTC and 300° azimuthal direction.**

## 5 Retrieval results and discussion

### 5.1 Example of daily horizontal NO₂ distribution

The variation of the MAX-DOAS horizontal distribution of tropospheric $NO_2$ VCDs as a function of time over the course of June 28 2019, is presented in Fig. 17. This particular day is chosen because airborne measurements took place above the Brussels region (see Sect. 5.2). The horizontal $NO_2$ profiles are plotted per azimuthal direction with the horizontal axis showing the time in UTC and the vertical axis the horizontal distance in km. Because of the quality check on the retrieved $NO_2$ horizontal profiles (see Sect. 4.3), some profiles were rejected (e.g. azimuthal direction equal to 262.5° and 265°).

During this day, maximum $NO_2$ columns are mainly observed around 05:00 UTC and 10:00 UTC, which correspond to 7am and noon local time. Early in the morning (05:00 UTC), high $NO_2$ columns are expected to be observed because of the low MLH ($MLH_{NO2}$ in the range of 300 – 600 m height) in combination with the morning rush hour $NO_2$ emissions. Around 10:00 UTC, the maximum $NO_2$ columns are detected in the north (N), northeast (NE), and northwest (NW) direction (see Fig. 18).

In the Brussels region, the main emission sources are located in the N and west (W) parts of the city and are linked to the motorway around Brussels (the so-called Ring), the Brussels city center, and the Drogenbos power plant (NW direction). Concerning the $NO_2$ peaks, they are located at a distance around 0 to 8 km from the measurement site. It can be seen from Fig. 18 that the Ring, the Brussels city center, and the Drogenbos power plant are located within these distances. As measured by





the meteorological station on the BIRA-IASB rooftop, the wind was coming from the NE direction during that day, resulting

in the progressive displacement of the $NO_2$ peak from the NNE to the W direction. On the contrary, the azimuthal directions

pointing towards a large forested area (i.e., 62.5°, 75°, and 105°), the Bois de la Cambre, detect considerably lower $NO_2$

columns than the other directions.


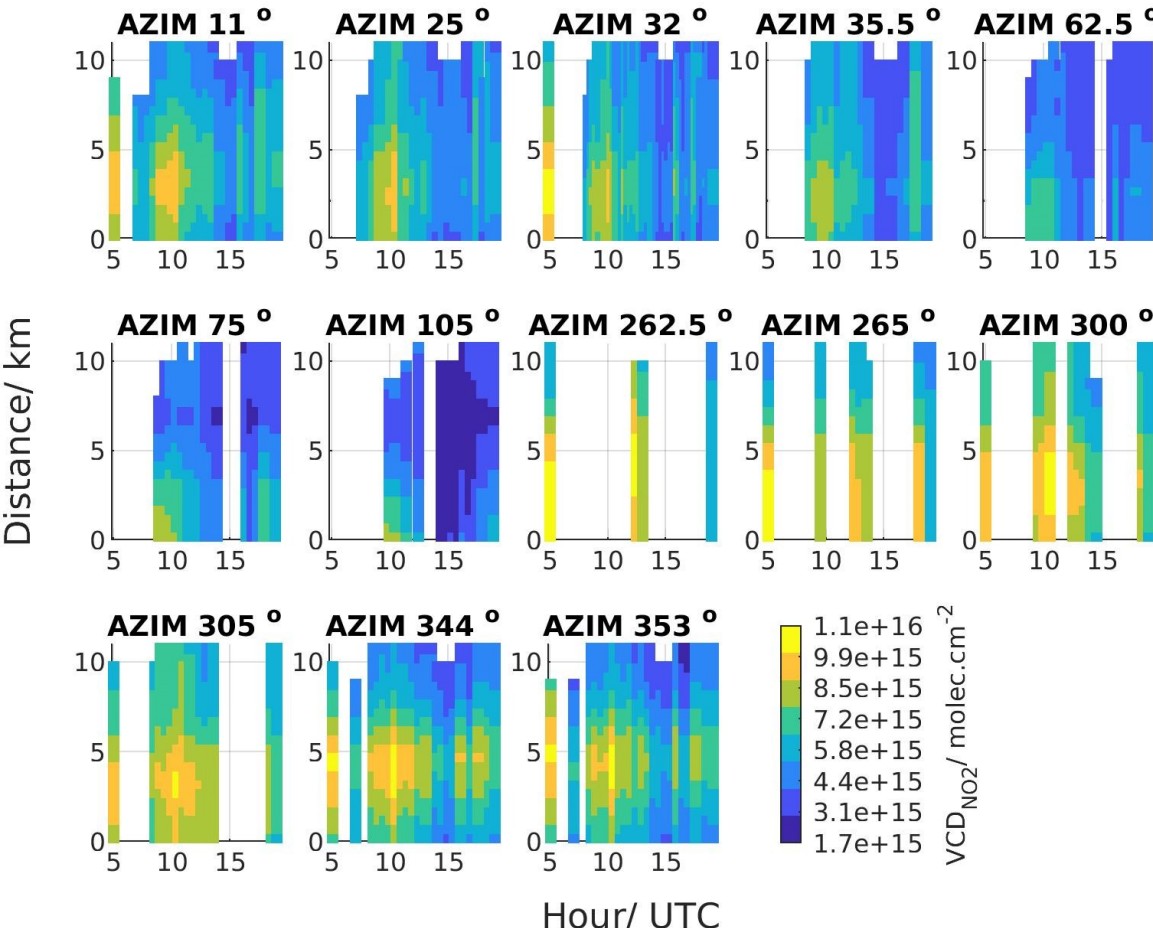

**Figure 17. Diurnal variation of the retrieved $NO_2$ horizontal profiles per azimuthal direction as a function of time (UTC) for June 28, 2019.**


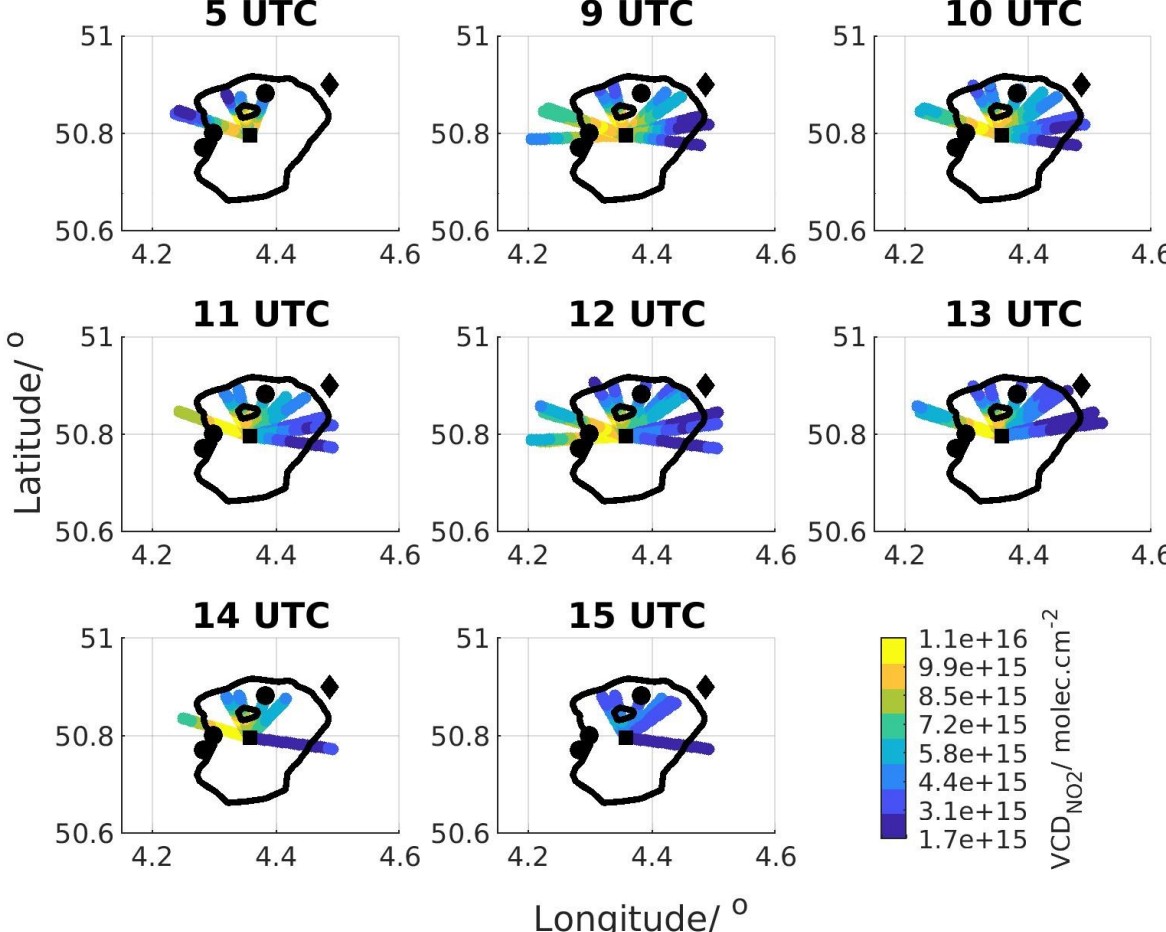

**Figure 18. Maps of hourly averaged NO₂ horizontal profiles per azimuthal direction for June 28, 2019 corresponding to Fig. 17. The wind direction was coming from the NE direction. The black square shows the MAX-DOAS instrument location, the black polygon the National Airport, the black dots the NO₂ hotspots emitting more than 10 kg of NOₓ per hour (Emission Inventory of the Belgian Interregional Environment Agency, 2017) , and the black line represents the Brussels Ring road.**






## 5.2 MAX-DOAS horizontal NO₂ distribution versus airborne, car mobile-DOAS, and TROPOMI: 28 June 2019 case study

For the S5P validation campaign over Belgium (S5PVAL-BE, https://s5pcampaigns.aeronomie.be/), airborne measurements of the two largest urban regions over Belgium, i.e., Antwerp and Brussels, took place from 26 to 29 June 2019 (Tack et al.,

2021). The Airborne Prism EXperiment (APEX) imaging spectrometer was used to measure the horizontal distribution of tropospheric NO₂ columns with a spatial resolution of approximately 75 m x 120 m (Tack et al., 2017; Tack et al., 2019).

The APEX tropospheric NO₂ columns are compared to the tropospheric NO₂ horizontal distribution as retrieved by applying our new MAX-DOAS inversion approach to the 28 June 2019 measurements. During the same day, TROPOMI pixels (OFFL 010302 product; see Table 1) selected over the Brussels region are compared to MAX-DOAS observations. MAX-DOAS

horizontal profiles of tropospheric NO₂ VCDs are selected around TROPOMI overpass time (±1 hour). The horizontal profile of MAX-DOAS NO₂ VCDs on each horizontal line-of-sight has a horizontal sampling of 0.5 km (see Fig. 13b). The MAX-DOAS NO₂ VCDs on the horizontal segment crossing a TROPOMI pixel and located inside the pixel are averaged and compared to the corresponding TROPOMI NO₂ VCD. It should be noted that the MAX-DOAS segments are not weighted by their relative length inside each pixel. APEX observations located inside each TROPOMI pixel were used to assign one APEX

NO₂ VCD value per pixel. Maps of co-located TROPOMI, averaged MAX-DOAS, and averaged APEX NO₂ VCDs for the 28 June 2019 are shown in Fig. 19. Two maps of APEX observations are presented: one with APEX in its initial resolution and one with spatially averaged APEX observations in the area covered by a TROPOMI pixel. The NO₂ plume as detected by APEX is covering the NW, N, and NE parts of the Brussels region. MAX-DOAS successfully detected the same NO₂ plume in the NW and N but not in the NE direction. The correlation and agreement between APEX and MAX-DOAS observations

is very good (R=0.83 and s=1.10). As we can observe in Fig. 20, the APEX tropospheric NO₂ VCDs tend to be larger than the MAX-DOAS ones, with an intercept equal to $-2.10 \times 10^{15}$ molec.cm$^{-2}$.

During the S5PVAL-BE flight over Brussels, car mobile-DOAS observations were performed by the BIRA-IASB mobile-DOAS, the so-called AEROMOBIL (Merlaud, 2013). The AEROMOBIL consists of a compact double Avantes spectrometer recording simultaneously scattered light in two channels (i.e., one at 30° elevation angle and one at zenith). The AEROMOBIL

was used to measure the spatial distribution of tropospheric NO₂ columns mainly over the Ring road of Brussels. Similarly as with APEX, the AEROMOBIL NO₂ VCDs, which are located inside a TROPOMI pixel are averaged and compared to the corresponding MAX-DOAS VCDs (see Fig. 19e and 19f). AEROMOBIL and MAX-DOAS agree perfectly on the location of maximum (i.e. NW direction) and minimum (i.e. SE direction) NO₂ tropospheric VCDs (Fig. 19d, 19e, and 19f). We can observe in Fig. 20b, that the correlation coefficient is moderate (R equal to 0.61) and the slope value is equal to 2.62. The

correlation plot between both datasets reveals that AEROMOBIL gives higher NO₂ tropospheric VCDs compared to MAX-DOAS ones. This finding could be partly explained by the fact that AEROMOBIL follows busy routes, where the NO₂



tropospheric VCDs reach maximum values because of the contribution of $NO_2$ production resulted by vehicles' engines via fossil fuel combustion.

The correlation between TROPOMI and MAX-DOAS tropospheric $NO_2$ columns during the day of the airborne measurements
above Brussels is presented in Fig. 20c. Excellent agreement is obtained, with a correlation coefficient value equal to 0.81. The slope value is equal to 0.72. During that day, MAX-DOAS and TROPOMI are in good agreement but TROPOMI tends to underestimate the tropospheric $NO_2$ columns. It should be noted that during that day, the range of observed $NO_2$ VCDs is from $3.4\times10^{15}$ to $8.7\times10^{15}$ molec.$cm^{-2}$, as retrieved by the MAX-DOAS observations.





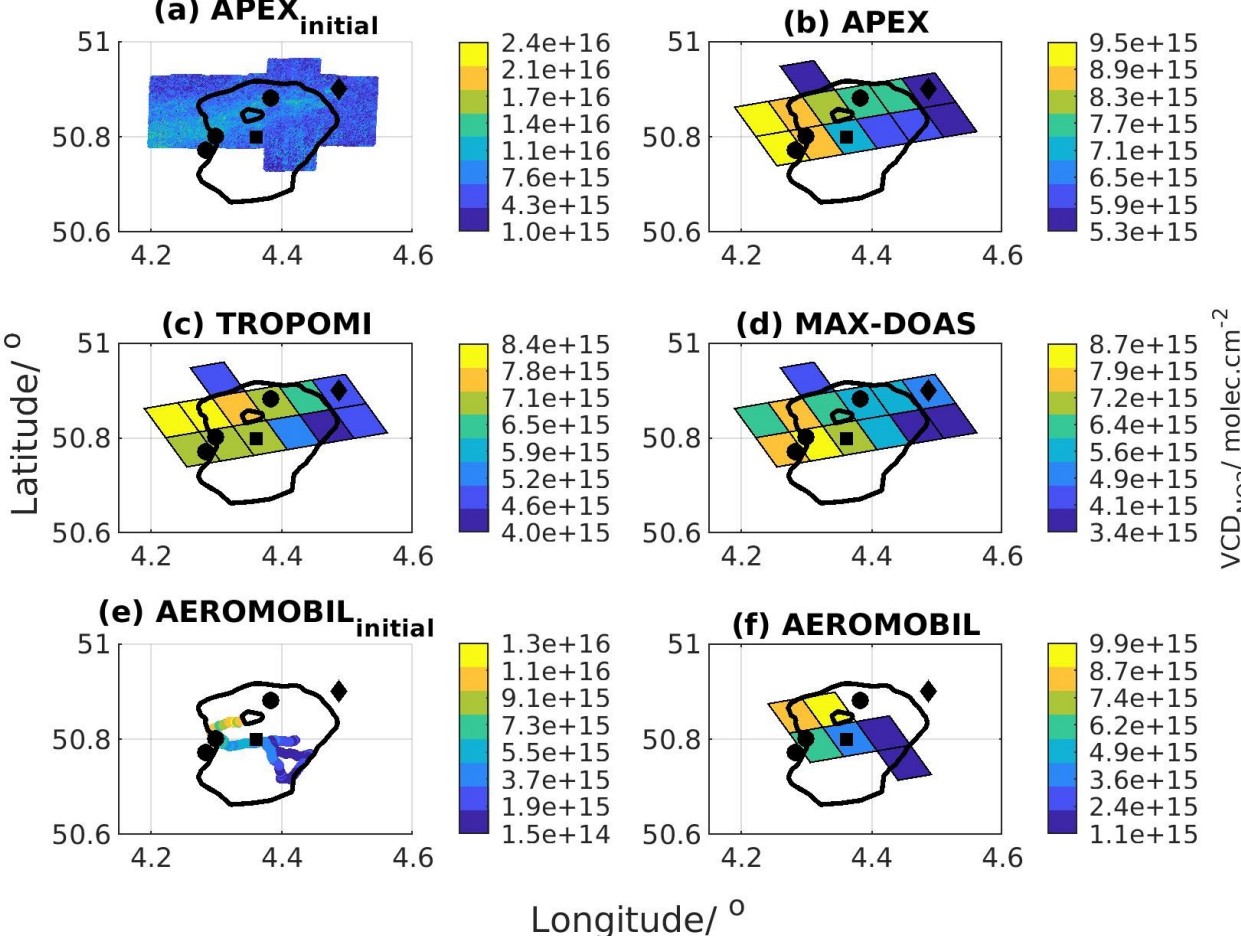

**Figure 19. (a) Tropospheric NO₂ VCD as detected by the APEX instrument in its initial spatial resolution. Tropospheric NO₂ VCD maps (TROPOMI pixels) as retrieved over Brussels on 28th of June 2019 by the (b) APEX, (c) TROPOMI, (d) MAX-DOAS instruments. (e) Tropospheric NO₂ VCD as retrieved by the AEROMOBIL in its initial spatial resolution and (f) AEROMOBIL tropospheric NO₂ VCD in the TROPOMI pixels. The black square shows the MAX-DOAS instrument location, the black polygon the National Airport, the black dots the NO₂ hotspots emitting more than 10 kg of NOₓ per hour (Emission Inventory of the Belgian Interregional Environment Agency, 2017) , and the black line represents the Brussels Ring road.**



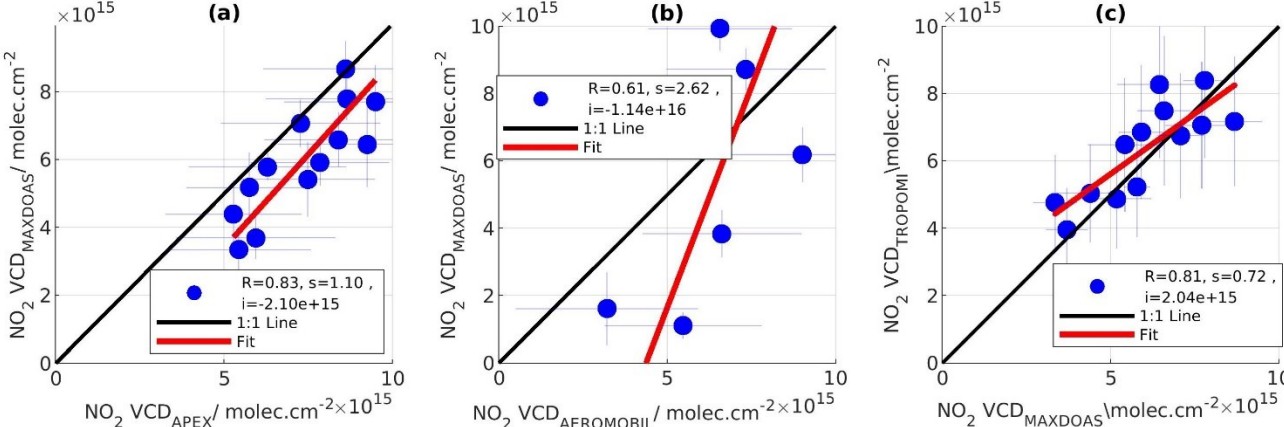


**Figure 20. Scatter plot between (a) the tropospheric NO₂ columns derived by airborne measurements (APEX) and the MAX-DOAS observations, (b) the tropospheric NO₂ columns derived by car mobile-DOAS measurements (AEROMOBIL), and the MAX-DOAS observations and (c) the tropospheric NO₂ columns derived by MAX-DOAS observations and the TROPOMI tropospheric NO₂ columns over Brussels on 28th of June 2019.**


## 5.3 MAX-DOAS horizontal NO₂ distribution versus TROPOMI observations

### 5.3.1 Comparison results of the March 2018-February 2020 period

To compare the TROPOMI and MAX-DOAS tropospheric NO₂ columns, the following 5-step approach is used, similarly as in Section 5.2:

1. Only MAX-DOAS horizontal profiles of tropospheric NO₂ VCDs retrieved around (±1 hour) TROPOMI overpass time are selected.

2. The time-coincident MAX-DOAS tropospheric NO₂ VCD horizontal grids from all the azimuthal directions are spatially averaged (i.e. one MAX-DOAS mean NO₂ VCD value per pixel) within the overlapping TROPOMI pixels.

3. To take into account the distance between each azimuthal direction crossing a TROPOMI pixel and the TROPOMI
pixel center, the MAX-DOAS average is a weighted mean with the weighting depending on their relative direction with respect to the direction of the TROPOMI pixel center. Consequently, the weights are equal to the difference between 360° and the azimuthal difference between MAX-DOAS grid and TROPOMI pixel central coordinates.

4. The horizontal profiles of MAX-DOAS NO₂ columns have a horizontal sampling of 0.5 km in every azimuthal direction. The coverage percentage is estimated as the ratio of the area covered by MAX-DOAS (i.e., number of
coincident MAX-DOAS NO₂ VCDs considering that every MAX-DOAS horizontal grid has a spatial resolution of 0.5 x 0.5 km²) inside each TROPOMI pixel to the total number of MAX-DOAS NO₂ VCDs that could fill-in the TROPOMI pixel.





5. TROPOMI and MAX-DOAS tropospheric $NO_2$ columns are compared, and the seasonally-averaged maps of those VCDs on the area covered by the TROPOMI pixels are created. To generate these maps, the ensemble of TROPOMI pixels recorded on 28 June 2019 is chosen as reference and TROPOMI pixels that coincide with this reference grid are averaged. The daily horizontal profiles of MAX-DOAS $NO_2$ columns are averaged on the daily TROPOMI grids and then, the reference grid is used to create the seasonally-averaged MAX-DOAS maps.

The seasonally and annually-averaged maps of TROPOMI and MAX-DOAS $NO_2$ VCDs are presented in Fig. 21 and Fig. 22. Only pixels including at least 20 comparison days are taken into account in the analysis. It is found that the locations of the $NO_2$ peaks and dips show a reasonably high degree of similarity between TROPOMI and MAX-DOAS during all seasons. The $NO_2$ peaks appear mainly above Brussels city center, the Drogenbos power plant (W direction) and the NW part of the Ring road, which are the main known emission sources, as mentioned earlier. These maps also indicate that the tropospheric $NO_2$ column over the Brussels area has a clear seasonal cycle, with a maximum during winter.

Figure 22c shows the annual relative biases (e.g., 100 x (TROPOMI – MAX-DOAS)/MAX-DOAS) per pixel. It is found that positive biases (i.e. TROPOMI larger than MAX-DOAS) are observed mainly in the pixels located away from the measurement site during all seasons, while negative biases are found close to the measurement site and in the Brussels city center.

The seasonal correlation plots for April 2018-February 2020 are displayed in Fig. 23. When all pixels are included, without any TROPOMI pixels coverage percentage filtering on MAX-DOAS data, the highest correlation is found during spring (R=0.66), while lower correlations are reported in autumn, summer, and winter, with correlation coefficient values of 0.65, 0.65, and 0.58, respectively. It should be noted that during spring (2018 and 2019), the number of comparison points is smaller than for the other seasons, because TROPOMI data start from end of April 2018. During spring, the slope value is equal to 0.90, while during winter, summer, and autumn, the slope values are smaller (0.64 0.56, and 0.64, respectively), which means that TROPOMI underestimates MAX-DOAS measurements up to 50 %. A similar underestimation has been reported in several studies (Verhoelst et al., 2021; Tack et al., 2021; Judd et al., 2020; Dimitropoulou et al., 2020; Ialongo et al., 2019). When seasonally-averaged TROPOMI and MAX-DOAS pixels (the pixels shown in Fig. 21) are compared one-by-one (see SEAS in Fig. 23), both correlation coefficient (R in the range of 0.57-0.93) and slope values (s in the range of 0.65-0.94) improve considerably.

In a second step, the impact of the spatial sampling is investigated. Generally, a varying number of MAX-DOAS $NO_2$ columns cover each TROPOMI pixel. The coverage percentage is estimated as the ratio of the covered area by MAX-DOAS (i.e., number of coincident MAX-DOAS $NO_2$ VCDs) inside each TROPOMI pixel to the total number of MAX-DOAS $NO_2$ VCDs that could fill-in the TROPOMI pixel. When selecting only TROPOMI pixels covered by at least a given percentage of MAX-DOAS grids (10% and 20%), it is found that the correlation between both datasets improves for all seasons, except summer for a coverage equal and greater than 20%. The most significant improvement is observed during spring. The correlation coefficient value is equal to 0.83 (instead of 0.66) when taking into account TROPOMI pixels covered more than 20% by



MAX-DOAS retrievals. Despite the better agreement in terms of correlation coefficient, TROPOMI columns are still 30 % lower than MAX-DOAS measurements, in line with previously published studies.

The seasonal regression analysis parameters between TROPOMI and dual-scan MAX-DOAS measurements derived in the present study are compared to the same parameters presented in Dimitropoulou et al. (2020). Both studies make use of the dual-scan MAX-DOAS instrument in Uccle. In addition to the different approach (i.e., the retrieval of $NO_2$ horizontal profiles), in the present study, almost two years of measurements are used, while in Dimitropoulou et al. (2020), only one year is exploited for the TROPOMI validation. In Table 4, for the present study, only one year of measurements are used to have a

comparable time coverage for both studies. As presented in Table 4, here, the largest slope value is found in spring, while in Dimitropoulou et al. (2020), in winter. The season in which the highest correlation coefficient is obtained differs between both studies (here, in spring, in autumn in Dimitropoulou et al. (2020)). The main advantage of the new approach is the larger number of comparison points between TROPOMI and MAX-DOAS leading to significantly more reliable statistics. In the present study, the deviation of the comparison points from the fitted regression line is increased mainly because of the

uncertainties in the horizontal inversion approach. The scatter increase is reflected in the correlation coefficient values, which are smaller for all seasons, except winter. Regarding the slope value, it is larger in spring and summer, and is smaller in autumn and winter.

Overall, our investigation about the spatial sampling lead to the following three important findings:

1. The dual-scan multi-wavelength approach allows a better identification of the main emissions sources in urban regions, in agreement with the spatial allocation of the main emission sources observed by APEX and TROPOMI.

     2. The characterization of the $NO_2$ concentration horizontal field using the dual scan multi-wavelength approach results in obtaining larger slope values between TROPOMI and MAX-DOAS observations. The high spatial resolution of TROPOMI requires ground-based measurement that can provide information about the horizontal distribution of

tropospheric $NO_2$ columns in urban regions.

     3. Even for a better spatial sampling between TROPOMI and ground-based observations, TROPOMI still underestimates the ground-based measurements (see Fig. 22). Therefore, this is an additional indication that this underestimation is caused by other factors.







**Figure 21. Seasonal tropospheric NO₂ VCD grids (TROPOMI grids) as retrieved over Brussels by the TROPOMI and MAX-DOAS instruments. The black square shows the MAX-DOAS position, the black polygon the National Airport, the black dots the NO₂ hotspots, and the black line represents the Brussels Ring motorway.**








**Figure 22. Annual (e.g., based over the two years of observations) tropospheric NO₂ VCD grids (TROPOMI grids) as retrieved over Brussels by the (a) TROPOMI and (b) MAX-DOAS instruments. (c) Annual bias between tropospheric**



**NO₂ VCD as observed by TROPOMI and MAX-DOAS instruments (the negative values are shown with blue color, zero with white, and positive values with red). The black square shows the MAX-DOAS instrument location, the black polygon the National Airport, the black dots the NO₂ hotspots, and the black line represents the Brussels Ring road.**








**Figure 23. Seasonal scatter plots of tropospheric NO₂ columns derived from the dual-scan MAX-DOAS and TROPOMI**
**measurements over Brussels. Magenta line: Regression analysis results when all the MAX-DOAS and TROPOMI pixels**
**are included in the comparison. Green and black lines: Regression analysis results when TROPOMI pixels covered**
**(i.e., COV) by more than 10 and 20 % of the horizontal profiles of MAX-DOAS NO₂ columns are included in the**
**comparison. Red line: Seasonal average analysis generated by the pixels in Fig. 21.**




**Table 4. Summary of the regression analysis parameters (e.g., correlation coefficient (R) and slope (s)) and the number of data points (N) derived in the present study during only one year of observations and in Dimitropoulou et al. (2020).**

| Season | Spring | Summer | Autumn | Winter |
|---|---|---|---|---|
| **R** | 0.66 | 0.60 | 0.57 | 0.62 |
| **R (seasonal)** | 0.93 | 0.88 | 0.46 | 0.75 |
| **R (Dimitropoulou et al., 2020)** | 0.69 | 0.77 | 0.85 | 0.60 |
| **s** | 0.90 | 0.76 | 0.56 | 0.60 |
| **s (seasonal)** | 0.94 | 0.87 | 0.74 | 0.70 |
| **s (Dimitropoulou et al., 2020)** | 0.47 | 0.58 | 0.61 | 0.81 |
| **N** | 139 | 247 | 106 | 92 |
| **N (Dimitropoulou et al., 2020)** | 16 | 58 | 36 | 13 |

### 5.3.2 Investigation of the a priori $NO_2$ profile shape and clouds in TROPOMI $NO_2$ retrievals

Three additional comparisons were conducted in this study. First, a TROPOMI tropospheric $NO_2$ column product with an improved FRESCO-S cloud retrieval was tested. As discussed in Dimitropoulou et al. (2020), clouds can significantly affect tropospheric $NO_2$ VCD retrievals from satellite observations. The dataset is available for four different periods in 2018 – 2019 (see Sect. 3). Fig. 24 shows that the slope value increases by about 56% (equal to 0.53 instead of 0.34 for the baseline product), as well as the correlation coefficient between both datasets (R equal to 0.68 instead of 0.45). This is in agreement with the TROPOMI Routine Operations Consolidated Validation Report (ROCVR; https://mpc-vdaf.tropomi.eu/), where the use of the improved FRESCO-wide resulted in a bias reduction with respect to ground-based $NO_2$ data.

Secondly, a new TROPOMI data product covering the November 2018 to February 2020 period is used. In this product, the coarse TM5-MP a priori $NO_2$ profiles are replaced by $NO_2$ profile shapes from the CAMS regional CTM ensemble at a spatial resolution of $0.1^o$ x $0.1^o$ (Douros et al., in preparation; Ialongo et al., 2019; Tack et al., 2021). As can be seen in Fig. 24, using a spatially finer a priori $NO_2$ vertical profile improves slightly the slope value, which is equal to 0.77 (instead of 0.75 for the baseline TROPOMI product). This represents an increase of the slope by about 3%. This finding indicates that part of the TROPOMI underestimation of tropospheric $NO_2$ columns is caused by inadequate a priori profiles in the TROPOMI retrievals





for urban conditions. On the other hand, the fact that the slope value is still lower than unity, even when CAMS regional a priori profiles are used, indicate that other factors contribute to the TROPOMI underestimation or that CAMS profiles are still sub-optimal, as suggested by results obtained when applying MAX-DOAS profiles to TROPOMI (see below).

Finally, the impact of the a priori profile in the TROPOMI $NO_2$ retrieval is investigated using MAX-DOAS profile data. For this test, TROPOMI $NO_2$ columns are recalculated, similarly as in Dimitropoulou et al. (2020), using daily median MAX-DOAS vertical profiles derived in the main azimuthal direction by applying the MMF inversion algorithm. Those TROPOMI $NO_2$ columns are then compared to the horizontally-resolved MAX-DOAS data, as in Sect. 5.3.1. Figure 25 presents the comparison results per season. When comparing it with Fig. 23, we find that the change in the $NO_2$ vertical profile shape

improves the slope value in the comparison with ground-based observations. Except for winter, the slopes are largely improved (slopes in the 0.56 - 1.11 range) due to an increase of the recalculated TROPOMI columns. This result confirms once again that the a priori profile in the TROPOMI retrieval is a key player in the TROPOMI underestimation of tropospheric $NO_2$ columns in urban conditions, as already stated in previous studies (see e.g. Dimitropoulou et al., 2020; Ialongo et al., 2019; Tack et al., 2021). The present study suggests that in urban conditions, daily median MAX-DOAS vertical profiles are more

suitable than $NO_2$ profile shapes from the CAMS regional CTM ensemble in order to be applied as a priori information in the TROPOMI retrieval.







**Figure 24. Scatter plots between the tropospheric NO$_2$ columns derived from the dual-scan MAX-DOAS instrument and the TROPOMI pixels over Brussels. The left plots are for the baseline TROPOMI dataset ((a) and (c) panels), while the right plots correspond to two new versions of TROPOMI datasets ((b): improved FRESCO-S cloud product; (d) NO$_2$ a priori profiles from the CAMS regional CTM ensemble).**







**Figure 25. Seasonal scatter plots between the horizontally-averaged MAX-DOAS NO₂ VCDs and TROPOMI NO₂ columns recalculated using median daily MAX-DOAS vertical profiles as a priori information.**

## 6 Conclusions

Two years (March 2018 to February 2020) of dual-scan MAX-DOAS measurements in Uccle (urban background site located in the south of the Brussels-Capital Region) were used to develop a new strategy for the retrieval of near-surface NO₂ concentrations and aerosol extinction horizontal profiles. A full dual-scan measurement is composed of one vertical scan at a fixed azimuthal direction pointing towards the city center and horizontal scans in ten azimuthal directions at a fixed low elevation angle (2°).



The first step of this new retrieval strategy is to analyze measured radiance spectra in six different fitting windows. This provides $O_4$ and $NO_2$ dSCDs at the following six wavelengths: 343 nm, 360 nm, 380 nm, 447 nm, 477 nm and 530 nm. Then, information about the vertical extent of $NO_2$ in the troposphere ($MLH_{NO2}$) is derived from profile retrievals in the main azimuthal direction performed using the OEM-based MMF algorithm. In the third step, a new parameterization technique is applied, with $MLH_{NO2}$, measured $O_4$ dSCDs, and measurement geometry being used as input parameters to retrieve the horizontal sensitivity of $NO_2$ and, consequently, the $NO_2$ near-surface concentrations and VCDs, and near-surface aerosol extinction in all the azimuthal directions for the six different wavelengths. Compared to the method presented in Dimitropoulou et al., (2020), the new retrieval method offers the possibility of the direct determination of $L_{NO2}$, and near-surface aerosol extinction based on the measured $O_4$ dSCDs.

The retrieved dual-scan $NO_2$ near-surface concentrations and VCDs are verified via comparisons to the MMF $NO_2$ vertical profiles in the main azimuthal directions and in three additional azimuthal directions. A good overall agreement is found for the two comparisons during the two years of measurements.

The dependence of the horizontal sensitivity on the wavelength is then used to develop a new OEM-based horizontal distribution inversion approach. Considering a horizontal box model, horizontal $NO_2$ and aerosol extinction profiles are retrieved in an output horizontal grid of 500m thickness starting from the instrument to each of the measurement maximum horizontal representative distance.

The daily variability of $NO_2$ horizontal profiles in all the azimuthal directions provides information about the location of the $NO_2$ hotspots in the Brussels-Capital Region and how the plumes are transported. Similarly, the $NO_2$ horizontal profiles' seasonal variability over March 2018-February 2020 reveals that the $NO_2$ hotspots are mainly found above the Brussels city-center, the Drogenbos power plant and the NW part of the Ring road during all seasons.

On 28 June 2019, airborne measurements (APEX) of $NO_2$ were performed over Brussels. The MAX-DOAS $NO_2$ VCD horizontal profiles are compared to APEX, mobile car-DOAS (i.e., AEROMOBIL), and TROPOMI measurements, and a good overall agreement is found between the different data sets for this day.

In the second part of the study, MAX-DOAS retrievals are compared to TROPOMI tropospheric $NO_2$ observations over the March 2018- February 2020 period. The comparison of seasonal maps shows a good overall agreement between both datasets as to the $NO_2$ horizontal distribution over the Brussels area. This agreement improves systematically when only TROPOMI pixels covered by a minimum of 20% of MAX-DOAS grid cells are compared, showing the benefit of ground-based measurements at high horizontal resolution for the validation of high-resolution space-borne air-quality measurements. Results also show that during all seasons, TROPOMI underestimates the MAX-DOAS tropospheric $NO_2$ columns. The role of the a priori $NO_2$ profile shape in the TROPOMI retrievals was investigated and TROPOMI tropospheric $NO_2$ columns are recalculated with the MAX-DOAS vertical profiles. We show that the knowledge of the $NO_2$ horizontal distribution derived by the MAX-DOAS measurements combined with a more adequate a priori profile in TROPOMI retrievals leads to a much better agreement between satellite and ground-based data.

To conclude, our study presents a new horizontal distribution inversion approach for $NO_2$ and aerosols developed by using dual-scan multi-wavelength MAX-DOAS measurements over an urban area. This approach provides a better characterization of the horizontal distribution of an important urban pollutant, $NO_2$, which leads to an improved agreement between satellite and MAX-DOAS measurements in moderate to highly polluted conditions. Based on our study, further modifications of the

measurement mode aiming at a better sampling of the vertical and horizontal $NO_2$ distribution could be implemented and investigated. For instance, performing vertical scans in several azimuthal directions throughout the day and/or horizontal scans in more than ten azimuthal directions could further improve our knowledge about the tropospheric $NO_2$ spatial variability in urban regions, and therefore the satellite validation results in those conditions.

*Data availability*. The datasets generated and analyzed in the present work are available from the corresponding author on request.

*Author contributions*. ED undertook the development and validation of the dual-scan multi-wavelength MAX-DOAS retrieval strategy in Uccle, exploited the MAX-DOAS retrievals during two year, performed the validation of the TROPOMI

tropospheric NO2 columns, and wrote the paper. FH supported and guided ED in the development of the dual-scan multi-wavelength MAX-DOAS retrieval strategy, provided general guidelines, and revised and edited the paper. MMF provided the MMF inversion algorithm and the RTM as well as supporting and guiding ED in the new OEM-based horizontal profile retrieval. FT provided the airborne APEX dataset and contributed to scientific discussions. GP provided the dataset of the TROPOMI tropospheric NO2 columns and supported ED in the TROPOMI validation approaches. AM provided the

AEROMOBIL dataset and contributed to scientific discussions. CF and CH provided technical and software support for the MAX-DOAS instrument in Uccle. CF developed the QDOAS software and guided ED in the DOAS analysis. FF provided the RIO model dataset. MVR supervised the present work, provided general guidelines and valuable comments during the whole process of the paper preparation, and revised and edited the paper. All authors reviewed, discussed and commented on the paper.


*Competing interests*. I declare that I or my co-authors have competing interests as follows: Michel Van Roozendael is associate editor of AMT.

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
