# Peer review of "Horizontal distribution of tropospheric NO2 and aerosols derived by dual-scan multi-wavelength MAX-DOAS measurements in Uccle, Belgium"

_Atmospheric Measurement Techniques, 2021_

## Author Comment (AC1)

**Response to Anonymous Referee #1**

We want to thank the referee for the thoughtful examination of our paper entitled "Horizontal distribution of tropospheric $NO_2$ and aerosols derived by dual-scan multi-wavelength MAX-DOAS measurements in Uccle, Belgium".

**Please find below our responses to each comment individually.**

**Please consider that:**

A) Green bold: Comments of the Referee

B) Black: The response to each comment posed by the referee.

**C) Black bold: Already existing text in the manuscript.**

D) Red bold: Added text in the manuscript according to referee's comments.

General comments:

1. **In my opinion a distinction between the VCD_NO2 and c_NO2 measured in main viewing direction (retrieved by traditional profile inversion) and the values retrieved in other azimuthal directions would help a lot for clarification (e.g. page 9 line210, page 10 equation 1, page 11 equation (4))**
   **Response:** The $VCD_{NO2}$ and $c_{NO2}$ derived by applying the MMF inversion algorithm (i.e., traditional profile inversion) are named as $VCD_{NO2,main}$ and $c_{NO2,main}$ (please see Section 4.1), while the $VCD_{NO2}$ and $c_{NO2}$ derived by applying the parameterization technique are kept the same.

2. **Inventing a new name for this new technique would help especially when comparing to other data-sets (e.g. in chapter 5.2). Usually, elevation scans are called "MAX-DOAS". Something like "Mapping MAX-DOAS" would help to distinguish between the different kind of retrievals.**
   **Response:** The ensemble of the developed methodology, as described in Section 4, is now called "Mapping MAX-DOAS technique" and the title of this Section is "**Description of the mapping MAX-DOAS technique**". The authors have made the necessary changes to help the reader to distinguish the different retrievals used in the present study. The changes are in the following sections of the revised manuscript: Abstract, Introduction, Section 4, Figure 3, Section 5.2, and Section 5.3.

3. **Is a seasonal analysis really useful for NO2? An analysis sorted for different temperature regimes could be more useful. Also, an analysis sorted for several wind regimes could be interesting, especially as 3 huge NO2 emitters are in the region.**

   **Response:** Our primary motivation for presenting a seasonal analysis for $NO_2$ is to allocate the three huge $NO_2$ emitters in the Brussels Capital Region on a seasonal basis with the aid of seasonal $NO_2$ maps. Based on this motivation, we present the comparisons between MAX-DOAS and TROPOMI on a seasonal basis, too.
   During the preparation of the present manuscript, we have investigated the effect of temperature on $NO_2$ VCD by using co-located temperature measurements, and no

temperature dependence is found for NO$_2$ (in contrast with HCHO; please see the article: https://ieeexplore.ieee.org/abstract/document/9553326).
The effect of wind direction on NO$_2$ VCD has been investigated in our previous publication. Generally, when the wind is blowing from the NE and SE direction, higher NO$_2$ are retrieved. The corresponding results can be found at: https://amt.copernicus.org/articles/13/5165/2020/.

4. **The RIO data is available in an hourly model. It would be interesting to see a comparison to this data set.**
   **Response:** In the revised version of the present manuscript, a comparison between the NO$_2$ horizontal profiles and the RIO data on an hourly basis is performed and presented in Section S1.

5. **In a future study the comparison could also be performed for AOD with the 1km resolution data from MODIS.**
   **Response:** In a future study, a comparison between the retrieved near-surface aerosol extinction coefficient and the AOD from MODIS could indeed be done for validating the retrieved dual-scan aerosol horizontal profiles.

6. **Harmonize axes and color bars of plots in one figure.**
   **Response:** Following the specific comments, we have harmonized axes and color bars of plots in the following figures: Fig. S1, S2, S3, S4, 12, and 19.

**Specific comments:**

1. **Page 2 line 29: If possible, restructure sentences for a better reading flow.**
   **Response:** The sentences have been rephrased as follows:
   **Aerosols with a small diameter can penetrate deeply into the lungs, causing millions of premature deaths around the world per year (Khomenko et al., 2021). Additionally,** aerosols influence the Earth's climate system by changing its radiation budget by scattering and absorbing sunlight (Quaas et al., 2008).

2. **Page 6 figure 2: Why are the reference lines not exactly at the maximum of O4 for all windows?**
   **Response:** In each chosen fitting window, the reference wavelength corresponds to the maximum of O4 absorption peak or close to this maximum. In some windows, the reference wavelength is close to the maximum. This choice is because a better DOAS fit in terms of RMS is achieved.

3. **Page 7: line 55: DDS is defined, what is the difference between DDS and DDS2?**
   **Response:** In the revised version of the present manuscript, we refer to this dataset as DDSv2 because the DDS data are made with version v2.1 and v2.2 (see Section S2).

4. **Page 7 table 1: Data products 01.03.00 and 01.03.01 have an overlap in time. What is used in this period?**
   **Response:** In reality, there is no overlap in time but a typo mistake in Table 1. We have corrected the end date of v01.03.00, which is 23/04/2019.

5. **Page 8 figure 3: In yellow box last line: "1 EA, many TAA"**

**Response:** The remark is correct and Figure 3 has been updated. Additionally, figure's caption has been modified as follows:

**Figure 3. Mapping MAX-DOAS technique flow chart.**

6. **Page 8 line 205: Is all data flagged for clouds or only the main viewing direction?**
   **Response:** The information about the cloud coverage comes from the main viewing direction. In every full scan (i.e., vertical and azimuthal scan), we use the main viewing direction to estimate the $MLH_{NO_2}$. When a vertical scan is characterized as cloudy, the full scan is excluded from the analysis (i.e., vertical and azimuthal scan).

7. **Page 9 line 211: Add an equation, as this value is used in all future calculations.**
   **Response:** An equation for estimating the MLH of $NO_2$ has been added (Equation 1).

8. **Page 10 line 235: Replace ":" by ",".**
   **Response:** We have replaced ":" by ','.

9. **Page 10 line 252: State clearer, that this sentence refers to the old paper.**
   **Response:** The paragraph has been slightly modified as follows:

   **Assuming a homogeneous $NO_2$ distribution inside the MLH, the MLH is derived from the $NO_2$ vertical profiles in the main azimuthal direction and is defined as the ratio of the $NO_2$ VCD to the near-surface concentration of $NO_2$. In Dimitropoulou et al. (2020), the RTM simulations were performed for eight different MLH values of aerosols and $NO_2$ in the range of 500-2000 m (i.e. eight different combinations) and for different measurement viewing geometries (Solar Zenith Angle (SZA), Relative Azimuth Angle (RAA) and the corresponding elevation angle of 2°). For every MAX-DOAS measurement, one value of the correction factor is given according to its viewing geometry and MLH value during the measurement. For further information, we refer the reader to Dimitropoulou et al. (2020).**

10. **Page 11 line 276: Equation (4) and the following sentence are confusing next to each other (I understood it after reading the next two pages). In the equation is written dSCD_NO2_simulated, in the text are mentioned simulated O4 DSCDs. Explain in more detail.**
    **Response:** Section 4.2.1 has been revised. Additionally, this sentence has been modified, as follows:
    **Here, $NO_2$ dSCDs and consequently $L_{NO_2}$ are simulated using the radiative transfer model VLIDORT version 2.7 (Spurr, 2006).**

11. **Page 11 line 281: Was the impact of this fixed NO2 concentration investigated? If typical high values for Brussels are used, does the retrieval significantly change?**
    **Response:** In the present study, the impact of this fixed $NO_2$ concentration is used to estimate the uncertainty related to the estimation of $L_{NO_2}$ from the RTM simulations. In the case of a box profile, the values of $1.5*10^{11}$ molec.cm$^{-3}$ and $1*10^{11}$ molec.cm$^{-3}$ are used. When other profile shapes are used, such as linearly decreasing vertical profiles, the near-surface $NO_2$ concentration was fixed to $1.5*10^{11}$ molec.cm$^{-3}$, with decreasing values in the

upper layers. This impact is estimated to be about 9.6% in the visible range (see Section 4.2.2).

**12. Page 12 line 298: "are" instead of "is" (or restructure sentence)**
**Response:** Corrected (see page 12).

**13. Page 13 line 307: Which order is the polynomial fit?**
**Response:** The polynomial fit is of second order. This information is already given in the caption of Figure 5.

**14. Page 13 line 310: Is this statement also true for the high pollution cases in Brussels?**
**Response:** In the Brussels-Capital Region, the pollution $NO_2$ levels will never be so high that the assumption of $NO_2$ as an optically thin absorber is not fulfilled.

**15. Page 13 line 314: (see Fig. 6 (a))**
**Response:** The sentence has been modified as follows:
**For example, a MAX-DOAS measurement with SZA=30°, RAA=60°, $MLH_{NO2}$=1km, and measured $O_4$ dSCD=$6.10^{43}$ $molec^2.cm^{-5}$ will have a $L_{NO2}$ equal to 15 km at 477 nm (see Fig. 6a).**

**16. Page 16 equation (6) and (7): For readability it would be good to include the missing step between the two equations**
**Response:** The missing step between equations (6) and (7) is now added.

**17. Page 17 equation (7): possible space character in subscript**
**Response:** Equation (7) has been checked and no space character was found in subscript.

**18. Page 22 equation (11), middle line: Confusing definition. If written like this, it should be x(first) >L_NO2 (the first value which extends over L_NO2).**
**Response:** In Equation 11 (now equation 14), as x(last), we are referring to the last point of the horizontal grid. To be more clear, we named x(last) as x(max).

**19. Page 23 line 536: If this is true, why do the lines differ in figure 13 and 14 for the first km from the a priori? And why is the information of this region used in the later comparisons if this information cannot be trusted?**
**Response:** In this region (extending from the MAX-DOAS instrument until the location of the $L_{NO2}$ estimated from the 344nm measurements), the information that we obtain from the MAX-DOAS measurements is a constant NO2 near-surface concentration. The role of the a priori profile is to give this region the shape of the NO2 horizontal profile, as shown in Figure 13. In our under-constrained problem, the a priori information is derived by an air-quality model.

**20. Page 25 figure 11: Add wavelengths and change colors (see https://www.color-blindness.com/coblis-color-blindness-simulator/). Add also time of the measurement.**
**Response:** Figure 11 has been modified according to this comment and the measurement time has been added to figure's caption as follows:

**Examples of weighting functions used in the new horizontal distribution inversion approach (11 September 2018, 11:26 UTC).**

21. **Page 26 figure 12: Same y-axis for all**
Response: The y-axis is now the same for all the subplots in Figure 12.

22. **Page 27 figure 12: Mention RIO in caption**
Response: The legend of Figure 12 has been modified as follows:

**"Figure 12. Example of seasonal RIO *a priori* $NO_2$ horizontal profiles for the new horizontal distribution inversion approach as a function of the horizontal distance from the MAX-DOAS instrument in six different azimuthal viewing directions, before the application of the scaling factor."**

23. **Page 27 figure 13: Adjust both axes to the same values, add (a) and (b)**
Response: Figure 13 has been modified according to a comment of Reviewer #2. Three different example days are displayed for low, medium and high $NO_2$ concentration values. We have added (a), (b), and (c) for the three different examples.

24. **Page 28 figure 14: Add a priori of aerosol distribution. Are errors available?**
Response: Similarly as in Figure 13, three examples of near-surface aerosol extinction coefficient profiles have been added. Additionally, the a priori near-surface aerosol extinction coefficient profiles and the retrieval error have been added in Figure 13.

25. **Page 32 figure 16: For consistency it would be good to see the values corresponding to figure 13**
Response: Figure 16 has been modified and the errors refer to the example used in Figure 13b.

26. **Page 33 figure 17: It is claimed before, that the measurement is insensitive for the first kilometers. It would be good to mark this range (e.g. draw a line). Why is the y-axis limited to 11km?**
Response: It is true that maximum and minimum $NO_2$ values cannot come from the MAX-DOAS measurements in this region. This horizontal region of insensitivity depends on the $L_{NO2}$ of the first MAX-DOAS measurement at the smaller wavelength, and consequently, is different for each retrieved horizontal profile. We have decided to not add this information in Figure 17 as it will become difficult to interpret. Concerning the y-axis, we have extended the range of this axis until the maximum $L_{NO2}$ estimated during that day.

27. **Page 33 line 664: Is the wind direction and speed stable over the measurement period? What is the wind speed?**
Response: As you can see in Figure 18, the wind direction is almost stable over the measurement period and its main directions are NE, E, and SE during that day. The wind

speed is higher during the morning hours (3.7 m/s for 5 UTC) and lower around 10 UTC (i.e., 2.2 m/s).

28. **Page 34 figure 18: The wind direction could be added to the plots. If possible, the latitude and longitude range could be shortened to make the plot bigger.**
**Response:** The wind direction as measured by the meteorological station in the rooftop of BIRA-IASB has been added to each subplot. Additionally, the subplots are more zoomed.

29. **Page 35 line 695: Mention typical overpass time of TROPOMI. Is every measurement from TROPOMI used? Even when several are available for one day? And is the exact time of the overpass used or the typical overpass time? The NO2 distribution can change quickly. Were also other time ranges used for comparison? (+/-30Min)?**
**Response:** We have added the following sentence to mention the overpass time of TROPOMI during this day:
**During this day, the TROPOMI overpass time was at 12:19 UTC.**
For other days, TROPOMI can have two overpasses above Brussels-Capital Region. When this is the case, the exact time of each overpass is used and accordingly, the MAX-DOAS horizontal profiles that are retrieved +/- 1 hour of each overpass are used.
During the 28th of June 2019, we performed the same comparison as in Section 5.2, by selecting a smaller time range (+/-30min) for the MAX-DOAS horizontal profiles and by applying a weighting (following comment). The results are the following:

[Figure]

As the comparison points diminished, we have decided to perform the comparisons between MAX-DOAS and TROPOMI observations around 1 hour of each TROPOMI overpass time.

30. **Page 35 line 698: How does this simplification (no weighting) influence the results?**
    **Response:** In this revised version of the manuscript, we have weighted the MAX-DOAS segments by their relative length inside each TROPOMI pixel. As you can see in Figure 20 of Section 5.2, the resulting correlation coefficient, slope and intercept change slightly.

31. **Page 35 line 706: Can this be explained/expected? (Different sensitivity…)**
    **Response:** Given the fact that with the dual-scan MAX-DOAS parameterization technique, we are sensitive to the near-surface $NO_2$ layer, it is possible that during that day, the APEX (airborne) instrument detected an additional $NO_2$ layer at an altitude higher than the estimated $MLH_{NO2}$ (which is around 700m during the APEX flights).
    Another possible explanation could be the different measurement techniques of both instruments (i.e., ground-based and airborne) and consequently, different instrument sensitivities as well as the different reference spectrum used in the DOAS analysis of measured spectra.

32. **Page 36 line 723: Are the retrieved values high/low, compared to typical values for this region?**
    **Response:** For the Brussels-Capital Region, the retrieved values during that day are low/medium. As we can see in Figure 23, the retrieved $NO_2$ tropospheric columns can be up to $2.5*10^{16}$ molec/cm$^2$.

33. **Page 37 figure 19: Add MAX-DOAS maps in original resolution for comparison with original APEX. Also, the RIO analysis could be added. Use the same color scale for all plots!**
    **Response:** In Figure 19, the MAX-DOAS map in original resolution has been added. Following a comment from Reviewer #2, the color-scale is different for the original APEX and AEROMOBIL measurements and the same for all the other subplots. A comparison with RIO dataset is presented in Section S1.

34. **Page 38 line 759: Point number 3 is unclear to me. A sketch could help to clarify.**
    **Response:** Point number 3 is indeed hard to be interpreted by the reader. In addition to that, an application of such a weight in the mean MAX-DOAS $NO_2$ columns does not change the comparison outcome. For these reasons, point number 3 has been modified as follows:
    **The MAX-DOAS segments are weighted by their relative length inside each TROPOMI pixel.**

    Consequently, the MAX-DOAS and TROPOMI comparisons presented in Section 5.3 have been modified.

35. **Page 39 line 768: Regridding routine is not completely clear to me. Are the averaged pixels weighted by their ratio by which they cover the reference pixel?**
    **Response:** To create the seasonal maps for MAX-DOAS and TROPOMI datasets, we have chosen the TROPOMI pixels (i.e., reference pixels) recorded during the APEX flights (28 June 2019; see figure below). For every measurement day, the TROPOMI pixels and MAX-DOAS grids that fell inside each reference pixel have been averaged without applying a weight equal by the ratio by which they cover each reference pixel.

The reference pixels are the following:

[Figure]

36. **Page 39 line 776: Agreement for summer seems to be worse than for other seasons. The maximum in TROPOMI is seen in the NW, in MAXDOAS it is south.**
**Response:** Indeed, the agreement for summer is worse than for other seasons. We have modified the text as follows:
**It is found that the locations of the NO₂ peaks and dips show a reasonably high degree of similarity between TROPOMI and MAX-DOAS during all seasons, except summer.**

37. **Page 39 line 782: Can this be explained?**
**Response:** As we can see in Figure 22c, large positive biases does not occur for our study. Although, the most common case (in each pixel) is to have a negative bias, which means that TROPOMI underestimates the tropospheric MAX-DOAS NO₂ columns. This finding is in agreement with several studies (Verhoelst et al., 2021; Tack et al., 2021; Judd et al., 2020; Dimitropoulou et al., 2020; Ialongo et al., 2019).

38. **Page 39 line 792: Explain SEAS, I guess season?**
**Response:** The sentence has been slightly modified as follows:
**When seasonally-averaged TROPOMI and MAX-DOAS pixels (the pixels shown in Fig. 21) are compared one-by-one (see seasonal (SEAS) in Fig. 23), both correlation coefficient (R in the range of 0.57-0.93) and slope values (s in the range of 0.65-0.94) improve considerably.**

39. **Page 40 line 806: "… presented in Dimitropoulou et al. (2020), see table 4."**
**Response:** The sentence has been modified.

40. **Page 41 figure 21: If possible, remove one color bar and use the free space to widen the plots in x-axis.**
**Response:** Concerning Figure 21, one color bar per season has been removed and we have zoomed more in the subplots.

41. **Page 44 figure 23: Red line is hardly visible**

**Response:** In Figure 23 and Figure S11, we have extended the red line to cover the whole range of tropospheric $NO_2$ columns.

42. **Page 45 table 4: Clarify R (seasonal) and s (seasonal). It probably corresponds to SEAS? Are the number of points the number of days or the number of pixels?**
    **Response:** In Table 4, the legend has been modified as follows:

    **Table 4. Summary of the regression analysis parameters (e.g., correlation coefficient (R) and slope (s)) and the number of data points (N) derived in the present study during only one year of observations (i.e., number of pixels) and in Dimitropoulou et al. (2020). Please note that R (seasonal) and s (seasonal) corresponds to SEAS in Figure 23.**

43. **Page 45 line 885: Add name of the improved FRESCO-S cloud retrieval.**
    **Response:** As mentioned above, the TROPOMI tropospheric $NO_2$ column product with an improved FRESCO-S cloud retrieval is called DDSv2. We have added:
    **First, a TROPOMI tropospheric $NO_2$ column product (DDSv2 product) with an improved FRESCO-S cloud retrieval was tested.**

44. **Page 45 line 887: "Fig. 24 (a) and (b) …"**
    **Response:** (a) and (b) have been added to the text.

45. **Page 46 line 910: Using MAX-DOAS profiles as a priori profiles for TROPOMI is suitable for a consistency check of the method. It should not be mistaken to be the "truth".**
    **Response:** The sentence has been modified as follows:
    **The present study suggests that in urban conditions, the $NO_2$ profile shapes from the CAMS regional CTM ensemble are not the most suitable a priori information that can be applied in the TROPOMI retrieval.**

46. **Supplement figure S1: Same y-axis for all plots, MLH could be in the title of the individual sub-plots.**
    **Response:** Figure S1 and S2 have been corrected according to this comment.

47. **Supplement figure S3: Same color bar for all plots**
    **Response:** Figure S3 has been modified accordingly.

48. **Supplement figure S4: Same y-axis, at least for AOD>0, this highlights the differences much stronger**
    **Response:** In Figure S4, same y-axis has been used in all the subplots, except for AOD equal to zero.

---

## Author Comment (AC2)

**Response to Anonymous Referee #2**

We want to thank the referee for the constructive examination of our paper entitled "Horizontal distribution of tropospheric $NO_2$ and aerosols derived by dual-scan multi-wavelength MAX-DOAS measurements in Uccle, Belgium".

**Please find below our responses to each comment individually.**

**Please consider that:**

**A) Green bold: Comments of the Referee**

B) Black: The response to each comment posed by the referee.

**C) Black bold: Already existing text in the manuscript.**

**D) Red bold: Added text in the manuscript according to referee's comments.**

**General comments**

**A. Dimitropoulou et al. 2021 present an interesting and new way to retrieve horizontal trace gas and aerosol profiles from MAX-DOAS data measured at various geometries and wavelengths. The approach utilizes data from profiling algorithms, further RTM calculations, and makes several assumptions on the atmosphere and the spatial distribution of its absorbers.**

**I recommend the publication of this manuscript after addressing some critical comments, because of the interesting idea rather than the convincing approach/results.**

**Before I explain further on the issues below, please let me give you my personal opinion on the approach and its assumptions. You start with the calculation of vertical profiles of MAX-DOAS measurements and calculate MLH. Even though validated, a MAX-DOAS profiling algorithm is not the truth but a smoothed representation of the true atmosphere constrained by the limited vertical resolution. The calculation of MLH is again an assumption of a box like distribution even though it was shown with MMF before that the profile was most likely not box-like at all. Then, further assumptions about a homogeneity of the MLH, calculated effective light path lengths and AODs are made (e.g. aerosol parametrization, profile shapes with percentages of bulk load based on statistics, fix values of all input parameters). The estimation of VMR based on Sinreich et al. 2013 is again just an approximation which is only valid under certain conditions. And then, everything is used in another inversion step. Knowing all this, it appears to be a miracle that your correlations are still good! To remove some of my doubts, please assess the error budget and the propagation of errors between the individual steps in more detail.**

**Response:** First, it is important to address this general comment concerning the assumptions made in the different steps of the proposed retrieval methodology.

The retrieval of profile information from MAX-DOAS measurements is typically made of several assumptions (i.e., the a priori profiles, the layering of the atmosphere, aerosol properties and other retrieval parameters). Given the fact that the retrieval problem is under constrained, these assumptions cannot be avoided.

To validate these retrievals, which is a crucial step, we perform validation studies based on available independent correlative datasets at the measurement site.

For the present study, we would like to underline that the application of the parameterization technique of Sinreich et al. 2013 has been widely used in the literature for pollution conditions similar to those encountered in the Brussels-Capital Region. In Dimitropoulou et al. (2020), we have validated the retrieved MAX-DOAS near-surface concentrations of $NO_2$ by performing a comparison with the in-situ stations located in the Brussels-Capital Region. For this reason, we have not performed again a similar comparison.

Additionally, the error budget is now estimated in more detail compared to the initial version of the manuscript. For instance, concerning the estimated error of the new OE-based inversion approach, we present the smoothing error in addition to the measurement error (see Figure 16).

The new Horizontal Distribution inversion approach has been validated by using independent remote sensing measurements (i.e., airborne and car-mobile DOAS) during one day of observations. Additionally, to this one-day comparison, we have added a comparison concerning the year 2018 between RIO and MAX-DOAS near-surface $NO_2$ concentrations in the Brussels-Capital Region (see Section S1). The main reason of not including this comparison in the principal manuscript is that the a priori NO2 horizontal profiles are estimated with the aid of the RIO model. Consequently, the RIO model is not a completely independent dataset to perform a proper validation.

In a future study, it is important to apply this novel approach in different measurement sites, in which more independent correlative datasets are available than those used in the present work.

**B. Three further issues of this manuscript need to be solved before final publication and will be addressed in detailed below: 1. This manuscript is much too long. 2. The presented approach is not validated sufficiently. 3. The explanation of the approach in Section 4.2.1 needs a revision.**

**Response:** Please consider the provided answers and modifications to our manuscript according to each of your specific comments below.

**Details on the three main issues:**

1. **Nowadays, many paper show more and more results which are sometimes extremely insignificant or do not match the purpose of the publication. The main purpose of this manuscript is the introduction of a novel approach about the retrieval of horizontal absorber concentrations. A manuscript with this topic should introduce and explain the novel approach and show a thoroughly performed validation study. In this manuscript, you also address the question of how to optimize and validate Tropomi measurements. This has nothing to do with the content of your paper and is addressed in more detail elsewhere. So please remove Section 5.3.2 fully (or move it to the supplement) and remove parts of section 5.3.1! E.g. the three concluding points on page 40 can be removed as they are already content of the conclusions. When you add a proper validation (see point below), please move Section 4.2.3 to the supplement.**

**Response:** Following the above-mentioned recommendations, significant modifications are made to the present manuscript. First, we have added a comparison between the RIO air-quality model and the MAX-DOAS horizontal profiles (Section S1). Secondly, in Section 5.3, we present the comparison between TROPOMI and MAX-DOAS retrievals and the previously numbered Section 5.3.2 has been moved to the Supplement as Section S2. Finally, Section 4.2.3 has not been removed from the manuscript because it concerns a verification of the dual-scan MAX-DOAS retrieval method and not a validation.

2. **Validation of a novel approach is a necessary step but needs independent measurements! Validation can never be done with the same instrument. Since you already mentioned the in-situ air quality network in Brussels, I would recommend the use of this data or another independent data set. However, no validation of other instruments/data sets by your novel approach can be shown without first validating itself! This means of course that validation of Tropomi data with your approach is not appropriate. It would make more sense to validate/verify your approach with an already validated Tropomi data set.**

**Response:** We agree that a validation of a novel approach is necessary and crucial. For this reason, we have performed a comparison between the air-quality model dataset and the horizontal profiles derived by the new OE-based inversion approach. This comparison can be found in Section S1 of the manuscript, in which hourly $NO_2$ concentration maps from the RIO model are compared with hourly averaged $NO_2$ horizontal profiles in the Brussels-Capital Region. However, the comparison between profiles retrieved using the standard OEM approach and the parameterization results (see Section 4.2.3) is an important sanity check for the consolidation and verification of the dual-scan MAX-DOAS retrieval method presented in the present manuscript. It should be noted again that this comparison is not a real validation since the two datasets are not independent since we use the information about the MLHNO2 in the dual-scan MAX-DOAS retrieval method.

3. **It was difficult to understand your explanation in section 4.2.1 because it is not directly clear for the reader which quantities are fitted how and when. Please revise this section and explain specifically which measured or inverted quantity goes in which polynomial-fitting or RTM-calculation step. Maybe a small flowchart would help or adapt and refer to the existing flowchart in Fig. 3. Furthermore, Fig. 4 and Fig. 5 show simulated columns. Please add also an example with real measurements and the polynomial fit in the same figures.**

**Response:** In Section 4.2.1, we have added additional information, which can help the reader understand the developed dual-scan MAX-DOAS retrieval method. The additional changes are the following:

1. Here, $NO_2$ dSCDs and consequently $L_{NO2}$ are simulated using the radiative transfer model VLIDORT version 2.7 (Spurr, 2006).
2. The $MLH_{NO2}$ is a known parameter and it is estimated per measurement scan, as the ratio of $VCD_{NO2}$ to the $NO_2$ near-surface concentration as retrieved in the main azimuthal direction by the MMF inversion algorithm.
3. For the six different wavelengths (343 nm, 360 nm, 380 nm, 447 nm, 477 nm, and 530 nm), we separately perform RTM simulations and $L_{NO2}$ (see Eq. 5) are simulated for the assumed SZA, RAA, $MLH_{NO2}$, $c_{NO2}$, and AOD input scenarios presented in Table 2.

4. **The simulated O$_4$ dSCDs** are a function of the input parameter AOD. The relation between the simulated O$_4$ dSCDs and the input AOD values is shown in Fig. 4a.
5. **Additionally, in Fig.4b, we can see that the relation between the O$_4$ dSCDs and the AOD values is valid for MAX-DOAS measurements.**
6. **We observe, also, in Fig. 5b, in which an example day of MAX-DOAS measurements is presented, that the L$_{NO2}$ as a function of the measured O$_4$ dSCDs have the same relation as the simulated quantities.**
7. **Since NO$_2$ is an optically thin absorber, L$_{NO2}$ is not a function of c$_{NO2}$ and consequently, a L$_{NO2}$ value can be estimated** by using the measured O$_4$ dSCD for each measurement.
8. **Based on this approach, the near-surface NO$_2$ concentration can be calculated at the six different wavelengths** by using the measured dSCD$_{NO2}$ together with the simulated L$_{NO2}$ value (Eq. 2).

Additionally, in Figures 4 and 5, an example with real measurements and their polynomial fit is included as well.

**Specific comments**

**1. P3, L88: Please write "telescope azimuth angle" so that the abbreviation (TAA) makes more sense.**

**Response:** The text has been modified according to this comment:

… **(1) a vertical scan in nine different elevation angles (EAs) in one fixed telescope azimuth angle (TAA; Northeast direction i.e., towards the city center and the national airport)** …

**2. P3, L89: Why did you use 2° instead of 1°? (see also P9, L219)**

**Response:** The primary motivation of using the elevation angle of 2° instead of 1° is to avoid obstacles across the line of sight of the different azimuthal directions used in the present study. According to Sinreich et al. (2013), the parameterization technique is valid for low elevation angles. In their study, they have tested the elevation angles of 1° and 3°. In Ortega et al.'s (2015) study, the parameterization technique is also applied for an elevation angle of 2°.

**3. P3, L92: Remove "directions". It is not needed.**

**Response:** The word "directions" has been removed from this sentence:

**The selection of more azimuthal directions towards the North, Northeast, and Northwest  was made considering the location of the main NO$_2$ emission sources and, consequently, the highly variable NO$_2$ horizontal distribution towards these directions.**

**4. P3, L94: Integration time of 60s? Why was it set to such a long time? Depending on the wind speed, many things can happen within one minute in an area with that strong spatial inhomogeneities!**

**Response:** By changing the experimental set-up of the MAX-DOAS instrument in Uccle from one to multiple azimuthal directions, our aim was to study the horizontal distribution of aerosols, $NO_2$, and HCHO. The selected experimental set-up is a trade-off between acceptable S/N ratio and having a maximum number of azimuthal and elevation directions.

HCHO is a weak absorber and for this reason, we had to adopt an acquisition time of the spectrum large enough to ensure that the measurements are not too noisy. HCHO results are not presented in this study but you can find the study in the following link:
https://ieeexplore.ieee.org/abstract/document/9553326

**5. Figure 1: When I remember correctly, you will never again talk about the individual tests shown in this plot. So please remove these tests and show only the applied azimuthal directions. Please zoom slightly in and show the power plants, ring freeway, and MAX-DOAS site, similar to Fig. 18.**

**Response:** The period we have used for the present study extends from March 2018 to February 2020. Consequently, different experimental set-ups are exploited during this study and we think it is important to make it clear to the reader. For this reason, we have decided to show the individual experimental set-ups in Figure 1. Additionally, we have modified this figure to show the main $NO_2$ emitters in the Brussels-Capital Region and the figure's caption accordingly.

**Figure 1. The experimental set-up of the BIRA-IASB dual-scan MAX-DOAS instrument. Each line is color-coded according to the different set-ups that were used from March 2018 to February 2020. The length of each line is equal to 20 km, which corresponds to the typical horizontal sensitivity for the MAX-DOAS measurements in the present study (see Fig. 18). The black square shows the MAX-DOAS instrument location, the black polygon the National Airport, the black dots the $NO_2$ hotspots emitting more than 10 kg of $NO_x$ per hour (Emission Inventory of the Belgian Interregional Environment Agency, 2017) , and the black line represents the Brussels Ring motorway.**

**6. Section 2.2: Please explain how you decided on these wavelength intervals. Did you do some optimization for the shown fit settings? For example, I was wondering why the window 510-540nm was chosen like that even though large $H_2O$ absorption is present at the start and end wavelengths. These absorption features might also explain slightly larger residual structures (compare Fig. 6).**

**Response:** The main criterion of deciding these wavelength intervals was to include, in each of them, one of the main maximum $O_4$ absorption bands that are available in the UV-Visible wavelength range. The choice of the optimal wavelength interval to retrieve the trace gas of interest is a compromise between maximizing the sensitivity of the trace gas and minimizing interference with other absorbers and the presence of residual structures in the DOAS fit results.
Concerning the last fitting window (510-540 nm), $H_2O$ is included in our fit.

**7. P5, L116: Why are not all reference wavelengths in the peak center of their corresponding $O_4$ absorption bands?**

**Response:** The reference wavelength, in each chosen fitting window, corresponds to the maximum of $O_4$ absorption peak or close to this maximum. In some windows, the reference wavelength is not exactly at the maximum because a better fit in terms of RMS is achieved.

**8. P5, L123-L125: You said that the $O_4$ cross section of Finkenzeller improves results in the UV. Does it show any change in the visible range? Why did you decide against using it here as well?**

**Response:** According to the presentation of H. Finkenzeller at the DOAS workshop in 13 July 2020, the main difference between the $O_4$ cross section of his study and the previously published $O_4$ cross section (Greenblatt et al., 1990; Thalman & Volkamer 2013; Hermans et al 2011) in the wavelength range of 308-500 nm is the transition at 344nm. More precisely, in the study of H. Finkenzeller, the O4 cross section values before and after the peak at 344 nm are between the O4 cross section values of Greenbalt et al., (1990) and Thalman & Volkamer (2013). Following this finding, we have decided to use the $O_4$ cross section of Finkenzeller only in the UV wavelength region.

**9. P7, L172: "measured radiance spectra ... is analyzed" to "are analyzed"**

**Response:** Corrected.

**10. P9, L195: Please cite some of the "several studies" you are referring to.**

**Response:** We have added three related studies. The interested reader is now referred to Table 1 of Wagner et al. (2019) for more studies.

**11. P9, L196: I would write of a scaling factor ≠ 1 because some studies suggest also larger scaling factors depending on spectral range, location and season.**

**Response:** The sentence has been corrected.

**12. P9, L208-L209: Why do you accept retrievals with homogeneous cloud coverage? The corresponding aerosol profile is wrong for sure! This means that your MLH is inaccurate because is is negatively affected by the wrong radiative transfer. All your RTM calculations of $L_{NO_2}$ are wrong as well and, therefore, your horizontal profiles!**

**Response:** The effects of the clouds on trace gas retrieval are very important when broken clouds are present because during an elevation scan, some measurements are influenced by clouds while some others are not depending on the elevation angle. Under homogeneous cloud cover, the quality of MAX-DOAS are much less affected by clouds (see e.g. Gielen et al., 2004 and Wagner et al., 2015). For this reason, we have chosen to only filter out elevation scans that are measured under broken cloud conditions.

**13. P9, L219: Why did you select an elevation angle of 2°? I would assume that 1°? is better suited for your purpose and assumptions about homogeneity are closer to the truth. (see also P3, L89)**

**Response:** This comment has been addressed in comment no. 2.

**14. P10, L251: Please move the full stop from the index of NO₂ to the normal level.**

**Response:** The text has been modified accordingly.

**15. P11, L276: "Here, O4...". If "here" refers to the equation above it should be "Here, NO2..."**

**Response:** The sentence has been modified as follows:

**Here, NO₂ dSCDs and consequently, L$_{NO2}$ are simulated using the radiative transfer model VLIDORT version 2.7 (Spurr, 2006).**

**16. Section 4.2.1 Please revise according to the general comment.**

**Response:** Following the general comment, we have revised Section 4.2.1, but it should be noted that it is still included in the main manuscript.

**17. P13, L324: "As discussed above, around 30%...". I would not call this a real discussion. You just mentioned it without showing any results of the analysis.**

**Response:** The sentence has been modified as follows:

**As mentioned above, around 30% of the total aerosols is expected to be found inside this layer.**

**18. Fig. S4, S5: Please discuss the dependence on SZA and RAA for Fig. S1 and S2, respectively, as well. I would assume that a significant contribution of the dependence is due to aerosols and the applied phase-function.**

**Response:** The dependence on SZA and RAA for Figures S1 and S2 has already been discussed in Dimitropoulou et al. (2020). The difference is that we present the L$_{NO2}$ directly in the present manuscript. Only the correction factors were presented in Dimitropoulou et al. (2020) (Sinreich et al., 2013).

More precisely, for small RAA, a large correction factor value is found, which means small values of L$_{NO2}$. L$_{NO2}$ increases with the RAA. The dependency of L$_{NO2}$ with SZA is linear until SZA reaches the value of 45°, where it increases rapidly. This result, together with the correction factor values for SZA values in the range of 45°-80° (Figure 2 in Dimitropoulou et al., 2020), are indicators concerning the limitations of the parameterization technique for low sun conditions.

**19. P13, L335: "less pronounced Rayleigh scattering" to "less pronounced Rayleigh and Mie scattering"**

**Response:** Corrected.

**20. Fig. 6: Please explain why the smaller deviations of L from the polynomial fit propagate into much larger deviations for the near surface concentration and vertical column density.**

**Response:** In Figure 6, we can see that the near-surface $NO_2$ concentrations at 360nm and 380 nm deviate more from the polynomial fit than the other points at larger wavelengths. Consequently, the $L_{NO2}$ at 360nm and 380nm (Figure 6a) are the points that deviate the most from the polynomial fit.

The error related to these near-surface $NO_2$ concentrations reflects this deviation.

**21. Eq 6 and 7: The step from Eq. 6 to 7 can not happen without further assumptions. I would rather prefer the calculation of the partial derivative than this assumption. Please explain this step in more detail. How is the uncertainty of the $O_4$ dSCD used?**

**Response:** Between Equations 6 and 7, we have included the intermediate step for the calculations of the error.

The uncertainty of the $O_4$ dSCD is not included in the error calculation of this study. According to Dimitropoulou et al. (2020), the $O_4$ DOAS fit error is up to 5 % and 6 % in the visible and UV range, respectively.

As first error source, we take into account the uncertainty related to the $NO_2$ dSCD. As second error source, we include the uncertainty related to the estimation of the $L_{NO2}$ from the RTM simulations. As third error source, we include the error related to the $MLH_{NO2}$ estimation (see Dimitropoulou et al., 2020).

The error source related to the $MLH_{NO2}$ estimation is included only in the error estimation of the near-surface $NO_2$ concentration and not in the $VCD_{NO2}$ because it would be accounted twice and conduct to a falsely larger error budget for the $VCD_{NO2}$.

Please find the modifications in Section 4.2.2.

**22. Section 4.2.3: This section needs to move to the supplement and has to be replaced by a real validation. The sanity check is interesting but should not be content of the main manuscript. The validation part is unfortunately no validation but rather a verification. Validation only works with independent measurements. Using just a different azimuthal direction from the same instrument is not at all independent. For a study like this, I would assume comparison with e.g. in-situ instruments. Please add a validation study from an independent instrument as e.g. in-situ data from the air quality network in Brussels.**

**Response:** Section 4.2.3 has been named as: **Verification of the dual-scan MAX-DOAS retrieval method**

The present parameterization technique is an updated form of the parameterization technique used in Dimitropoulou et al. (2020), in which a comparison between MAX-DOAS and in-situ observations has been already performed. For this reason, in the present manuscript, we did not perform a comparison with the in-situ air quality network in the Brussels-Capital Region.

Additionally, in the revised manuscript, we now perform a comparison between the RIO air quality model dataset and the MAX-DOAS $NO_2$ horizontal profiles over one year of observations (now, Section S1).

Please consider our response about this section in the general comments.

**23. Eq 11: Please give more information on this approximation/definition. Is the weighting function for aerosol extinction coefficients defined in a similar way? With this definition, you suppress variability in the horizontal direction which means that you consider your effective light path lengths as perfect. I also have a problem with the fact that the weighting itself was arbitrarily defined as horizontal step width divided by whatever is found from your simulation of L. Is the information content not large enough to allow a more flexible implementation? Please discuss this further.**

**Response:** First, the weighting functions for the aerosol extinction coefficients are defined in a similar way as for $NO_2$. Concerning the approximation/definition of the weighting functions, we have used a simple horizontal box model in which we assume a geometric approximation where the sensitivity along the horizontal distance per measurement is constant. The comparisons performed during the 28[th] of June 2019 show that the $NO_2$ horizontal profiles are in good agreement with ancillary data. Additionally, the comparison between the MAX-DOAS $NO_2$ horizontal profiles and the RIO model data show a good agreement, too. These findings indicate that the approximation concerning the weighting functions is reasonable and lead to realistic results.
In a future study, the calculation of more appropriate weighting functions, considering the horizontal variability of the light intensity along the horizontal dimension could be investigated.

**24. P22, L502: How large is this mean scaling factor? You already add a bias here by applying a mean factor. It would be interesting to know if the unscaled a priori profiles would lead to a better agreement with ancillary instruments or if it just destabilizes your retrieval.**

**Response:** As mean scaling factor, we are referring to the mean ratio between the six MAX-DOAS measurements (i.e., $NO_2$ near-surface concentrations as retrieved in the six different wavelengths) and the unscaled a priori profile in every measurement direction. Taking the example of 28[th] of June 2019, the scaling factor are within a range of 0.27 – 0.56. This finding is in agreement with several studies in which comparisons between remote sensing and in-situ instruments are performed (see e.g. Section 4.2 of Dimitropoulou et al., 2020).

If we do not apply these scaling factors, the comparisons between MAX-DOAS and airborne, and MAX-DOAS and TROPOMI for June 28, 2019 are the following:

[Figure]

As you can see, if we don't apply any scaling factor, a worse agreement is found between the MAX-DOAS horizontal profiles and the ancillary data.

**25. Fig. 13 and 14: Please add two further examples for profiles with larger concentrations and examples with a small aerosol/NO$_2$ load. The readability should not suffer with two more curves in these plots. Please add the a priori profiles and errorbars for the aerosol horizontal profiles as well. In Fig. 14, you see a larger deviation of measured and simulated extinctions for the middle L$_{O\_4}$ values as well as for the first data point. Since the reader cannot see the a priori, it is hard to assess how this propagates into your retrieved profile. Please discuss this deviation together with the a priori profile and the retrieval errors.**

**Response:** In Figures 13 and 14, we have added two additional examples with profiles that have larger and smaller NO$_2$ values and aerosol load, respectively. Additionally, a discussion concerning these two figures is included in Section 4.3.

**26. Fig. 15: Since AK are the multiplication of the gain matrix G with your weighting function matrix K, your averaging kernels show the sensitivity based on your definition of the weighting function. Please show for this scenario the corresponding a priori profile and retrieval result. I would assume that the features of your AK matrix are strongly dominated by L$_i$ and your a priori profile. It is difficult to understand why the sensitivity for blue and red are the highest at exactly the same distance (similar with purple and green). If this figure would be a good representation of the derivative of concentration with respect to the true concentration, individual peaks should be found at different distances. Please discuss!**

**Response:** As it is stated in the present comment, the AKs are calculated by multiplying the gain matrix G with the Weighting function matrix. Figure 15 has been modified and is referring now to the a priori profile and retrieval result of Figure 13a.

Indeed, the features of the AK matrix depend strongly on the location of $L_i$ and the a priori profile shape, as we can see in Figure 13a and Figure 15. For the first three sampling grids of Figure 15 (blue, red, yellow), the information is coming mainly by the two first measurements (i.e., two first $L_i$), which can explain the co-located peaks of these sampling grids.

**27. P29, L601:  "close their" to "close to their"**

**Response:** corrected.

**28. P29, L610-L611: Why should this be the case? What about the smoothing error? Since you have many constraints due to your a priori assumptions, I would assume that the smoothing error has a significant contribution to your total error.**

**Response:** Indeed, the smoothing error has a significant contribution to the total error of the retrieval. In order to estimate the smoothing error, we have used RIO horizontal $NO_2$ profiles (after applying a mean scaling factor) to construct a covariance matrix.

The measurement error and the smoothing error are now shown in Figure 16.

**29. Table 3:  "Medium RMS" to "Median RMS". What do the DOFS and RMS values in brackets in the last row mean? If this refers to the total accepted retrievals, why did you write different thresholds in the text? Furthermore, it is hard to assess the range of RMS and DOFS values based on these numbers only. Please add a figure of the frequency distribution of RMS and DOFS values and discuss it together with this table!**

**Response:** In Table 3, we have corrected *medium RMS* to *median RMS*.

Please consider a correction in the last row, which was written by mistake in the first version of the manuscript.

We have added Figure S7 and S8 in the Supplement showing the probability density function (PDF) of RMS and DOFS values for our study.

Additionally, in Section 4.4, the text has been modified as follows:

**To eliminate the unsuccessful retrievals, the percentage of accepted retrievals with respect to the total number of retrievals during the four seasons is investigated when a specific filtering on RMS and DOFS is applied (see Table 3 and Figures S7 and S8). As we can see in Fig. S8, DOFS are in the range of 1.2-2.5. From these tests, it is found that most of the retrievals have DOFS larger than 1.5 (see Fig. S8). RMS is defined as the root-mean-square deviation between measured and simulated $c_{NO2}$ normalized by the mean of the measured $c_{NO2}$ (e.g., same RMS as in Fig.10). Table 3 and Fig. S7 indicate that RMS values are in the range of 0-30% and most of the retrievals have an RMS smaller than 6% with a**

**median RMS value of around 4.5% during all seasons. Based on these investigations, DOFS>1.5 and RMS<6% are used as retrieval quality control criteria.**

**30. Fig. 18: Please zoom in and increase the quality of this figure so that details can better be seen. Please add a similar plot showing the near-surface concentration/extinction in the supplement. Do near-surface values support your finding that air pollution in Brussels is mainly driven by the power plant and traffic emissions? Can larger values be found in the distance of the Ring-motorway?**

**Response:** Figure 18 has been modified according to this comment. Additionally, the wind direction as measured in the meteorological station at the rooftop of BIRA-IASB has been added to each subplot.

In the supplement, you can find two new figures: one for the near-surface $NO_2$ concentration (Figure S5) and one for the near-surface aerosol extinction coefficient (Figure S6).

In Section 5.1, we have added the following sentences:

**Same $NO_2$ horizontal distribution is found when investigating the $NO_2$ near-surface concentrations for this day (see Fig. S5).**

and,

**Maximum near-surface aerosol extinction coefficient values are observed during all day long and detected in the N, NW and NE direction (see Fig. S6). $NO_2$ and aerosol peaks are co-located towards the N and NW direction.**

**31. P35, L698: Why are the segments not weighted? If there is just a tiny fraction of light path within a pixel, it should not be weighted with a similar factor as contributions with much longer light paths. Please change this or discuss why it is not possible/reasonable.**

**Response:** Following your comment, we have weighted the MAX-DOAS segments by their relative length inside each TROPOMI pixel for the results presented in Section 5.2 and Section 5.4.

The text has been modified as follows:

**It should be noted that the MAX-DOAS segments are weighted by their relative length inside each pixel.**

**32. Fig. 19: Please change the range of colors for subfigure a) so that the plume is better visible. For the original data of APEX and the AEROMOBIL I would not assume the same color-scale. However, subplot b), c) and d), should have the same color-scale! Please change this.**

**Response:** Figure 19 has been modified according to this comment.

**33. P35, L706: How do you explain this intercept?**

**Response:** Given the fact that with the dual-scan MAX-DOAS parameterization technique, we are sensitive to the near-surface $NO_2$ layer, it is possible that during this day, the APEX (airborne) instrument detected an additional $NO_2$ layer in an altitude higher than the estimated $MLH_{NO2}$ (which is around 700m).

Another possible explanation could be the different measurement techniques of both instruments (ground-based and airborne) and their different sensitivities as well as the different reference spectrum used in the DOAS analysis of both measured spectra.

**34. P35, L709: "channels" to "elevation angles" or "geometries"**

**Response:** The sentence has been corrected.

**35. Fig. 20b: Please correlate the not averaged data as well by comparing MAX-DOAS and AEROMOBIL data points which are close to each other. 6 data points are not statistically significant enough for such a comparison. Especially not when you can compare data in higher resolution.**

**Response:** We have performed a comparison between the not averaged MAX-DOAS and AEROMOBIL data for 28 June 2019, which is shown in Fig.17b. Moderate correlation coefficient is found between both datasets (R=0.74) and a slope equal to 0.55.

**36. Section 5.2: Please add the exact overpass times you used for the comparison of TROPOMI and add a reference to Fig. 18 in the text so that the reader can also compare the MAX-DOAS data in higher resolution at the overpass times with Fig. 19. Furthermore, add the start and end time for the AEROMOBIL measurement in the text and the caption of Fig. 19.**

**Response:** The exact overpass time used for the comparison of TROPOMI has been added to the text as follows:

**During this day, the TROPOMI overpass time was at 12:19 UTC.**

A reference to Figure 18 has been added so that the reader can compare the MAX-DOAS data at the overpass time with Figure 19 as follows:

**During the TROPOMI overpass (i.e., 12:19 UTC) above the Brussels-Capital Region, dual-scan MAX-DOAS tropospheric $NO_2$ columns are retrieved, as it can be seen in Fig. 18.** **The correlation between TROPOMI and MAX-DOAS tropospheric $NO_2$ columns during the day of the airborne measurements above Brussels is presented in Fig. 20c.**

The start and end time for the AEROMOBIL measurements have been added to the text as follows:

**The AEROMOBIL was used to measure the spatial distribution of tropospheric $NO_2$ columns mainly over the Ring road of Brussels with start measurement time at 8:30 UTC and end time at 15:42 UTC.**

And the caption of Fig.19 had been modified as follows:

**Figure 19. (a) Tropospheric NO₂ VCD as detected by the APEX instrument in its initial spatial resolution. Tropospheric NO₂ VCD maps (TROPOMI pixels) as retrieved over Brussels on 28th of June 2019 by the (b) APEX, (d) MAX-DOAS and (f) TROPOMI (overpass time at 12:19 UTC) instruments. Tropospheric NO₂ VCD as retrieved by the (c) MAX-DOAS and (e) AEROMOBIL (between 8:30 UTC and 15:42 UTC) in its initial spatial resolution. The black square shows the MAX-DOAS instrument location, the black polygon the National Airport, the black dots the NO₂ hotspots emitting more than 10 kg of NOₓ per hour (Emission Inventory of the Belgian Interregional Environment Agency, 2017) , and the black line represents the Brussels Ring road.**

**37. Section 5.3.1: In addition to my general remark about the purpose of this document, I would like to ask you to change this section either to a "validation of the new algorithm" or to a "comparison only" section.**

**Response:** Section 5.3.1 is now Section 5.3 entitled as:

**Comparison of MAX-DOAS horizontal NO2 distribution versus TROPOMI observations**

**38. Fig. 21: By just looking at this plot, I would say the agreement is not that good. Why is the difference in the SW direction so large in Summer and Winter? Why are the values close to the MAX-DOAS site much higher for the MAX-DOAS data in Spring? This is especially interesting because your algorithm was described of having a poor sensitivity close to the instrument (compare P26 L599-L600)!**

**Response:** If we observe Figure 22 and Figure 23 (i.e., seasonal results), one can see that indeed summer and winter show a less good agreement compared to spring and autumn. In Figure 22, in the SW direction, we obtain the highest positive bias (around 9%), and close to the instrument, the lowest bias (around − 36%).

Several studies (cited in the manuscript) have reported the same underestimation of tropospheric NO₂ columns as observed by TROPOMI when compared to ground-based remote sensing ones. Consequently, our findings is in agreement with previous studies.

With the phrase "poor sensitivity close to the instrument", we aim to describe that in the first kilometers (i.e., starting from the instrument until the first measurement), the information of the tropospheric NO₂ columns in the horizontal profiles comes only from the first MAX-DOAS measurement and consequently, the horizontal distribution is highly influenced by the a priori NO₂ profile.

A possible explanation about the spatial distribution of the bias between TROPOMI and MAX-DOAS could be the change of the $MLH_{NO2}$ along each line of sight, which is related to the elevation angle of observation and the horizontal sensitivity.

For our study, several factors, that can be improved in future studies, play an important role in the MAX-DOAS retrieval such as the estimation of the $MLH_{NO2}$ from one azimuthal direction, the non-homogeneously distributed azimuthal directions, and the assumptions of the NO₂ vertical profile as a box profile to estimate the near-surface NO₂ concentration.

**39. P39, L780-L782: Similar to the comment above, I am wondering why there is a large negative Bias close to the instrument and a positive one for the pixels in large distance to the MAX-DOAS site? I was wondering if the values at the contour-legend are correct? Is the positive bias just smaller than 1%? If yes, this bias might be negligible. Please check the values and the figure and give an explanation for theses biases if possible.**

**Response:** Please find our response concerning the bias in the previous comment.

Additionally, Figure 22 has been modified and the contour-legend is now correct with values ranging from -36% to 9%.

**40. P39, L790: I would not talk of an under-/overestimation by one of the instruments if it is not clear which instrument/algorithm shows the more accurate results.**

**Response:** This sentence and the following one have been modified as follows:

**During spring, the slope value is equal to 0.90, while during winter, summer, and autumn, the slope values are smaller (0.64 0.56, and 0.64, respectively). Similar findings have been reported in several studies (Verhoelst et al., 2021; Tack et al., 2021; Judd et al., 2020; Dimitropoulou et al., 2020; Ialongo et al., 2019).**

**41. P40 L818-828: This belongs to the conclusion and should be removed from this section.**

**Response:** This part has been removed from this Section and has been placed to **Section 6: Conclusions**.

**42. Section 5.3.2: Remove this section from the manuscript or move it to the supplement (see general comments).**

**Response:** Section 5.3.2 has been removed from the main manuscript and placed to the Supplement as Section S2.

**43. P49, L953-954: and AODs**

**Response:** The sentence has been modified according to this comment.

**44. Conclusions: Please change the conclusions according to the general and specific comments.**

**Response:** Section 6, which gives the conclusions of the present study has been modified according to the general and specific comments.

**References**

**Sinreich, R., Merten, A., Molina, L., and Volkamer, R.: Parameterizing radiative transfer to convert MAX-DOAS dSCDs into near-surface box-averaged mixing ratios, Atmos. Meas. Tech., 6, 1521–1532, doi:10.5194/amt-6-1521-2013, 2013.**

---

## Referee Report (RR1)

**General comments**

The manuscript entitled 'Horizontal distribution of tropospheric NO2 and aerosols derived by dual-scan multi-wavelength MAX-DOAS measurements in Uccle, Belgium' by Dimitropoulou et al. describes a novel method for the retrieval of information on the horizontal distribution of trace gases and aerosols from MAX-DOAS measurements by exploiting the wavelength dependence of the light path length of scattered sunlight. I was asked by the Associate Editor after the open discussion phase to review the paper and these are my first comments to the manuscript. In the following, line and page numbers are referring to the manuscript version with tracked changes (amt-2021-308-ATC1.pdf).

So far, retrievals of information on the spatial distribution of trace gases and aerosols from MAX-DOAS measurements were mostly restricted to the vertical dimension. Based on the wavelength dependence of the light path through the atmosphere, the novel method proposed here enables to gain also information on the horizontal distribution, which is a great benefit, in particular if the spatial distribution of trace gases is highly variable, such as traffic emissions in urban areas. Therefore, the manuscript fits well in the scope of AMT.

A main problem of the manuscript is that the description of the methods is lacking conciseness, and it is not always clear whether certain approaches are based on physical principles or rather represent simplifying assumptions. In particular, the description of the horizontal distribution inversion approach (Section 4.3.) conveys the impression that a constant weighting function as a function of distance with a sharp drop at the estimated light path length is a fact, while this is in actually a very simplifying and physically incorrect assumption. In reality, I would rather expect that the sensitivity decreases exponentially with distance from the observer in accordance with the Beer-Lambert law, as already discussed by Kern et al. [2010] and Vogel et al. [2011]. In the light of these (over-) simplifications, it is surprising to see that the retrieved horizontal distribution compares very well with airborne and satellite measurements.

I wonder if it is not possible to infer the correction factor discussed in Section 4.2 directly from the aerosol and NO2 profiles retrieved by MMF, which directly provide dSCD(NO2) and c(NO2) and thus L(NO2) as the ratio of both.

Some of the measurements used for intercomparison are not explained, or only later in the manuscript. For example, it is not clear what CIMEL measurements are, and airborne measurements are mentioned already in Section 4.3, but the context is only given much later in the manuscript in Section 5.2. I suggest to add a short Section (4) with a short description of the ancillary data used for intercomparison (CIMEL, APEX, car DOAS, etc.).

The level of agreement between the horizontal distributions of NO2 from MAX-DOAS on the one hand and from satellite, car- and airborne measurements on the other hand presented in Section 5.2. is quite impressive. However, I recommend a major revision of the manuscript due to significant deficiencies in the methodology and in the description of the inversion approach. The inversions of the horizontal distribution need to be re-done using physically correct weighting functions.

**Specific Comments**

Section 4.3: I find the description of the forward model quite confusing, and I feel that this Section requires substantial revision. Moreover, I think the inversion needs to be re-done with appropriate weighting functions. Equation 12 suggests that the forward model for the calculation of a mean concentration is given as the integral over the concentration divided by the light path length. First of all, I guess that the model is not based on numerical integration, but that it is rather based on a discrete sum over the horizontal grid. Second, what is missing in this equation is an appropriate weight as provided by a weighting function that represent correct physics. The assumption of a constant sensitivity between the instrument and the effective light path length L is not realistic. Instead, an exponential decrease should be chosen as weighting function in accordance with the Beer-Lambert law. Also, the light path through the boxes is not horizontal but slanted, so the weighting function needs to contain the cosine of the elevation angle (this is however only a small effect at 2° elevation angle). I furthermore think it would be more appropriate to use the observed dSCDs directly as measurement vector $y$ instead of (weighted) mean concentrations, which are not a very useful quantity. The weighting function as a function of wavelength $\lambda$ and distance $x$ to the box would then be

$$K(\lambda, x) = (1 - \exp(1\text{-}x/L(\lambda))) * \Delta x/\cos(\alpha)$$

with $\Delta x$ being the width of the boxes, $\alpha$ the elevation angle, and $L(\lambda)$ the effective light path estimated from the measurements. The forward model would then simply be

$$SCD(\lambda) = \sum_i K(\lambda, x_i) * c_i$$

with $x_i$ being the distance between observer and box $i$ and $c_i$ the NO2 concentration in this box. This equation can be readily inverted using OEM.

P29, L527ff: It cannot be 'seen' from Figure 11 that the weighting functions are constant up to a distance L, but instead this is an assumption, which is physically incorrect – see my comments above.

P29, L552: Here you mention CIMEL observations without explaining what the nature of these measurements are and where they have been perfromed. A short description of these measurements should be part of an extra Section on ancillary measurements further up in the manuscript – see general comments.

P29, L550ff: Do I understand it right that the linear decrease in a priori AOD is a decrease in the horizontal dimension? If so, what are the motivations for this assumption? A higher AOD at your measurement site than anywhere else in the surroundings is hard to justify (except if there were strong aerosol sources next to your instrument).

P30, L567: What kind of airborne observations are these? This is only explained later in the manuscript – I suggest to move the introduction of APEX from Section 5.2. to a Section further up in the manuscript describing all ancillary data used in this study (see general comments).

P30, L580ff, and Section 4.4: It is stated several times that there is no information on the horizontal distribution at distances closer than the shortest scattering distance. The averaging kernels are, however, not zero at these regions (see revised Fig. 15). Instead, it seems that parts of the information coming from short distances are falsely attributed to distances further away. For example, the 8.75 km averaging kernel has a constant value of 0.04 up to a distance of approx. 8 km. Unfortunately, no averaging kernels for distances closer to the instrument are shown.

Figure 15: Given the small peak values of the averaging kernels (at most 0.05), I wonder if the fine horizontal grid of 500 m is really useful or if a coarser grid would have been more appropriate. Furthermore, I could imagine that the averaging kernels would look more smoothly if more realistic (exponentially decreasing) weighting functions rather than the arbitrary step-like functions would have been used.

Section 5.3.1: Here you describe the methodology for comparison between the different datasets, but the NO2 gas maps based this method have already been shown in Section 5.2, if I understand it right. Is there something different in the data processing (filtering and spatio-temporal binning) for the production of Figure 21 compared to Figure 19 (except that Fig. 21 shows seasonal averages)? If not, then the description of this method should appear at the beginning of Section 5.2.

**Technical Corrections**

P6, L133: The O4 cross sections by Finkenzeller have recently been published [Finkenzeller and Volkamer., 2022]. Please add the according reference.

P8, L185: 'in six different fitting windows' - > 'in the six different fitting windows listed in Section 2.2.'

P9, L187: Explain abbreviations/acronyms 'OEM' and 'MMF'

P9, L189: Explain abbreviation 'MLH'

P9, L195: Here you should state that the NO2 near-surface concentrations and VCDs and the near-surface aerosol extinction are retrieved as a function of distance from the instrument.

P9, L197: Three times 'horizontal' in one sentence. Please rephrase.

Section 4.2: Equation (3) is just a trivial rearrangement of Equation (2) and therefore obsolete. Please remove one of these.

P16, L348: 'Regarding the aerosols' - > 'Regarding aerosols'

P23, L434: I suggest replacing the term 'sanity' with 'consistency'

Equation 14: I suggest to replace $dx$ with $\Delta x$ since $dx$ can be confused with the differential in Equation 11.

P30, L579: Insert a comma before the year number.

P40, L670: Replace 'error' with 'covariance matrix', and mention that the error is given as the square root of its diagonal elements.

Caption of Figure 16: 'measurement error' $\rightarrow$ 'measurement and smoothing error'

Figure 18: The bottom-right panel should have the same width as the other panels.

Title of Section 5.3: 'Comparison between MAX-DOAS horizontal NO2 distribution and TROPOMI observations'

P70, L1105: The approach is better than what?

P70, L1108: Do you mean a slope closer to unity?

P70, L1114: Delete this paragraph on the role of the a priori as it refers to a Section that has been removed in the revised manuscript.

**References**

Finkenzeller, H. and Volkamer, R.: O2–O2 CIA in the gas phase: Cross-section of weak bands, and continuum absorption between 297–500 nm, 279, 108063, https://doi.org/https://doi.org/10.1016/j.jqsrt.2021.108063, 2022.

Kern, C., Deutschmann, T., Vogel, L., Wöhrbach, M., Wagner, T., and Platt, U.: Radiative transfer corrections for accurate spectroscopic measurements of volcanic gas emissions, 72, 233–247, https://doi.org/10.1007/s00445-009-0313-7, 2010.

Vogel, L., Galle, B., Kern, C., Delgado Granados, H., Conde, V., Norman, P., Arellano, S., Landgren, O., Lübcke, P., Alvarez Nieves, J. M., Cárdenas Gonzáles, L., and Platt, U.: Early in-flight detection of $SO_2$ via Differential Optical Absorption Spectroscopy: a feasible aviation safety measure to prevent potential encounters with volcanic plumes, 4, 1785–1804, https://doi.org/10.5194/amt-4-1785-2011, 2011.

---

## Author Response (AR2)

Dear Thomas von Clarmann,

We thank you for your valuable contribution to ameliorating the present paper.

Following the recommendation of the third reviewer concerning the weighting functions, we have decided to use the proposed formula to estimate the weighting functions, as it is more physically more adequate. Please note that all the analysis and, consequently, Figures 11 to 23 and S5 to S11 have been modified.

In the present document, we list a point-to-point response to the reviews.

**Please find below our responses to each comment individually.**
**Please consider that:**
A) **Green bold: Comments of the Referee**
B) Black: Response to each comment addressed by the referee.
C) **Black bold: Already existing text in the manuscript.**
D) **Red bold: Added/corrected text in the manuscript according to referee's comments.**

**Response to Anonymous Referee #1**

1. **Thanks for adding more examples to Fig 13. and 14. Please discuss the new examples in 2-3 sentences. Especially the larger deviation of a priori and retrieval result in Fig. 14 is interesting. Would you consider the a priori information as good enough?**

   **Response:** Please consider the following additional discussion concerning Fig. 13 and 14:

   **Three examples of the retrieved NO$_2$ horizontal profile are presented in Fig. 13, together with corresponding measured and simulated $\bar{c}_{NO2}$ at the six different wavelengths for July 2, 2018 (fig. 13a; low NO2 abundance condition), September 11, 2018 (fig. 13b; medium NO2 abundance condition), and September 30, 2018 (fig. 13c; medium NO2 abundance condition). RMS is calculated between measured and simulated NO$_2$ near-surface concentrations of the horizontal retrieval normalized by the mean of the measured NO$_2$ near-surface concentrations (upper panels in Fig. 13).. As the NO$_2$ values become larger, the agreement between measured and simulated $\bar{c}_{NO2}$, expressed via the RMS value, is improved.**

   **Similarly, examples of measured and retrieved near-surface aerosol extinction coefficient and retrieved aerosol horizontal profile are shown in Fig. 14, for different aerosol load conditions (low (Fig. 14a), medium (Fig. 14b) and high (Fig. 14c)) over the Brussels-Capital Region. We observe that the agreement between simulated and measured near-surface aerosol extinction coefficient at the six different wavelength tends to be worse than for NO$_2$. This could be due to the use of a constructed (constant)**

a priori aerosol horizontal profile due to the lack of information on the aerosol extinction horizontal distribution in the Brussels-Capital region.

2. **Please add a small discussion of the smoothing error in Fig 16.**

**Response:** A small discussion of the smoothing error in Fig. 16 has been added as follows:

The horizontal profiles of the measurement and smoothing error in percentage are shown in Fig. 16. As can be seen, the smoothing error is significantly larger than the measurement error (range of 3%-10% and 14%-40%, respectively). The smoothing error becomes also larger as the horizontal distances from the instrument become larger. This is mainly because of the exponential decrease of the sensitivity as a function of the horizontal distance (see weighting functions in Fig. 11), and consequently, the larger impact of the difference between the a priori profile and the true state of the atmosphere.

3. **Thanks for adding the validation Section S1 in the supplement. Please refer to this section in the main manuscript.**

**Response:** We are referring the reader to Section S1 in Section 5.3 in the following point:

A mean scaling factor equal to the mean ratio between the measured and RIO $NO_2$ near-surface concentrations is applied because of the systematic underestimation of $NO_2$ near-surface concentrations by MAX-DOAS when compared to in-situ measurements (see Dimitropoulou et al., 2020 and Section S1).

**Response to Anonymous Referee #3**

**General comments**

1. **A main problem of the manuscript is that the description of the methods is lacking conciseness, and it is not always clear whether certain approaches are based on physical principles or rather represent simplifying assumptions. In particular, the description of the horizontal distribution inversion approach (Section 4.3.) conveys the impression that a constant weighting function as a function of distance with a sharp drop at the estimated light path length is a fact, while this is in actually a very simplifying and physically incorrect assumption. In reality, I would rather expect that the sensitivity decreases exponentially with distance from the observer in accordance with the Beer-Lambert law, as already discussed by Kern et al. [2010] and Vogel et al. [2011]. In the light of these (over-) simplifications, it is surprising to see that the retrieved horizontal distribution compares very well with airborne and satellite measurements.**

**Response:** We would like to thank the reviewer for rising this issue concerning the weighting function of the inversion approach. Please consider our response to specific comment #1.

2. **I wonder if it is not possible to infer the correction factor discussed in Section 4.2 directly from the aerosol and NO2 profiles retrieved by MMF, which directly provide dSCD(NO2) and c(NO2) and thus L(NO2) as the ratio of both.**

   **Response:** The experimental set-up in Uccle contains 10 different azimuthal directions. In order to ensure that L(NO2) is appropriate for each of them under different viewing geometries (i.e., SZA, RAA, and elevation angle), we preferred to perform simulations separately from the OEM-based MMF algorithm, which can only retrieve information in the main azimuthal direction in which elevation scans are performed.

3. **Some of the measurements used for intercomparison are not explained, or only later in the manuscript. For example, it is not clear what CIMEL measurements are, and airborne measurements are mentioned already in Section 4.3, but the context is only given much later in the manuscript in Section 5.2. I suggest to add a short Section (4) with a short description of the ancillary data used for intercomparison (CIMEL, APEX, car DOAS, etc.).**

   **Response:** In the revised version of the manuscript, we have added a new Section (**4.Ancillary measurements**), in which short descriptions of CIMEL, APEX, RIO, in-situ and car-mobile DOAS measurements are presented.

4. **The level of agreement between the horizontal distributions of NO2 from MAX-DOAS on the one hand and from satellite, car- and airborne measurements on the other hand presented in Section 5.2. is quite impressive. However, I recommend a major revision of the manuscript due to significant deficiencies in the methodology and in the description of the inversion approach. The inversions of the horizontal distribution need to be re-done using physically correct weighting functions.**

   **Response:** Please consider our response about the issue that you have raised concerning the weighting functions in the Specific Comment #1.

**Specific Comments**

1. **Section 4.3: I find the description of the forward model quite confusing, and I feel that this Section requires substantial revision. Moreover, I think the inversion needs to be re-done with appropriate weighting functions. Equation 12 suggests that the forward model for the calculation of a mean concentration is given as the integral over the concentration divided by the light path length. First of all, I guess that the model is not based on numerical integration, but that it is rather based on a discrete sum over the horizontal grid. Second, what is missing in this equation is an appropriate weight as provided by a weighting function that represent correct physics. The assumption of a constant sensitivity between the instrument and the effective light path length L is not**

realistic. Instead, an exponential decrease should be chosen as weighting function in accordance with the Beer-Lambert law. Also, the light path through the boxes is not horizontal but slanted, so the weighting function needs to contain the cosine of the elevation angle (this is however only a small effect at 2° elevation angle). I furthermore think it would be more appropriate to use the observed dSCDs directly as measurement vector $y$ instead of (weighted) mean concentrations, which are not a very useful quantity. The weighting function as a function of wavelength $\lambda$ and distance $x$ to the box would then be

$$K(\lambda, x) = (\exp(\text{-}x/L(\lambda))) * \Delta x/\cos(\alpha)$$

with $\Delta x$ being the width of the boxes, $\alpha$ the elevation angle, and $L(\lambda)$ the effective light path estimated from the measurements. The forward model would then simply be

$$F(\lambda) = \Sigma(\lambda, x_i) * c_i$$

with $x_i$ being the distance between observer and box $i$ and $c_i$ the NO2 concentration in this box. This equation can be readily inverted using OEM.

**Response:** After the issue raised with the weighting functions, we have decided to re-do the inversion with appropriate weighting functions.

First, we confirm that the model is based on a discrete sum over the horizontal grid. Second, we agree that the assumption of a constant sensitivity (i.e., constant weighting function) between the instrument and L and no sensitivity beyond L is likely not the most realistic one . For this reason, we have modified the weighting function as suggested:

$$K(\lambda, x) = [\exp(\text{-}x/L(\lambda))] * \Delta x/\cos(\alpha)$$

By using these weighting function, L(NO2) corresponds to an effective distance, which can be interpreted as the distance at which the sensitivity drops to 1/e. Beyond L(NO2), the sensitivity continues to reduce. In the revised version of this paper, we have decided to consider also distances beyond L(NO2) in the inversion with the maximum distance being defined as the distance where the sensitivity is 10% or less of the sensitivity close to the measurement site.

Consequently, we have performed again the analysis with the new weighting functions and Figures (11 to 23 and S5-S11) and Tables (3 to 4) have been modified accordingly. The main changes and improvements/degradations in the results are the following:

- An increase in the retrieved $NO_2$ near-surface horizontal profile
- An increase in the RMS between the measured and retrieved $NO_2$ near-surface concentrations

- A decrease in the DOFs values of the retrieved NO$_2$ near-surface horizontal profile
- A change in the horizontal shape of the averaging kernels, measurement error and smoothing error
- A better agreement between tropospheric NO2 columns derived by the MAX-DOAS observations and the airborne, car mobile-DOAS and satellite observations on 28$^{th}$ of June 2019
- An improvement of the correlation coefficient value and a degradation of the slope values in the seasonal comparisons between TROPOMI and MAX-DOAS tropospheric NO2 columns

2. **P29, L527ff: It cannot be 'seen' from Figure 11 that the weighting functions are constant up to a distance L, but instead this is an assumption, which is physically incorrect – see my comments above.**
**Response:** Please consider our response to your comment above. The sentence has been modified as follows:

The sensitivity decreases exponentially up to a distance corresponding to the differential effective light path length of each measurement. More precisely, each measurement is highly sensitive to the MAX-DOAS instrument location. This sensitivity decreases exponentially as a function of the horizontal distance. Then, it reaches a value equal to 1/e to the horizontal distance equal to the differential effective light path length of each measurement.

3. **P29, L552: Here you mention CIMEL observations without explaining what the nature of these measurements are and where they have been perfromed. A short description of these measurements should be part of an extra Section on ancillary measurements further up in the manuscript – see general comments.**
**Response:** After considering General comment #3, a Section (now, Section 4 – Ancillary measurements) has been added.

4. **P29, L550ff: Do I understand it right that the linear decrease in a priori AOD is a decrease in the horizontal dimension? If so, what are the motivations for this assumption? A higher AOD at your measurement site than anywhere else in the surroundings is hard to justify (except if there were strong aerosol sources next to your instrument).**
**Response:** Indeed, as an a priori AOD horizontal profile, we have chosen a horizontally decreasing profile. Given the fact that there are no strong aerosol sources close to the instrument, in the revised version of the manuscript, we now use a horizontally constant a priori AOD profile.

5. **P30, L567: What kind of airborne observations are these? This is only explained later in the manuscript– I suggest to move the introduction of APEX from Section 5.2. to a Section further up in the manuscript describing all ancillary data used in this study (see general comments).**
**Response:** Please consider our response to General comment #3.

6. **P30, L580ff, and Section 4.4: It is stated several times that there is no information on the horizontal distribution at distances closer than the shortest scattering distance. The averaging kernels are, however, not zero at these regions (see revised Fig. 15). Instead, it seems that parts of the information coming from short distances are falsely attributed to distances further away. For example, the 8.75 km averaging kernel has a constant value of 0.04 up to a distance of approx. 8 km. Unfortunately, no averaging kernels for distances closer to the instrument are shown.**
**Response:** As expected, the use of the new weighting functions, (see Fig. 11) have an impact to the retrieved NO2 horizontal profile and the form of the averaging kernels. In Figure 15 of the revised manuscript, we observe that the AKs for distances closer to the instrument (d=2.25 km and d=4.75 km) have the maximum values and decrease exponentially as a function of the horizontal distance from the instrument. Consequently, the statement that there is no information on the horizontal distribution at distances closer than the shortest scattering distance is not correct and has been removed from the revised manuscript.

7. **Figure 15: Given the small peak values of the averaging kernels (at most 0.05), I wonder if the fine horizontal grid of 500 m is really useful or if a coarser grid would have been more appropriate. Furthermore, I could imagine that the averaging kernels would look more smoothly if more realistic (exponentially decreasing) weighting functions rather than the arbitrary step-like functions would have been used.**
**Response:** The information content of the measurements (DOFs) determine the choice of the retrieval grid. For our retrieval, the DOFs are generally larger than unity and smaller than two. This means that we can only retrieve the NO2 near-surface concentration in 1-2 horizontal boxes. In practice, in the atmospheric remote sensing, we use much thinner retrieval grid. During the development of the horizontal OEM-based inversion approach, we have tested the use of wider horizontal boxes and we found that this choice does not have a considerable effect on the NO2 horizontal retrieval.

The use of more realistic weighting functions have a significant impact on the form of the averaging kernels, which are smoother and show maximum values for distances closer to the instrument.

 **Section 5.3.1: Here you describe the methodology for comparison between the different datasets, but the NO2 gas maps based this method have already been shown in Section 5.2, if I understand it right. Is there something different in the data processing (filtering and spatio-temporal binning) for the production of Figure 21 compared to Figure 19 (except that Fig. 21 shows seasonal averages)? If not, then the description of this method should appear at the beginning of Section 5.2.**

**Response:** In Section 5.3.1 and 5.2 (now, Section 6.3.1 and 6.2), the NO2 maps for the Brussels-Capital Region have been produced by using the same data processing. Please consider the following modification in the revised paper (Section 6.3.1):

To compare the TROPOMI and MAX-DOAS tropospheric NO₂ columns, the similar approach is used as in Section 6.2. **Additionally, TROPOMI and MAX-DOAS tropospheric NO₂ columns are compared** in a seasonal basis**, and the seasonally-averaged maps of those VCDs on the area covered by the TROPOMI pixels are created. To generate these maps, the ensemble of TROPOMI pixels recorded on 28 June 2019 is chosen as reference and TROPOMI pixels that coincide with this reference grid are averaged. The daily horizontal profiles of MAX-DOAS NO₂ columns are averaged on the daily TROPOMI grids and then, the reference grid is used to create the seasonally-averaged MAX-DOAS maps.**

**Technical Corrections**

1. **P6, L133: The O4 cross sections by Finkenzeller have recently been published [Finkenzeller and Volkamer., 2022]. Please add the according reference.**
   **Response:** The reference has been added to the manuscript.

2. **P8, L185: 'in six different fitting windows' - > 'in the six different fitting windows listed in Section 2.2.'**
   **Response:** The sentence has been modified accordingly.

3. **P9, L187: Explain abbreviations/acronyms 'OEM' and 'MMF'**
   **Response:** The abbreviations/acronyms are explained in the revised version of this paper.

4. **P9, L189: Explain abbreviation 'MLH'**
   **Response:** The abbreviations is explained.

5. **P9, L195: Here you should state that the NO2 near-surface concentrations and VCDs and the near-surface aerosol extinction are retrieved as a function of distance from the instrument.**
   **Response:** The sentence is modified as follows:

Then, in the next step, a new dual-scan parameterization technique is applied to the $O_4$ and $NO_2$ dSCDs at the six different wavelengths and in all the azimuthal directions with $MLH_{NO2}$, measured $O_4$ dSCDs, and measurement geometry being the main input parameters to retrieve the horizontal sensitivity of $NO_2$ and, consequently, the $NO_2$ near-surface concentrations and VCDs, and near-surface aerosol extinction as a function of distance from the instrument (see Section 5.2).

6. **P9, L197: Three times 'horizontal' in one sentence. Please rephrase.**
**Response:** The phrase has been rephrased as follows:
In the final step, a new OEM-based horizontal distribution inversion approach is developed using the six near-surface $NO_2$ concentrations and aerosol extinction values per azimuthal direction to retrieve $NO_2$ and aerosol extinction horizontal profiles in an output grid of 500m thickness (see Section 5.3).

7. **Section 4.2: Equation (3) is just a trivial rearrangement of Equation (2) and therefore obsolete. Please remove one of these.**
**Response:** Equation (3) has been removed and the text has been adjusted as follows:
$L_{O_4}$ is calculated by using Eq. 2 for $O_4$.

8. **P16, L348: 'Regarding the aerosols' - > 'Regarding aerosols'**
**Response:** The text has been modified according to this comment.

9. **P23, L434: I suggest replacing the term 'sanity' with 'consistency'**
**Response:** The term sanity is replaced by the term consistency.

10. **Equation 14: I suggest to replace $dx$ with $\Delta x$ since $dx$ can be confused with the differential in Equation 11.**
**Response:** dx is replace with Δx in Equation 14 (now, Equation 13).

11. **P30, L579: Insert a comma before the year number.**
**Response:** A comma has been inserted before the year number.

12. **P40, L670: Replace 'error' with 'covariance matrix', and mention that the error is given as the square root of its diagonal elements.**
**Response:** The word 'error' has been replaced with 'covariance matrix'. Additionally, the following sentence has been added:
Then, the retrieval noise error is given as the square root of the diagonal elements of the noise covariance matrix.

**13. Caption of Figure 16: 'measurement error' → 'measurement and smoothing error'**
**Response:** The caption of Figure 16 has been modified accordingly.

**14. Figure 18: The bottom-right panel should have the same width as the other panels.**
**Response:** Figure 18 has been modified accordingly.

**15. Title of Section 5.3: 'Comparison between MAX-DOAS horizontal NO2 distribution and TROPOMI observations'**
**Response:** The title of Section 5.3 (now, Section 6.3) has been modified.

**16. P70, L1105: The approach is better than what?**
**Response:** The word 'better' has been replaced by the word 'good'.

**17. P70, L1108: Do you mean a slope closer to unity?**
**Response:** Indeed, we do mean a slope closer to unity. The text has modified accordingly.

**18. P70, L1114: Delete this paragraph on the role of the a priori as it refers to a Section that has been removed in the revised manuscript.**
**Response:** This paragraph has been deleted from the manuscript.

A list of all relevant changes made in the manuscript is presented below. Please consider also the uploaded manuscript using track changes in Word.

**List of changes in the manuscript**

(Page 1, Line 22), (Page 3, Line 70-76), (Page 5, Line 131), (Page 8, Line 181-204), (Page 8-9, Line 205-212), (Page 9, Line 217-219), (Page 10, Line 227-228), (Page 10, Line 251), (Page 10, Line 253), (Page 11, Line 274-279), (Page 12, Line 297), (Page 12, Line 306-316), (Page 13, Line 323-326), (Page 13, Line 331), (Page 13, Line 337), (Page 14, Line 365), (Page 17-18, Line 413-415), (Page 18, Line 441-444), (Page 21, Line 473-480), (Page 21, Line 487), (Page 21-22, Line 494-503), (Page 22-23, Line 511-532), (Page 23, Line 537-554), (Page 23-24, Line 559-491), (Page 25, Line 600), (Figure 11), (Figure 13), (Figure 14), (Page 32-35, Line 655-710), (Figure 15), (Figure 16), (Page 38, Line 727-733), (Figure 17), (Figure 18), (Page 43, Line 771-776), (Page 43-44, Line 791-803), (Page 44, Line 810-814), (Figure 19), (Figure 20), (Page 48-49, Line 849-911), (Figure 21), (Figure 22), (Figure 23 and legend), (Table 4), (Page 57, Line 993), (Page 58, Line 1024-2043), (Page 60, Line 1095), and (Page 60, Line 1115-1119).

Best regards,

Ermioni Dimitropoulou